# HAP-E: Hessian-Aware Structured Pruning of LLMs for Efficient Inference

## Abstract

Large language models (LLMs) deliver strong performance across diverse tasks, yet their heavy compute and memory demands make deployment on real-time edge devices challenging. Structured pruning has become the standard approach to reduce these costs, yet accurately estimating which blocks can be removed remains challenging at scale. Second-order methods such as Optimal Brain Surgeon (OBS) are computationally intractable at LLM scale. Existing approaches rely on static budgets that ignore cross-layer dependencies, and common proxies like FLOPs misestimate real hardware latency. We introduce *HAP-E*, a scalable, Hessian-aware pruning framework for post-training compression of LLMs. *HAP-E* adaptively reallocates budgets across layers using global screening and selective second-order analysis on a candidate set guided by cross-layer sensitivity estimation. It further performs OBS-equivalent batch pruning that certifies and removes multiple blocks at once while exactly matching the greedy OBS sequence, thereby reducing weight updates and numerical drift. A lightweight latency predictor ensures that the compressed model satisfies inference-time constraints. Experiments on LLaMA and OPT models show that *HAP-E* improves accuracy by up to 3% over state-of-the-art structured pruning methods at comparable pruning ratios.

## 1 Introduction

Large language models (LLMs) have achieved state-of-the-art performance across diverse tasks (Bommasani, 2021), but their substantial computational and memory demands hinder deployment in latency-sensitive or resource-constrained environments (Zhou et al., 2024). In such scenarios, achieving low inference latency, high energy efficiency, and preserving data privacy are critical requirements (Wu et al., 2019). Structured pruning (Guo et al., 2025; Kwon et al., 2022; An et al., 2024), which removes entire blocks such as attention heads or feed-forward neurons, has become the standard approach to reduce these costs. By aligning naturally with existing hardware and inference frameworks, structured pruning directly translates into tangible latency and memory reductions. The central challenge lies in accurately estimating which blocks can be removed with minimal impact, a problem that becomes increasingly difficult at the scale of modern LLMs (Kim et al., 2024; Frantar & Alistarh, 2023).

Recent advances have shown that second-order (Hessian-based) information, as in Optimal Brain Surgeon (OBS) (Hassibi & Stork, 1992; Frantar & Alistarh, 2022) inspired methods (Ling et al., 2024; Wei et al., 2024), can effectively guide pruning by capturing the curvature of the loss landscape and making locally optimal choices. However, existing approaches face four major challenges at LLM scale: (1) Computing and updating full Hessian inverses is memory- and compute-intensive, and even incremental strategies require repeated inverse updates that are costly and introduce numerical drift. (2) Standard OBS prunes one block at a time, demanding many sequential weight updates that slow pruning and make it impractical for billion-parameter models. (3) Conventional methods impose static, layer-wise pruning budgets fixed at the start of pruning, ignoring cross-layer dependencies where pruning in one layer alters the importance of blocks in subsequent layers. (4) Finally, they rely on proxy metrics such as FLOPs or sparsity ratios, which misrepresent real hardware latency and require repeated tuning (Kurtić et al., 2023).

**Contributions.** To address these challenges, this paper introduces *HAP-E*, a scalable, Hessian-aware structured pruning framework for post-training compression of LLMs. (i) *HAP-E* adaptively

reallocates pruning budgets across layers. To do so, it first performs inexpensive global screening and then applies selective second-order analysis on a candidate set chosen dynamically. This process is guided by cross-layer sensitivity estimation that captures both local and propagated effects (Section 4.2). (ii) It introduces a greedy-consistent batch pruning mechanism. Each certified batch matches exactly the sequence that greedy OBS would remove one-by-one, but is pruned jointly in a single step. This yields the same accuracy with far fewer weight updates, reducing computational overhead and mitigating numerical drift (Section 4.1). (iii) Finally, *HAP-E* integrates a lightweight latency predictor into the pruning loop to ensure that the compressed model meets real inference-time constraints (Section 4.3). Together, these components make OBS-style pruning tractable at LLM scale, delivering up to 3% higher accuracy than state-of-the-art pruning methods at comparable pruning ratios.

## 2 BACKGROUND: STRUCTURED OPTIMAL BRAIN SURGEON PRUNING

Given a small calibration dataset, we first collect representative input activations for each layer. Consider a linear layer with input activations $X \in \mathbb{R}^{T \times C_{\text{in}}}$ and weight matrix $W \in \mathbb{R}^{C_{\text{in}} \times C_{\text{out}}}$, where $T$ is the number of input tokens, and $C_{\text{in}}, C_{\text{out}}$ are the input/output dimensions, respectively. The structured pruning objective seeks compressed weights $\tilde{W}$ that approximate the original output under a predefined structural constraint $\mathcal{C}$:

$$\min_{\tilde{W} \in \mathcal{C}} \|XW - X\tilde{W}\|_F^2 \tag{1}$$

Let $H = X^\top X + \lambda I$ denote the Hessian of Equation 1, where $\lambda$ is a small positive constant to improve numerical stability (Hassibi & Stork, 1993; Frantar & Alistarh, 2022). Suppose $S \subseteq \{1, \ldots, C_{\text{out}}\}$ denotes the indices of a candidate block containing $k$ columns of weights ($|S| = k$), e.g., sets of columns corresponding to an attention head. For any matrix $A$ and index set $S$, $A_{S,:}$, $A_{:,S}$, and $A_{S,S}$ denote row, column, and submatrix restrictions, respectively. The OBS then provides closed-form solutions for the minimal error ($E(S)$) incurred by the pruning block $S$, along with the optimal update $\Delta_S$ applied to the remaining weights (Frantar & Alistarh, 2022; 2023):

$$E(S) = \sum_{i=1}^{C_{\text{in}}} W_{i,S}\big((H^{-1})_{S,S}\big)^{-1}W_{i,S}^\top \tag{2}$$

$$\Delta_S = -W_{:,S}\big((H^{-1})_{S,S}\big)^{-1}(H^{-1})_{S,:} \tag{3}$$

To account for inter-block correlations, vanilla structured OBS pruning typically removes blocks sequentially (Chen & et al., 2024; Li, 2024). At each step, it selects the block $S$ with the smallest $E(S)$, applies $\Delta_S$ the remaining weights, and updates $H^{-1}$ using Gaussian elimination rather than recomputing it from scratch:

$$H^{-1} \leftarrow H^{-1} - (H^{-1})_{:,S}\big((H^{-1})_{S,S}\big)^{-1}(H^{-1})_{S,:} \tag{4}$$

This iterative approach yields locally optimal structured pruning decisions while maintaining computational efficiency.

While theoretically appealing, such an approach is impractical for LLM-scale layers: storing and updating $H^{-1} \in \mathbb{R}^{d \times d}$ incurs $\mathcal{O}(d^2)$ memory and $\mathcal{O}(kd^2)$ update cost, and each candidate additionally requires submatrix extraction and inversion with no amortization. Repeated Gaussian elimination downdates further introduce numerical drift, degrading importance accuracy, and weight updates. These issues make naïve structured OBS infeasible for layers with tens of thousands of columns, motivating a redesign that *localizes* Hessian storage, *batches* updates, and *avoids* touching the full inverse at every step.

## 3 RELATED WORK

Given the substantial computational and memory demands of LLMs, numerous compression techniques have been explored, such as pruning (Zhang et al., 2024;?; Ma et al., 2023), quantization (Lin et al., 2024), and low-rank decomposition (Yuan et al., 2023), to enable efficient deployment. Unstructured pruning methods, including SparseGPT (Frantar & Alistarh, 2023), Wanda (Sun et al.,

2023), and E-Sparse (Li et al., 2023), remove individual weights based on criteria such as Hessian-based importance, combined weight–activation statistics, or information entropy. Although effective in reducing parameters, these methods often require specialized hardware or software to achieve latency gains, limiting their practical applicability (Ashkboos et al., 2024).

**OBS-based Structured Pruning.** OBS-based structured pruning leverages second-order information to minimize post-pruning reconstruction error, offering strong theoretical guarantees and empirical performance. SlimGPT (Ling et al., 2024) extends OBS to structured settings via grouped Cholesky decomposition, which efficiently computes the joint importance of all columns within a block, e.g., an attention head. However, it still requires updating the Hessian inverse and weight magnitudes after pruning, inheriting OBS's scaling bottlenecks. It further mitigates performance loss through incremental, non-uniform layer-wise pruning rates, but remains fundamentally layer-local and static in budget allocation, limiting its ability to exploit global inter-layer dynamics. SoBP (Wei et al., 2024) uses global importance scores from first-order Taylor expansions to assign fixed pruning ratios across layers, followed by local greedy refinement. Yet, its static allocation cannot adapt during pruning, hindering its ability to capture evolving sensitivities and cross-layer interactions.

**Low-Rank Decomposition.** Low-rank methods, such as LoRD (Kaushal et al., 2024), ASVD (Yuan et al., 2023), and recent advancements like MoDeGPT (Lin et al., 2025) and SVD-LLM (Wang et al., 2025b;a), reduce parameter counts by approximating weight matrices via SVD or related techniques. While effective for memory reduction, they typically achieve less latency reduction than structured pruning on standard hardware. Furthermore, many of these approaches require substantial retraining to restore accuracy, posing challenges for large-scale deployment.

**Global and Adaptive Pruning.** Recent works have explored global sparsity allocation to mitigate cross-layer mismatch. Approaches such as OWL (Yin et al., 2024) and SparseLLM (Bai et al., 2024) formulate global objectives; however, they typically rely on static or one-shot sensitivity metrics computed prior to pruning. Similarly, global gradient-based methods like LLM-Pruner (Ma et al., 2023) and GBLM (Das et al., 2023) utilize first-order Taylor approximations to estimate importance, but typically fix the pruning mask or sparsity ratios at initialization. Because these budgets are determined prior to pruning, they cannot capture how the loss landscape and parameter importance evolve as weights are removed. Furthermore, calculating these global gradients in methods like GBLM requires backpropagation through the entire network, which incurs prohibitive memory costs for large models. Other approaches, such as ECoFLaP (Sung et al., 2024), adopt coarse-to-fine strategies driven by zeroth-order heuristics, but likewise lack explicit second-order curvature updates. Finally, evolutionary methods such as DarwinLM (Tang et al., 2025) depend on search over a precomputed configuration database, which introduces computational overhead and is limited by the static nature of the database.

**Positioning of this work.** *HAP-E* advances OBS-based pruning along three axes. (i) It allocates candidates adaptively across layers via recursive, second-order sensitivity, enabling dynamic budget reallocation to capture evolving curvature and overcoming static or first

---

**Algorithm 1:** *HAP-E* Pruning Framework

---

**Require:** $M$ (pre-trained model), $Lat_{\text{target}}$, $D_{\text{cal}}$
**Ensure:** $M_{\text{pruned}}$
Measure $Lat(M)$
**while** $Lat(M) > Lat_{\text{target}}$ **do**
  ▷ **1. Lightweight importance estimation**
  $\text{Imp}(B_i) \leftarrow \sqrt{\frac{1}{|W_i|} \sum_{w \in W_i} w^2}$
  ▷ **2. Layer sensitivity estimation (recursive)**
  $S^{(\ell) \to (\ell+1)} \leftarrow \text{Tr}((X^{(\ell+1)})^\top X^{(\ell+1)} + \lambda I)$
  $S^{(\ell)} = S^{(\ell) \to (\ell+1)} + \beta S^{(\ell+1)}$
  ▷ **3. Candidate budget allocation**
  $\text{CV}^{(\ell,\tau)} \leftarrow \sigma^{(\ell,\tau)} / \mu^{(\ell,\tau)}$
  $K^{(\ell,\tau)} \leftarrow \min(CK, N^{(\ell,\tau)}) \cdot \frac{\text{CV}^{(\ell,\tau)}}{S^{(\ell)} + \varepsilon}$
  ▷ **4. OBS scoring with partial inverse**
  Solve $HX = E_\Pi$ for candidate panel $\Pi$
  $G_{\Pi,\Pi} \leftarrow (G_{:,\Pi})^\top E_\Pi$;
  $E(B_S) \leftarrow \sum_j W_{S,j}^\top (G_{SS})^{-1} W_{S,j}$;
  $\widetilde{E}(B_S) \leftarrow S^{(\ell)} \cdot E(B_S)$
  Select $K$ blocks with smallest $\widetilde{E}$
  ▷ **5. Certify greedy-consistent batch and prune**
  $A_c' \leftarrow G_{cc} - G_{cJ} G_{JJ}^{-1} G_{Jc}$;
  $\mathcal{E}'(c \mid J) \leftarrow \|(A_c')^{-1/2} W_{c,:}\|_F^2$;
  Grow $J$ by repeatedly adding $c^\star$; set $P \leftarrow J$
  $\Delta W_R \leftarrow -H_{RP} H_{PP}^{-1} W_P$; $W_{P,:} \leftarrow 0$
  ▷ **6. Incremental Hessian update (sub-block only)**
  $Q \leftarrow \Pi \setminus P$;
  $G_{QQ}' \leftarrow G_{QQ} - G_{QP} G_{PP}^{-1} G_{PQ}$
  ▷ **7. Latency update**
  Measure $Lat(M)$
**end while**
**return** $M_{\text{pruned}}$

---

-order heuristics. (ii) It integrates a hardware-calibrated latency predictor directly into the pruning loop, ensuring that pruning decisions satisfy real device constraints without relying on proxy metrics or repeated sweeps as in prior methods. (iii) It introduces a greedy-consistent batch pruning mechanism with theoretical guarantees, certifying equivalence to the sequential OBS solution while requiring far fewer updates.

## 4 PROPOSED METHOD

We propose *HAP-E*, an adaptive, Hessian-aware structured pruning framework that compresses large language models to meet a user-specified hardware latency target while maintaining accuracy. The method is entirely post-training and operates in an iterative loop, progressively removing the least important structural blocks until the measured latency satisfies the constraint.

As shown in Algorithm 1, at a high level, each iteration of *HAP-E* proceeds in four stages: (1) *Lightweight importance estimation*: assign each block, e.g., attention head and FFN neuron, an inexpensive saliency score based on parameter magnitude. (2) *Sensitivity analysis*: estimate the tolerance of each layer to perturbations via a recursive Hessian-based approximation that captures both local and propagated effects. (3) *Candidate selection and refinement*: allocate a candidate budget across layers according to sensitivity and variability, then refine these candidates using exact OBS scores computed efficiently from partial Hessian solves. (4) *Greedy-consistent batch pruning*: certify the largest set of blocks that greedy OBS would remove sequentially, then prune them jointly in a single step. This guarantees equivalence to the one-by-one greedy OBS sequence while requiring far fewer weight updates, followed by an incremental update of the relevant Hessian sub-blocks.

By combining coarse-grained heuristics for global ranking with selective, exact OBS for a small candidate subset, *HAP-E* concentrates expensive second-order computation where it yields the most benefit, avoids full Hessian recomputation, and terminates as soon as the latency target is achieved. This yields a hardware-aware, scalable pruning algorithm that achieves high accuracy under strict inference budgets.

### 4.1 HYBRID OPTIMAL BRAIN SURGEON FOR BATCHED PRUNING

To make second-order pruning tractable for LLMs, we introduce a method that preserves the exactness of greedy OBS while avoiding its prohibitive computational and memory costs. Our approach prunes *batches* of blocks at once, but guarantees that each batch coincides with the initial segment of the greedy OBS sequence, that is, the same set of blocks that greedy OBS would have removed sequentially up to that point. In this way, pruning them jointly yields exactly the same weights and accuracy as performing greedy OBS step-by-step, while requiring far fewer weight updates.

**Notation.** Consider a depth-2 linear layer, e.g., the output projection of an MHA block, in module $\tau \in \{\text{MHA}, \text{FFN}\}$ of Transformer layer $\ell$, with input dimension $d$ (number of input columns) and block size $k$ (number of columns per structural block). Let the current candidate set be $K^{(\ell,\tau)} = \{c_1, \ldots, c_m\}$, where $m = |K^{(\ell,\tau)}|$ is the number of candidate blocks selected for module $\tau$ of layer $\ell$ in the current iteration, and each $c_j$ represents $k$ input columns (e.g., an attention head in MHA). We define the panel index $\Pi(K^{(\ell,\tau)}) \subseteq \{1, \ldots, d\}$ as the union of the column indices belonging to $K^{(\ell,\tau)}$, with panel size $|\Pi^{(\ell,\tau)}(K^{(\ell,\tau)})| = mk$. Let $W \in \mathbb{R}^{d \times C_{\text{out}}}$ be the weight matrix of this linear layer, where $C_{\text{out}}$ is the output dimension. Let $H$ be the Hessian from Equation 1, and define $G := H^{-1}$ as its inverse. The subscripts in these matrices, e.g., $H_{R,P}$, denote submatrices formed by selecting the row and column indices corresponding to index sets $R$ and $P$.

**Panel construction via selective inverse computation.** The primary bottleneck of OBS is explicitly forming the $d \times d$ inverse Hessian $G$. We circumvent this by computing only the columns of $G$ relevant to our candidate set $K_i$:

$$H X = E_{\Pi(K^{(\ell,\tau)})}, \qquad X \in \mathbb{R}^{d \times (mk)}, \tag{5}$$

where $E_{\Pi(K^{(\ell,\tau)})}$ is the matrix selecting the panel indices. The solution $X = G_{:,\Pi(K^{(\ell,\tau)})}$ contains the required columns, from which we extract the inverse panel $G_{K^{(\ell,\tau)},K^{(\ell,\tau)}} \equiv G_{\Pi(K^{(\ell,\tau)}),\Pi(K^{(\ell,\tau)})} \in \mathbb{R}^{(mk) \times (mk)}$. This reduces memory from $\mathcal{O}(d^2)$ to $\mathcal{O}(dmk)$. In terms of computation, a full Cholesky factorization of $H$ costs $\mathcal{O}(d^3)$, whereas solving Equation 5 for $mk$

columns with a pre-computed Cholesky factor costs $\mathcal{O}(d^2 mk)$. Since $m$ is bounded by the candidate budget, in large LLM layers we have $mk \ll d$, yielding substantial savings over full inversion while still retaining exactness (in the edge case $mk = d$, the cost reduces to full inversion).

**Conditioned scoring for sequential (batch) selection.** Greedy OBS removes one block at a time, recomputing scores after each update. To form a batch, we grow a set $J \subseteq K_i$ of *certified* blocks, blocks that greedy OBS would remove in this order, without updating the weights. For any $c \in K_i \setminus J$, we define the *conditioned block metric* via the Schur complement:

$$A'_c := G_{cc} - G_{cJ}G_{JJ}^{-1}G_{Jc} \in \mathbb{R}^{k \times k}. \tag{6}$$

The conditioned score is

$$\mathcal{E}'(c \mid J) = \|(A'_c)^{-1/2}W_c\|_F^2, \tag{7}$$

where $W_c$ are the weights for block $c$. The score $\mathcal{E}'(c \mid J)$ equals the OBS error that would be computed if we actually updated the weights after pruning $J$. Therefore, the certification procedure ranks candidates in exactly the same order as greedy OBS, ensuring that the certified set $J$ matches the greedy OBS sequence up to the stopping point (formal proof in Appendix A.1).

**Incremental Cholesky for fast certification.** Naively recomputing $G_{JJ}^{-1}$ for each certified set $J$ would cost $\mathcal{O}((|J|k)^3)$. Instead, we maintain the Cholesky factorization $G_{JJ} = L_{JJ}L_{JJ}^{\top}$ and update it incrementally. When adding a new block $c$ to $J$, the update proceeds in three steps: i) solve two triangular systems with $L_{JJ}$ to obtain $Y^{\top} = L_{JJ}^{-1}G_{Jc}$, ii) form the Schur complement $S_c = G_{cc} - Y^{\top}Y$, iii) compute the Cholesky factorization of $S_c$. Each candidate score can then be updated at $\tilde{\mathcal{O}}(k^2|J|) + \tilde{\mathcal{O}}(k^3)$, and the appending of a block has the same complexity. This incremental update is asymptotically cheaper in $|J|$ than either recomputing $G_{JJ}^{-1}$ from scratch ($\mathcal{O}((|J|k)^3)$) or applying it separately to each candidate block ($\mathcal{O}((|J|k)^2)$ per candidate).

**Maximal greedy-consistent prefix.** Let $K_i$ be the current candidate set and $J_t = \{c_1, \ldots, c_t\}$ the certified prefix after $t$ steps. Define the stopping index $T$ as

$$T = \min\Big\{ t \geq 1 \ \Big| \ \underset{c \in K_i \setminus J_t}{\arg\min}\, \mathcal{E}'(c \mid J_t) \ \neq \ \underset{c \in K_i \setminus J_t}{\arg\min}\, \mathcal{E}(c \mid W^{(t)}) \Big\}, \tag{8}$$

where $W^{(t)}$ is the weight matrix obtained by pruning $J_t$ via OBS. The certified set $J_T$ is therefore the largest prefix consistent with one-by-one greedy OBS; stopping here guarantees greedy equivalence for the entire batch (formal proof in Appendix A.2). This equivalence holds strictly under the local quadratic reconstruction objective utilized by OBS.

**Joint weight update.** Once $P = J$ is certified, we perform a single joint OBS update using Equation 3. This joint update produces exactly the same final weights as applying the corresponding sequence of single-block OBS updates in order (formal proof in Appendix A.3).

## 4.2 CANDIDATE BLOCK SELECTION

We begin candidate construction with a simple proxy: the average L2 norm of each block's weights (Li et al., 2017; Molchanov et al., 2017). Although this measure ignores second-order effects, Figure 1 shows that it correlates well with OBS, achieving Jaccard overlaps of 0.7–0.85 and expanding the pool by around $1.5\times$ suffices to cover all OBS top-$K$ blocks. This suggests that L2 magnitudes, while imperfect, are adequate for inexpensive initial screening. The total number of candidates is then set by a global hyperparameter and distributed across layers according to their estimated sensitivities (detailed in Section 4.2.1) and intra-layer score variability (see Section 4.2.2). The resulting compact but representative sets are then passed to Hybrid-OBS (Section 4.1) for accurate second-order scoring and *batched* pruning.

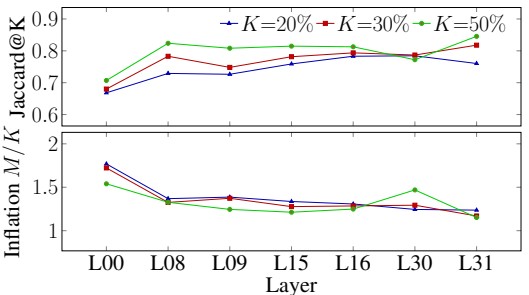

Figure 1: Alignment of L2 and OBS rankings across layers. *Top:* Jaccard@K similarity at pruning ratios 20%, 30%, and 50%. *Bottom:* Inflation $M/K$, the factor by which L2 candidate pools must expand to fully cover OBS top-$K$. Higher Jaccard and lower inflation indicate stronger agreement.

### 4.2.1 COMPUTING TRANSFORMER LAYER SENSITIVITY VIA HESSIAN APPROXIMATION

Initial importance scores ignore how pruning errors in one layer can propagate and amplify in later layers, making it crucial to account for the sensitivity of the transformer layer when selecting candidate blocks. To account for this, we estimate layer sensitivity using an efficient second-order approximation instead of the full Hessian, which is infeasible for large LLMs.

For consecutive Transformer layers $\ell$ and $\ell + 1$, let $h^{(\ell)}$ be the output of transformer layer $\ell$ and $f^{(\ell)}(\cdot)$ denotes the function of the next transformer layer. For a perturbation $\Delta h^{(\ell)}$, the change in the output of transformer layer $\ell + 1$ is quantified as

$$g(h^{(\ell)}) = \left\| f^{(\ell+1)}(h^{(\ell)} + \Delta h^{(\ell)}) - f^{(\ell+1)}(h^{(\ell)}) \right\|_2^2 \tag{9}$$

A second-order Taylor expansion for small $\Delta h^{(\ell)}$ gives $g(h^{(\ell)}) \approx \frac{1}{2}(\Delta h^{(\ell)})^\top H^{(\ell+1)} \Delta h^{(\ell)}$, where $H^{(\ell+1)} \approx (X^{(\ell+1)})^\top X^{(\ell+1)} + \lambda I$ is the Hessian approximation, $X^{(\ell+1)}$ is the input to transformer layer $\ell + 1$, and $\lambda$ ensures stability. The local sensitivity between layers $\ell$ and $\ell + 1$ is given by the Hessian trace $\text{Tr}(\cdot)$:

$$S^{(\ell)\to(\ell+1)} = \text{Tr}(H^{(\ell+1)}) \tag{10}$$

To capture global effects without the full Hessian, we recursively propagate sensitivities backward from the final transformer layer, with $\beta$ controlling the influence of downstream layers on earlier ones:

$$S^{(\ell)} = S^{(\ell)\to(\ell+1)} + \beta S^{(\ell+1)} \tag{11}$$

This recursion efficiently captures how pruning perturbations propagate across the model, enabling more informed candidate block selection.

### 4.2.2 SELECTING CANDIDATE BLOCKS AND DETERMINING BLOCKS TO PRUNE

To allocate candidates fairly across the model, we evaluate each module (MHA or FFN) within layer $\ell$ separately. For a given module $\tau \in \{\text{MHA}, \text{FFN}\}$ of transformer layer $\ell$, we define a module-level metric that incorporates both the variability in block importances within the module and the sensitivity of its parent layer:

$$R^{(\ell,\tau)} = \frac{\text{CV}^{(\ell,\tau)}}{S^{(\ell)} + \epsilon}, \qquad \text{with} \quad \text{CV}^{(\ell,\tau)} = \frac{\sigma^{(\ell,\tau)}}{\mu^{(\ell,\tau)}}, \tag{12}$$

Here, $\mu^{(\ell,\tau)}$ and $\sigma^{(\ell,\tau)}$ are the mean and standard deviation of block importance scores, and $\epsilon$ ensures stability. The number of candidate blocks per module is

$$K^{(\ell,\tau)} = \min(C \cdot K, \, N^{(\ell,\tau)}) \cdot \frac{R^{(\ell,\tau)}}{\sum_{\ell',\tau'} R^{(\ell',\tau')}}, \tag{13}$$

This design favours modules where pruning is less risky (low $S^\ell$) and where block importances vary widely (high $\text{CV}^{\ell,\tau}$), ensuring that the candidate pool adapts to both inter-layer sensitivity and intra-layer variability. We then compute OBS scores for all candidates (Equation 2), rescale them by the shared layer sensitivity $S^\ell$ to penalize fragile layers, and globally rank blocks to figure out how many blocks should be pruned in each module $\tau$. We then compute OBS scores for all candidates (Equation 2), rescale them by the shared layer sensitivity $S^\ell$ to penalize fragile layers, and globally rank blocks. Let $r_{\ell,m}^{(\tau)}$ denote the number of blocks to prune from module $\tau$ in layer $\ell$, such that $\sum_{\ell,\tau} r^{(\ell,\tau)} = K$ with $r^{(\ell,\tau)} \leq K^{\ell,\tau}$. We repeat the certification and pruning steps in Section 4.1 until exactly $r_{\ell,\tau}^{(\cdot)}$ blocks have been removed from module $\tau$ of layer $\ell$.

### 4.3 LATENCY ESTIMATION

We employ a learned latency model to guide pruning toward a target runtime without repeated on-device profiling. For each Transformer module (MHA or FFN) in layer $\ell$, we first measure its execution time on the target hardware under different pruning configurations and record the features

$$\mathbf{x}^{(\ell)} = [S, \, d_{\text{model}}, \, h^{(\ell)}, \, d_{\text{ffn}}^{(\ell)}], \tag{14}$$

where $S$ is the sequence length, $d_{\text{model}}$ denotes the hidden dimension, $h^{(\ell)}$ shows the number of active heads, and $d_{\text{ffn}}^{(\ell)}$ is the FFN intermediate dimension after pruning. We then train separate regressors $f_{\text{MHA}}$ and $f_{\text{FFN}}$ using linear regression to predict module-level latencies.

For a pruned model $\mathcal{A}$, the block-level predictions are aggregated using a lightweight linear model:

$$\hat{L}_{\text{tot}}(\mathcal{A}) = \alpha_0 + \sum_{b=1}^{B} \alpha_b \, f_{\tau(b)}(\mathbf{x}_b) \tag{15}$$

where $\tau(b) \in \{\text{MHA}, \text{FFN}\}$ indicates the block type and the coefficients $\alpha_b$ are also fitted via linear regression on end-to-end latency samples from pruned models. This two-stage design corrects for non-additive effects such as memory allocation and kernel fusion, while also capturing variation across sequence length and width, making the estimator tailored to Transformer architectures.

## 5 EXPERIMENTS

**Setup.** We implement *HAP-E* in PyTorch (Paszke et al., 2019) with HuggingFace Transformers (Wolf et al., 2019). Following SlimGPT (Ling et al., 2024), we calibrate on 256 sample with sequence length 2048 from C4 dataset. In all experiments, Pruning is strictly post-training without any fine-tuning. All pruning experiments are conducted on a single NVIDIA A100 (80GB). For edge deployment, models are compiled with ExecuTorch and benchmarked on Jetson Xavier NX and HiKey970 CPUs at batch size 1, averaged over 10 runs with 2 warm-ups. Detailed hyper-parameters are provided in Appendix K for reproducibility.

**Models and Datasets.** We evaluate models from the LLaMA family (Touvron et al., 2023), OPT family (Zhang et al., 2022), and TinyLLaMA. Compressed models are assessed using `lm-eval-harness` (Gao et al., 2024) on seven zero-shot benchmarks: ARC-c, ARC-e (Clark et al., 2018), WinoGrande (Sakaguchi et al., 2021), BoolQ (Clark et al., 2019), HellaSwag (Zellers et al., 2019), OpenBookQA (Mihaylov et al., 2018), and PIQA (Bisk et al., 2020). We report average accuracy (%) across tasks, consistent with prior work. All results are averaged over four different random seeds for pruning and calibration sample selection.

**Baselines.** We compare against six state-of-the-art compression methods: FLAP (An et al., 2024), SliceGPT (Ashkboos et al., 2024), LLM-Pruner (Ma et al., 2023), SlimGPT (Ling et al., 2024), SoBP (Wei et al., 2024), and ASVD (Yuan et al., 2023). The set includes both pruning-based and decomposition-based methods to cover the dominant strategies for reducing LLM complexity.

### 5.1 RESULTS ON LLAMA AND OPT MODELS

Tables 1(a) and (b) summarize pruning results on LLaMA-7B/13B/30B and OPT-6.7B/13B/30B, respectively. On the LLaMA family, our method consistently outperforms post-training baselines across pruning ratios and model scales. At moderate pruning (20–30%), we achieve around 1.5% higher accuracy than SoBP and SlimGPT. For instance, on LLaMA-13B at 20% pruning, our method reaches 67.8%, compared to 66.9% (SoBP) and 66.4% (SlimGPT). At more aggressive pruning (40–50%), the gap widens: on LLaMA-30B at 50% pruning we obtain 68.0%, roughly 2.5% higher than SlimGPT and nearly 8% higher than LLM-Pruner. These results demonstrate that adaptive block allocation with OBS reconstruction provides robustness under severe compression.

For OPT models, dense baselines start at lower accuracies, and the margins across methods are smaller. Nonetheless, our method consistently preserves accuracy. At 10–20% pruning, we nearly match dense performance (e.g., OPT-13B: 59.0% vs. 59.2% dense). At 30% pruning, our approach still maintains the best accuracy among all methods, showing up to 3–4% improvements over ASVD. These consistent gains highlight the effectiveness of our adaptive candidate allocation strategy in maintaining model quality even when pruning larger OPT variants.

**Extended Evaluations on Modern Families.** To demonstrate the robustness and scalability of *HAP-E*, we provide extensive additional results in the Appendices. **Appendix E** details LLaMA-2 evaluations against decomposition baselines (MoDeGPT, SVD-LLM v2) and reasoning benchmarks, while **Appendix G** validates compatibility with recovery fine-tuning. Furthermore, **Appendix F** validates performance on state-of-the-art architectures, including LLaMA-3.1-8B and Qwen-2.5-14B.

Table 1: Average accuracy (%) on commonsense reasoning tasks under different pruning rates. (a) LLaMA family. (b) OPT family. Per-task results are in Appendix I and Appendix J.

(a) LLaMA-7B/13B/30B

| Model | | LLaMA-7B | | LLaMA-13B | | LLaMA-30B | |
|---|---|---|---|---|---|---|---|
| Prune% | Method | #Params | Avg↑ | #Params | Avg↑ | #Params | Avg↑ |
| 0% | Dense | 6.7B | 66.05 | 13.0B | 68.21 | 32.5B | 71.92 |
| 20% | SliceGPT | 6.1B | 56.16 | 11.8B | 60.66 | 29.5B | 64.45 |
| | ASVD | 5.4B | 61.55 | 10.4B | 65.29 | 26.1B | 70.22 |
| | SoBP | 5.4B | 62.19 | 10.4B | 66.96 | 26.1B | 70.87 |
| | LLM-Pruner | 5.4B | 61.50 | 10.4B | 65.68 | 26.0B | 69.99 |
| | SlimGPT | 5.4B | 63.81 | 10.4B | 66.37 | 26.0B | 71.13 |
| | *HAP-E* | 5.4B | **65.01** | 10.4B | **67.83** | 26.0B | **71.88** |
| 30% | SliceGPT | 5.3B | 46.90 | 10.2B | 54.26 | 25.5B | 58.05 |
| | ASVD | 4.8B | 45.55 | 9.2B | 57.47 | 22.9B | 61.88 |
| | SoBP | 4.8B | 59.61 | 9.2B | 64.50 | 22.9B | 69.62 |
| | *HAP-E* | 4.8B | **61.21** | 9.2B | **65.92** | 22.9B | **71.02** |
| 40% | SliceGPT | 4.5B | 39.64 | 8.6B | 47.00 | 21.5B | 48.90 |
| | ASVD | 4.1B | 36.79 | 7.9B | 40.13 | 19.7B | 49.79 |
| | SoBP | 4.1B | 56.10 | 7.9B | 60.34 | 19.7B | 67.20 |
| | *HAP-E* | 4.1B | **58.40** | 7.9B | **62.84** | 19.7B | **69.50** |
| 50% | LLM-Pruner | 3.4B | 48.35 | 6.5B | 53.22 | 16.3B | 59.47 |
| | SlimGPT | 3.4B | 54.26 | 6.5B | 59.89 | 16.3B | 65.59 |
| | *HAP-E* | 3.4B | **56.66** | 6.5B | **61.79** | 16.3B | **67.99** |

(b) OPT-6.7B/13B/30B

| Model | | OPT-6.7B | | OPT-13B | | OPT-30B | |
|---|---|---|---|---|---|---|---|
| Prune% | Method | #Params | Avg↑ | #Params | Avg↑ | #Params | Avg↑ |
| 0% | Dense | 6.7B | 58.16 | 13.0B | 59.15 | 30.0B | 61.85 |
| 10% | FLAP | 6.0B | 57.31 | 11.6B | 58.10 | 27.0B | 59.26 |
| | SliceGPT | 7.1B | 57.07 | 13.5B | 59.18 | 31.3B | 61.61 |
| | ASVD | 6.0B | 55.18 | 11.6B | 56.32 | 27.0B | 59.11 |
| | *HAP-E* | 6.0B | **57.96** | 11.6B | 59.11 | 27.0B | **62.19** |
| 20% | FLAP | 5.4B | 54.72 | 10.3B | 55.36 | 24.0B | 56.52 |
| | SliceGPT | 6.2B | 55.50 | 11.9B | 57.84 | 27.5B | 60.86 |
| | ASVD | 5.4B | 45.11 | 10.3B | 39.20 | 24.0B | 49.48 |
| | *HAP-E* | 5.4B | **57.83** | 10.3B | **59.02** | 24.0B | **61.82** |
| 30% | FLAP | 4.7B | 52.77 | 9.1B | 50.81 | 21.1B | 52.61 |
| | SliceGPT | 5.4B | 54.16 | 10.3B | 55.92 | 23.8B | 59.49 |
| | ASVD | 4.7B | 37.86 | 9.1B | 36.85 | 21.1B | 41.12 |
| | *HAP-E* | 4.7B | **57.60** | 9.1B | **58.46** | 21.1B | **61.29** |

## 5.2 LATENCY MODEL VERIFICATION

We train a *whole-model* latency predictor on 1500 pruned configurations and evaluate on a 300-sample held-out set. Each configuration varies the sequence length $S \in \{128, 256, 384, 512, 1024\}$ and structured sparsity.

To generate module-level features, we record block runtimes on the target hardware: for MHA, we measure execution time with $0, \ldots, (N_{\text{heads}} - 1)$ heads pruned; for FFN, we measure runtime as the intermediate dimension shrinks by factors of $0.9^i$ for $i = 0, \ldots, 42$ (10% relative steps up to 99% sparsity), following prior work (Kurtić et al., 2023). All CPU experiments use INT8 post-training quantization. Predictor evaluations on Jetson Xavier NX and HiKey970 use *LLaMA-3.2-1B*, while A100 experiments use *LLaMA-7B* at batch size 16.

We compare against a *lookup-table* baseline that estimates whole-model latency by summing layer-wise measurements (Kurtić et al., 2023). This approximation ignores inter-layer effects (e.g., fusion, scheduling), leading to weaker prediction fidelity and reduced pruning accuracy. Figure 2 shows the *target-attainment* plot, where the $y$-axis is the ratio of measured to target latency (ideal $\approx 1.0$). Table 2 reports error metrics on the test set. Our predictor consistently achieves $R^2 \approx 0.97$ with attainment ratios close to 1.0, while the lookup baseline diverges ($R^2 < 0.91$, up to 8% off target).

Table 2: Latency predictor accuracy on 200 test samples.

| Device / Method | MSE (ms$^2$) | RMSE (ms) | $R^2$ |
|---|---|---|---|
| Jetson NX (*HAP-E*) | 190 | 13.8 | 0.972 |
| Jetson NX (lookup) | 510 | 22.6 | 0.889 |
| HiKey970 (*HAP-E*) | 270 | 16.4 | 0.968 |
| HiKey970 (lookup) | 640 | 25.3 | 0.884 |
| A100-b16 (*HAP-E*) | 310 | 17.6 | 0.965 |
| A100-b16 (lookup) | 780 | 27.9 | 0.902 |

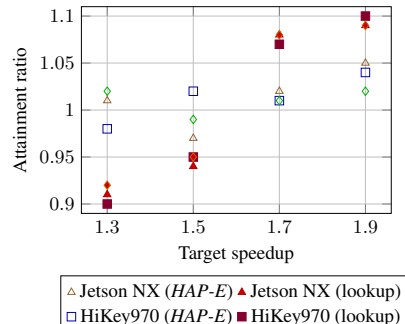

Figure 2: Target-attainment chart for end-to-end latency. Ratios near 1.0 indicate predictor-guided pruning meets runtime targets.

## 5.3 HARDWARE-AWARE LATENCY–ACCURACY EVALUATION

To validate the practical efficiency of our pruning strategy, we evaluate accuracy–latency trade-offs across two distinct hardware regimes: low-power edge CPUs (Jetson Xavier NX, HiKey970) using ExecuTorch, and high-performance GPUs (NVIDIA A100) using PyTorch.

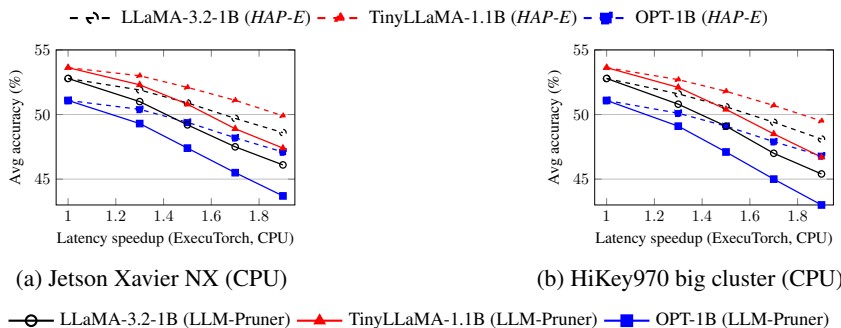

(a) Jetson Xavier NX (CPU)    (b) HiKey970 big cluster (CPU)

Figure 3: Accuracy–latency trade-offs with ExecuTorch on (a) Jetson Xavier NX and (b) HiKey970 CPUs. Dashed = Ours, solid = LLM-Pruner.

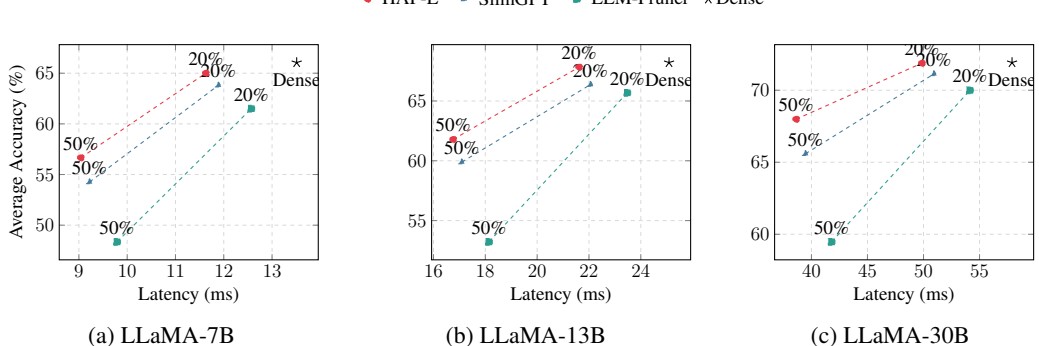

(a) LLaMA-7B    (b) LLaMA-13B    (c) LLaMA-30B

Figure 4: Accuracy vs. Latency Comparison across LLaMA-7B, 13B, and 30B models at 20% and 50% pruning ratios.. HAP-E = red circles, SlimGPT = blue triangles, and LLM-Pruner = green squares.

**Edge CPU Deployment.** We first benchmarked three small-scale LLMs—LLaMA-3.2-1B, TinyLLaMA-1.1B, and OPT-1B—on the CPUs of the Jetson Xavier NX and the HiKey970 (big cluster). We targeted aggressive latency speedups of $1.3\times$, $1.5\times$, $1.7\times$, and $1.9\times$, comparing *HAP-E* directly against LLM-Pruner. As illustrated in Figure 3, our approach consistently yields higher accuracy at every speedup level, with particularly significant gains on the weaker HiKey970 processor. notably, at $1.9\times$ speedup, *HAP-E* limits the accuracy degradation to approximately 4 percentage points, whereas LLM-Pruner suffers a much steeper drop across all models. These results demonstrate that the adaptive second-order sensitivity modeled in *HAP-E* effectively preserves critical structures under strict end-to-end latency constraints, a crucial advantage for edge environments where compute budgets are severely limited.

**Scalability on GPUs.** To assess scalability beyond edge devices, we conducted a comprehensive latency evaluation on NVIDIA A100 GPUs. We measured the prefill latency (batch size 1, sequence length 2048) for LLaMA-7B, 13B, and 30B models at 20% and 50% structured sparsity, comparing against state-of-the-art baselines SlimGPT and LLM-Pruner. This setting reflects real-world inference workloads where prefill latency is often a bottleneck.

As shown in Figure 4, *HAP-E* consistently pushes the Pareto frontier of accuracy versus latency across all model sizes. Crucially, the performance margin of *HAP-E* over SlimGPT and LLM-Pruner is maintained—and often widened—as model size increases to 30B parameters. This confirms that our recursive Hessian-trace sensitivity and greedy-consistent batch updates remain robust even as the curvature landscape becomes more complex in deeper networks. Furthermore, under identical sparsity levels, *HAP-E* achieves lower wall-clock latency than the baselines while preserving higher task accuracy. Overall, these experiments establish that the efficiency gains observed on edge CPUs successfully translate to large-scale GPU deployments, validating *HAP-E* as a hardware-agnostic solution that adapts robustly to diverse compute regimes.

Table 3: Ablations on LLaMA models. Each cell = **%Acc**(±std)/**Time**(min)/**Mem**(GB).

| Variant | LLaMA-7B | | LLaMA-13B | | LLaMA-30B | |
|---|---|---|---|---|---|---|
| | 1.3× | 1.9× | 1.3× | 1.9× | 1.3× | 1.9× |
| *HAP-E* (ours) | **64.9**(0.31)/**9.8**/**4.5** | **58.2**(0.44)/**22.0**/**4.5** | **67.7**(0.29)/**15.3**/**7.1** | **62.6**(0.31)/**34.4**/**7.1** | **71.8**(0.24)/**25.7**/**9.2** | **69.3**(0.37)/**58.0**/**9.2** |
| w/o cross-layer adapt. | 64.2(0.47)/8.4/4.4 | 55.9(0.62)/18.7/4.4 | 66.1(0.23)/13.0/7.0 | 60.0(0.41)/29.2/7.0 | 70.3(0.27)/22.0/9.0 | 67.7(0.39)/50.0/9.0 |
| w/o greedy batch | 64.8(0.58)/31.6/4.5 | 57.0(0.61)/74.5/4.5 | 66.7(0.52)/49.4/7.1 | 61.4(0.59)/116.3/7.1 | 71.0(0.35)/55.7/9.2 | 68.1(0.37)/130.2/9.2 |
| w/o latency predictor | 64.9(0.28)/12.3/4.5 | 58.1(0.51)/27.5/4.5 | 67.6(0.36)/19.8/7.1 | 62.5(0.42)/43.0/7.1 | 71.7(0.21)/32.0/9.2 | 69.2(0.33)/72.0/9.2 |
| vanilla OBS | 63.8(0.62)/43.0/8.0 | 56.4(0.69)/101.0/8.0 | 65.9(0.56)/67.0/12.3 | 60.6(0.51)/158.0/12.3 | 70.5(0.31)/79.6/21.1 | 67.6(0.38)/187.4/21.1 |

## 5.4 Ablation: Runtime, Memory, and Accuracy of OBS Variants on GPU

We evaluate *HAP-E* and controlled variants on an NVIDIA A100 (80GB), where all LLaMA-7B/13B/30B models can be executed reliably. Variants include: (i) **HAP-E (ours)**, with all components enabled; (ii) **w/o cross-layer adaptivity**, which fixes layer budgets statically at the start; (iii) **w/o greedy batch**, reverting to one-by-one OBS updates; (iv) **w/o latency predictor**, which requires multiple pruning runs to meet a speedup target; and (v) **vanilla OBS**, a layer-by-layer baseline with $mk = d$ and no candidate screening. We target $1.3\times$ and $1.9\times$ end-to-end GPU latency reductions relative to dense baselines, reporting average task accuracy, pruning runtime (including calibration), and peak GPU memory during pruning. Table 3 shows the obtained results. As can be seen, *HAP-E* consistently preserves accuracy while keeping runtime and memory practical on GPU. Cross-layer adaptivity is most impactful under aggressive compression: at $1.9\times$ speedup on LLaMA-7B, static layer budgets reduce accuracy from 58.2% to 55.9%, showing that adaptive budget reallocation is critical to avoid accuracy degradation. Greedy-consistent batching is the main efficiency driver. For instance, on LLaMA-30B at $1.9\times$, *HAP-E* prunes in 58 minutes versus 130 minutes without batching, a $>2\times$ runtime reduction at equal accuracy. The latency predictor eliminates wasted sweeps: without it, LLaMA-13B takes 43 minutes at $1.9\times$ (three redundant pruning runs), compared to 34 minutes with predictor guidance. Finally, vanilla OBS underscores the scalability challenge: at $1.9\times$ on LLaMA-30B, it demands 187 minutes and 21 GB memory—over $3\times$ slower and $>2\times$ the footprint of *HAP-E*—despite offering no accuracy benefit. Together, these results confirm that *HAP-E* is the only configuration that achieves OBS-level accuracy while scaling efficiently on modern GPUs. We further ablate prune fraction $K$ and candidate pool ratio $M/K$ (Appendix B), calibration budget (Appendix C), and sensitivity coefficient $\beta$ (Appendix D).

## 6 Conclusion

We introduced *HAP-E*, a scalable, Hessian-aware structured pruning framework that makes OBS-style pruning tractable for large language models. By combining global screening with selective second-order refinement, cross-layer sensitivity analysis, and greedy-consistent batch pruning, our method achieves the same theoretical guarantees as greedy OBS while dramatically reducing computational overhead and numerical drift. The integration of a lightweight latency predictor further ensures that pruning decisions directly meet hardware-specific runtime constraints. Extensive experiments on the LLaMA and OPT families demonstrate that *HAP-E* consistently outperforms state-of-the-art pruning baselines across sparsity levels. On commonsense reasoning benchmarks, it improves average accuracy by up to 2–3% over SlimGPT and SoBP at comparable pruning ratios, while retaining robustness under aggressive 40–50% block removal. The latency predictor achieves $R^2 \approx 0.97$ against measured runtimes, allowing the pruned model to meet target latencies in a single pass without iterative sweeps. Hardware benchmarks confirm that our approach sustains accuracy under strict latency budgets, while ablation results highlight the efficiency benefits of cross-layer adaptivity and greedy batching.

Although our study prioritizes training-free post-training pruning, Appendix G demonstrates that *HAP-E* models are inherently compatible with recovery fine-tuning. A minimal LoRA tuning step yields substantial accuracy gains (+1.8%), confirming that our structured pruning preserves a high-quality feature space suitable for further optimization. Extending *HAP-E* to training-aware or continual-learning settings remains a promising direction for future work. Moreover, we consider pruning in isolation, whereas extending the framework to hybrid pruning–quantization pipelines could further enhance efficiency for deployment.

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

# A  Proofs of Lemmas in Section 4

## A.1  Lemma 1 (Greedy-equivalence of certified batch)

**Lemma 1** (Greedy-equivalence of certified batch). *Let $(c_1, \ldots, c_t)$ be the blocks selected by the certification procedure, where at step $\tau$,*

$$c_\tau = \arg \min_{c \notin J_{\tau-1}} E'(c \mid J_{\tau-1}), \quad with \ J_{\tau-1} = \{c_1, \ldots, c_{\tau-1}\} \tag{16}$$

*Then this sequence matches the first $t$ selections of standard greedy OBS for any $t \leq T$, up to the certification stopping point.*

*Proof.* In greedy OBS, the block chosen at step $\tau$ after pruning $J_{\tau-1}$ is

$$c_\tau^{\text{greedy}} = \arg \min_{c \notin J_{\tau-1}} E\left(c \,\middle|\, W^{(\tau-1)}\right) \tag{17}$$

where $W^{(\tau-1)}$ are the weights after applying the OBS update for $J_{\tau-1}$. Let $G = H^{-1}$ be the inverse Hessian prior to pruning. Eliminating $J_{\tau-1}$ updates the effective inverse sub-block for any remaining $c$ to the Schur complement

$$A'_c = G_{cc} - G_{cJ} G_{JJ}^{-1} G_{Jc} \tag{18}$$

and the OBS error for $c$ *after* pruning $J_{\tau-1}$ becomes

$$E\left(c \,\middle|\, W^{(\tau-1)}\right) = \left\|(A'_c)^{-1/2} W_c\right\|_F^2 = E'(c \mid J_{\tau-1}) \tag{19}$$

Thus, at every step the certification score $E'(c \mid J_{\tau-1})$ equals the greedy-OBS score computed after actually pruning $J_{\tau-1}$. Therefore the $\arg \min$ choices coincide step-by-step, and by induction the sequences match up to the certification horizon $T$. $\qquad\square$

## A.2  Lemma 2 (Batch = maximal greedy-consistent prefix)

**Lemma 2** (Batch = maximal greedy-consistent prefix). *If, during certification, the identity of the next best block changes after appending a candidate, then the current $J$ is the largest prefix that matches the greedy OBS sequence. Stopping here preserves greedy equivalence for the entire certified batch.*

*Proof.* Suppose after certifying $J_t$ the certification rule selects

$$\hat{c} = \arg \min_{c \notin J_t} E'(c \mid J_t) \tag{20}$$

while greedy OBS, after actually pruning $J_t$, selects

$$c^\star = \arg \min_{c \notin J_t} E\left(c \,\middle|\, W^{(t)}\right) \tag{21}$$

If $\hat{c} \neq c^\star$, a divergence occurs at $t+1$. From Lemma A.1, for any prefix that matches greedy so far, $E'(c \mid J_t) \equiv E\left(c \mid W^{(t)}\right)$; hence the first possible mismatch is exactly at $t+1$. Therefore $J_t$ is the *maximal* prefix consistent with greedy OBS. Halting certification at this point guarantees that the certified batch equals the greedy sequence prefix. $\qquad\square$

## A.3  Lemma 3 (Batch update equivalence)

**Lemma 3** (Batch update equivalence). *Applying a single joint OBS update for $P = J$ yields the same final weights as applying $t$ one-by-one OBS updates sequentially for $(c_1, \ldots, c_t)$*

*Proof.* Let $P = \{c_1, \ldots, c_t\}$ and let $R$ index the surviving blocks. The joint OBS update that zeroes $W_P$ while minimizing the quadratic loss with Hessian $H$ is

$$\Delta W_R = - H_{RP} H_{PP}^{-1} W_P \tag{22}$$

This is precisely the block Gaussian-elimination solution obtained by eliminating $P$ in one step. On the other hand, sequential greedy OBS eliminates the same set $P$ via a sequence of rank-$k$ Schur complements. Block Gaussian elimination is order-invariant with respect to the eliminated set: eliminating the union $P$ in any order (or jointly) produces the same reduced system over $R$ and the same solution for $\Delta W_R$. Hence the final weights after the joint update equal those after $t$ sequential single-block OBS updates. $\qquad\square$

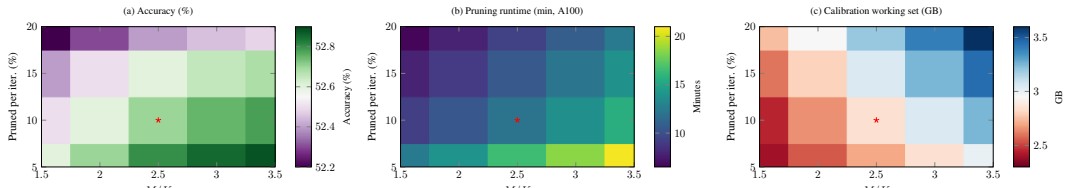

Figure 5: Ablation at a fixed $1.3\times$ latency target on Jetson Xavier NX (CPU) for LLaMA-3.2-1B. Rows vary prune-per-iteration $K$; columns vary candidate pool ratio $M/K$. (a) Final *absolute* accuracy (%) on lm-eval (avg). (b) *Total pruning runtime* (minutes) on A100 (80GB). (c) Peak calibration working set (GB). The starred point ($K$=10%, $M/K$=2.5) achieves near-dense accuracy ($\approx$52.7%), $\sim$12 minutes runtime, and moderate memory while meeting the $1.3\times$ budget.

## B  ABLATION ON CANDIDATE POOL RATIO AND PRUNE FRACTION

We study how the per-iteration prune fraction $K$ and the candidate pool ratio $M/K$ shape outcomes when compressing *LLaMA-3.2-1B* to a fixed *$1.3\times$ latency speedup* on *Jetson Xavier NX (CPU)* with batch size 1 (ExecuTorch runtime). Figure 5 reports three metrics: (a) the final average accuracy (%) across seven LM-Eval benchmarks after deployment; (b) the total pruning runtime on an NVIDIA A100 (80GB); and (c) the peak calibration working set (GB) required during pruning.

The trends are consistent with structured OBS pruning. **Accuracy** improves (gently) as updates become less aggressive (smaller $K$) and as the candidate pool widens (larger $M/K$), reflecting better coverage of high-gain removals and fewer destabilizing steps. **Runtime** grows when $K$ is smaller (more iterations to reach the same global budget) and when $M/K$ is larger (more candidates to score each step). **Memory** increases smoothly with both $K$ and $M/K$, since larger batches and wider pools expand the active calibration set and per-iteration working set.

Overall, the configuration $K$=10%, $M/K$=2.5$\times$ (starred) offers the best balance: it reaches the hardware-constrained $1.3\times$ speedup with near-dense accuracy ($\approx$52.7%, within 0.1–0.2 of the dense Jetson baseline of 52.79%), completes pruning in $\sim$12 minutes on A100, and maintains a moderate memory footprint. We adopt this setting throughout the main experiments, and we observe analogous behavior on larger models and alternate targets.

## C  ABLATION ON CALIBRATION COUNT

We further analyze how calibration budget influences pruning outcomes when targeting a fixed *$1.3\times$ end-to-end latency* on *Jetson Xavier NX (CPU)* (batch=1, ExecuTorch). We vary the number of calibration samples (64, 128, 256, 512), fixing the per-iteration prune fraction at $K$=10% and candidate ratio at $M/K$=2.5$\times$. We report: (a) final accuracy across seven lm-eval tasks, (b) pruning runtime on an NVIDIA A100 (80GB), and (c) peak calibration working set during pruning.

Accuracy rises with more calibration but saturates quickly: 64 samples trail the dense baseline (52.79%) by about one point, 128 nearly closes the gap, and 256 reaches 52.6–52.7%, effectively matching dense. Going to 512 yields only marginal gains ($\sim$0.1 points), well within variance. In contrast, runtime and memory scale nearly linearly with calibration size: from 7 minutes / 3.0 GB at 64 samples to 21 minutes / 6.2 GB at 512. Overall, 256 samples strike the best trade-off, preserving near-dense accuracy while keeping pruning practical on a single GPU.

## D  ABLATION ON SENSITIVITY PROPAGATION COEFFICIENT $\beta$

In Section 4.2.1, we introduced a coefficient $\beta \in [0, 1]$ to control how strongly downstream sensitivities influence earlier layers during recursive propagation:

$$S^{(\ell)} = S^{(\ell)\to(\ell+1)} + \beta S^{(\ell+1)}.$$

When $\beta = 0$, sensitivities are purely local, i.e., layer $\ell$ only accounts for its immediate perturbation effect $S^{(\ell)\to(\ell+1)}$. When $\beta = 1$, full downstream influence is considered, effectively chaining sen-

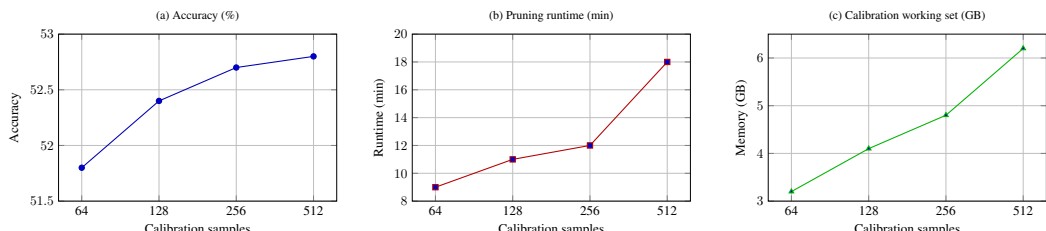

Figure 6: Ablation over calibration sample count when pruning LLaMA-3.2-1B to $1.3\times$ latency on Jetson Xavier NX (CPU). (a) Accuracy compared to the dense baseline (dashed). (b) Pruning runtime on A100 (80GB). (c) Peak calibration working set during pruning.

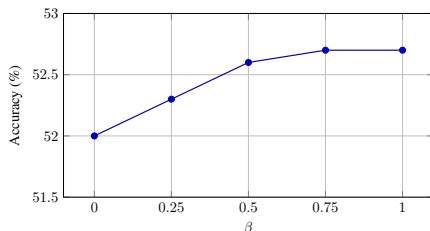

Figure 7: Effect of $\beta$ on pruning LLaMA-3.2-1B to $1.3\times$ latency on Jetson NX (CPU). Accuracy improves as $\beta$ increases up to $0.5$–$0.75$, reflecting the benefit of incorporating downstream sensitivities. Runtime and memory remain essentially unchanged.

sitivities across the network. Intermediate values interpolate between these two extremes, balancing local stability with global robustness.

We prune *LLaMA-3.2-1B* to a fixed *$1.3\times$ latency speedup* on *Jetson Xavier NX (CPU)* and vary $\beta \in \{0.0, 0.25, 0.5, 0.75, 1.0\}$. Figure 7 reports average accuracy across seven common reasoning benchamrks. As shown in the paper, $\beta = 0.75$ offers the best trade-off. Purely local sensitivities ($\beta = 0$) underestimate error propagation and reduce accuracy, while $\beta = 1.0$ yields no further gains.

# E    EXTENDED RESULTS ON LLAMA-2 FAMILY

In this section, we provide extended comparisons against recent decomposition-based methods (MoDeGPT, SVD-LLM v2) and evaluate robustness on complex reasoning tasks.

## E.1    COMPARISON WITH DECOMPOSITION BASELINES

We benchmarked *HAP-E* against strong structured pruning and decomposition baselines on LLaMA-2-7B and LLaMA-2-13B. As shown in Table 4 and Table 5, *HAP-E* achieves superior accuracy across diverse zero-shot tasks, particularly at higher compression ratios (30%).

## E.2    COMPLEX REASONING AND GENERATION

To demonstrate robustness beyond standard multiple-choice tasks, we evaluated MMLU (5-shot, grouped by domain), GSM8K (Math), and WikiText-2 Perplexity (Generation). As shown in Table 6, *HAP-E* significantly outperforms baselines, achieving the highest MMLU average and the lowest perplexity.

# F    SCALABILITY TO MODERN ARCHITECTURES

To validate the generalizability of *HAP-E* to state-of-the-art architectures, we conducted experiments on LLaMA-3.1-8B and Qwen-2.5-14B Instruct. We compare against DarwinLM (Tang et al., 2025),

Table 4: LLaMA-2-7B Results. Comparison against SoBP, MoDeGPT, SlimGPT, and SVD-LLM v2 at 20% and 30% pruning ratios.

| Pruning | Method | BoolQ | PIQA | HellaS. | WinoG. | ARC-e | ARC-c | OBQA |
|---------|--------|-------|------|---------|--------|-------|-------|------|
| 0% | Dense | 77.71 | 79.05 | 76.00 | 68.98 | 74.58 | 46.33 | 44.20 |
| 20% | SoBP | 71.19 | 73.50 | 67.27 | 66.22 | 59.81 | 37.63 | 38.40 |
| | MoDeGPT | – | 74.05 | 69.05 | 68.03 | 69.07 | 42.06 | – |
| | SlimGPT | 73.43 | 77.58 | 72.62 | 68.82 | 69.99 | 42.32 | 42.00 |
| | SVD-LLM v2 | 61.42 | 72.89 | 63.55 | 66.77 | 58.12 | 38.76 | 40.87 |
| | *HAP-E* (Ours) | **75.24** | **78.61** | **74.29** | **69.77** | **71.86** | **44.03** | **43.68** |
| 30% | SoBP | 71.19 | 73.50 | 67.27 | 66.22 | 59.81 | 37.63 | 38.40 |
| | MoDeGPT | – | 70.40 | 63.26 | 67.32 | 63.26 | 38.73 | – |
| | SVD-LLM v2 | 58.62 | 70.45 | 61.18 | 64.23 | 54.97 | 36.41 | 37.89 |
| | *HAP-E* (Ours) | **71.82** | **76.73** | **70.68** | **68.04** | **68.47** | **41.98** | **43.59** |

Table 5: LLaMA-2-13B Results. Comparison at 20% and 30% pruning ratios.

| Pruning | Method | ARC-c | ARC-e | BoolQ | HellaS. | OBQA | PIQA | WinoG. |
|---------|--------|-------|-------|-------|---------|------|------|--------|
| 0% | Dense | 49.23 | 77.48 | 80.58 | 79.37 | 45.20 | 80.52 | 72.30 |
| 20% | MoDeGPT | 46.16 | 74.07 | – | 68.96 | – | 74.53 | 70.32 |
| | SVD-LLM v2 | 44.15 | 71.05 | 70.35 | 65.75 | 43.95 | 77.10 | 71.00 |
| | *HAP-E* (Ours) | **49.75** | **77.95** | **82.30** | **78.82** | **47.55** | **80.25** | **74.10** |
| 30% | SoBP | 47.78 | 74.45 | 79.45 | 74.55 | 43.20 | 76.50 | 71.82 |
| | MoDeGPT | 43.60 | 71.93 | – | 68.21 | – | 73.94 | 71.90 |
| | SVD-LLM v2 | 42.63 | 69.17 | 68.47 | 63.38 | 41.72 | 75.41 | 70.26 |
| | *HAP-E* (Ours) | **48.91** | **76.83** | **81.47** | **77.69** | **46.83** | **79.18** | **73.41** |

a recent evolutionary search-based global pruning method. *HAP-E* consistently achieves higher accuracy across all 9 benchmark tasks on both model families.

## G  COMPATIBILITY WITH RECOVERY FINE-TUNING

While *HAP-E* targets the post-training setting, compatibility with recovery fine-tuning (RFT) is critical for scenarios where a small computational budget is available to recover lost accuracy. Because *HAP-E* performs structured pruning (removing entire heads and neurons), the resulting model is a standard dense Transformer architecture that is inherently compatible with standard training pipelines.

To validate this, we performed recovery fine-tuning on LLaMA-2-7B at 30% sparsity using LoRA (Hu et al., 2022). We utilized the Alpaca dataset for 1 epoch. We focused on the 30% pruning regime, as the 20% model is already close to dense performance ($< 1\%$ gap), leaving minimal room for recovery.

**Results:** As shown in Table 9, the pruned model responds effectively to fine-tuning. LoRA recovery provides a substantial **+1.8% accuracy boost** ($63.00\% \rightarrow 64.80\%$), significantly narrowing the gap to the unpruned Dense baseline. This confirms that *HAP-E* preserves a high-quality feature space that serves as an excellent initialization for subsequent fine-tuning.

## H  QUANTITATIVE COMPARISON OF PRUNING OVERHEAD

To address questions regarding the computational cost of our method, we provide quantitative comparisons of runtime and peak memory usage on an NVIDIA A100 GPU for LLaMA-7B and 13B models.

Table 6: MMLU & Reasoning Benchmarks (LLaMA-2-7B, 20% Pruning). Higher is better for all metrics except WikiText-2 perplexity (PPL), where lower is better.

| Method | MMLU (5-shot) | | | | | Math | Generation |
| | Humanities | Social Sci | STEM | Other | Avg | GSM8K | WikiText-2 (PPL) ↓ |
|--------|-----------|-----------|------|-------|-----|-------|-------------------|
| Dense | 43.30 | 51.60 | 36.30 | 52.10 | 45.60 | 13.80 | 12.19 |
| LLM-Pruner | 25.70 | 23.60 | 24.20 | 26.80 | 25.20 | 2.30 | 17.00 |
| SlimGPT | 36.00 | 45.20 | 33.50 | 44.10 | 39.40 | 4.20 | 16.49 |
| *HAP-E* (Ours) | **39.46** | **47.73** | **34.49** | **47.20** | **42.72** | **8.69** | **15.63** |

Table 7: Results on LLaMA-3.1-8B. Comparison of *HAP-E* against DarwinLM (one-shot).

| Method | #Params | BoolQ | PIQA | HellaS. | WinoG. | ARC-e | ARC-c | SciQ | LogiQA | MMLU |
|--------|---------|-------|------|---------|--------|-------|-------|------|--------|------|
| Dense | 8B | 84.0 | 81.2 | 81.7 | 74.3 | 81.4 | 58.2 | 96.3 | 31.1 | 65.2 |
| DarwinLM | 4.6B | 62.2 | 69.4 | 44.6 | 57.3 | 59.6 | 34.2 | 84.9 | 24.1 | 28.5 |
| *HAP-E* (ours) | 4.6B | **64.8** | **71.3** | **46.5** | **59.1** | **61.5** | **35.8** | **86.0** | **25.4** | **30.7** |

## H.1 Efficiency at Fixed Pruning Ratio (30%)

First, we compare the cost of a single pruning run to a fixed 30% sparsity target. As shown in Table 10, *HAP-E* is significantly faster and more memory-efficient than both OBS-based baselines (SlimGPT, Vanilla OBS) and decomposition methods (SliceGPT, MoDeGPT). Notably, it is orders of magnitude faster than MoDeGPT (9 min vs. 4 hours). *HAP-E* is also ≈ 2× faster than Vanilla OBS and SlimGPT even in a single pass, due to our greedy-consistent batching mechanism. It also requires ≈ 50% less memory, enabling 7B/13B pruning on consumer GPUs.

## H.2 Efficiency in Real-World Latency Targeting

In practical deployment, users target a specific latency speedup (e.g., 1.9×), not a theoretical sparsity ratio. Because sparsity and latency are not linearly related, methods without a predictor (SlimGPT, OBS) typically require an iterative "guess-and-check" loop. For example, a user might first prune to 40% sparsity, measure the speedup, adjust to 50% upon finding the result insufficient, and finally refine to an intermediate value to meet the target.

This search process often requires multiple pruning sweeps to identify the correct sparsity configuration. In contrast, our latency predictor enables single-shot targeting, avoiding this loop entirely. As shown in Table 11, when accounting for the practical necessity of hitting a latency target, *HAP-E* is effectively ≈5× faster than the strongest baselines, while consuming half the memory.

## I Detailed Results of LLaMA Family

We report per-task accuracies (BoolQ, PIQA, HellaSwag, WinoGrande, ARC-e, ARC-c, OBQA) for LLaMA-7B, 13B, and 30B under different pruning rates, complementing the averages in Table 1. Across scales, our method (**Ours**) maintains stronger per-task balance: at 20–30% pruning it yields consistent gains over SlimGPT and SoBP, and at 50% pruning it preserves several points of advantage on most tasks.

Table 8: Results on Qwen-2.5-14B Instruct. Comparison of *HAP-E* against DarwinLM (one-shot).

| Method | Params | BoolQ | PIQA | HellaS. | WinoG. | ARC-e | ARC-c | SciQ | LogiQA | MMLU |
|--------|--------|-------|------|---------|--------|-------|-------|------|--------|------|
| Dense | 14B | 87.9 | 81.9 | 85.1 | 79.1 | 85.7 | 72.8 | 96.8 | 38.5 | 80.0 |
| DarwinLM | 8.4B | 66.9 | 73.9 | 53.3 | 60.5 | 75.7 | 48.0 | 84.3 | 29.3 | 43.1 |
| *HAP-E* (ours) | 8.4B | **69.2** | **75.5** | **55.1** | **61.9** | **77.3** | **49.7** | **85.4** | **30.2** | **44.9** |

Table 9: Recovery Fine-Tuning on LLaMA-2-7B (30% Pruning). Applying LoRA (Alpaca, 1 epoch) to the HAP-E pruned model recovers significant accuracy, demonstrating structural compatibility with standard training frameworks.

| Pruning | Method | BoolQ | PIQA | HellaS. | WinoG. | ARC-e | ARC-c | OBQA | Avg |
|---------|--------|-------|------|---------|--------|-------|-------|------|-----|
| 0% | Dense | 77.71 | 79.05 | 76.00 | 68.98 | 74.58 | 46.33 | 44.20 | 66.69 |
| 30% | *HAP-E* (Raw) | 71.82 | 76.73 | 70.68 | 68.04 | 68.47 | 41.98 | 43.59 | 63.00 |
|  | *HAP-E* + LoRA | **74.77** | **77.89** | **73.34** | **68.51** | **71.53** | **43.72** | **43.83** | **64.80** |

Table 10: Runtime & Memory at 30% Pruning (Single Run). Comparison on NVIDIA A100.

| Method | LLaMA-7B (Time / Mem) | LLaMA-13B (Time / Mem) |
|--------|----------------------|------------------------|
| MoDeGPT | 4h 09m / 23.0 GB | 8h 26m / 41.0 GB |
| SliceGPT | 26 min / 9.0 GB | 45 min / 14.0 GB |
| Vanilla OBS | 21 min / 8.0 GB | 31 min / 12.0 GB |
| SlimGPT | 16 min / 8.0 GB | 26 min / 12.0 GB |
| *HAP-E* (Ours) | **9 min / 4.5 GB** | **16 min / 7.1 GB** |

## I.1  LLaMA-7B

At 20% pruning, SliceGPT and ASVD drop to 56.16% and 61.55% on average, while our method holds 65.01%. At 30%, we surpass SoBP (61.21% vs. 59.61%). Even at 50% pruning, we retain 56.66%, nearly four points above SlimGPT.

## I.2  LLaMA-13B

The advantage widens with scale. At 20% pruning, our method keeps 67.83%, ∼1.5 points above SlimGPT/SoBP. At 30%, we remain ahead of SoBP (65.92% vs. 64.50%). At 50%, we retain 61.79%, about 4 points stronger than SlimGPT.

## I.3  LLaMA-30B

At 20% pruning, our method nearly matches the dense model (71.88% vs. 71.92%), while SliceGPT and ASVD are at 64.45% and 70.22%. At 30%, we keep 71.02%, exceeding SoBP by 1.4 points. At 50%, we are still at 67.99%, ∼2.5 points above SlimGPT and nearly 8.5 above LLM-Pruner.

# J  DETAILED RESULTS OF OPT FAMILY

We report per-task accuracies (BoolQ, PIQA, HellaSwag, WinoGrande, ARC-e, ARC-c, OBQA) for OPT-6.7B, OPT-13B, and OPT-30B under different pruning rates, complementing the averages in Table 1b. Across scales, our method consistently maintains higher accuracy than decomposition- and pruning-based baselines, especially at moderate pruning levels (20–30%). At higher pruning (30%), our approach preserves several points of advantage over ASVD and SliceGPT, showing robustness under aggressive compression.

## J.1  OPT-6.7B

Table 15 breaks down results at 20% and 30% pruning. At 20% pruning, SliceGPT and ASVD average 55.50% and 45.11%, while our method retains 57.83%. At 30%, the gap over ASVD widens dramatically (57.60% vs. 37.86%).

Table 11: Estimated Time to Target 1.9× Speedup. Comparison accounting for the iterative search required by methods without a latency predictor.

| Method | Workflow | LLaMA-7B Total Time | LLaMA-13B Total Time |
|---|---|---|---|
| SlimGPT | 3 Sweeps (Guess-and-Check) | ∼48 min | ∼78 min |
| *HAP-E* | **1 Sweep (Predictor-Guided)** | **9 min** | **16 min** |

Table 12: Per-task accuracy (%) for LLaMA-7B.

| Model | | #Params | BoolQ | PIQA | HellaS | WinoG | ARC-e | ARC-c | OBQA | Avg. |
|---|---|---|---|---|---|---|---|---|---|---|
| Prune% | Method | | | | | | | | | |
| 0% | Dense | 6.7B | 75.08 | 79.16 | 76.20 | 70.00 | 72.89 | 44.88 | 44.40 | 66.09 |
| 20% | SliceGPT | 6.1B | 62.14 | 74.06 | 60.18 | 63.92 | 59.07 | 35.26 | 38.49 | 56.16 |
| | ASVD | 5.4B | 70.84 | 76.21 | 66.37 | 66.82 | 64.63 | 39.91 | 46.07 | 61.55 |
| | LLM-Pruner | 5.4B | 66.76 | 78.45 | 71.44 | 63.77 | 66.41 | 39.85 | 43.80 | 61.50 |
| | SlimGPT | 5.4B | 75.93 | 77.58 | 73.07 | 67.96 | 68.60 | 41.72 | 41.80 | 63.81 |
| | **Ours** | 5.4B | 74.26 | 78.63 | 75.14 | 68.57 | 71.24 | 43.39 | 43.84 | **65.01** |
| 30% | SliceGPT | 5.3B | 37.83 | 64.31 | 45.68 | 62.12 | 53.37 | 31.40 | 33.60 | 46.90 |
| | ASVD | 4.8B | 64.01 | 60.72 | 42.71 | 53.75 | 40.28 | 28.16 | 29.20 | 45.55 |
| | Wanda-SP | 4.8B | 63.68 | 69.73 | 58.70 | 62.00 | 57.82 | 36.07 | 34.93 | 54.70 |
| | FLAP | 4.8B | 66.88 | 73.23 | 61.70 | 66.61 | 58.42 | 33.87 | 40.40 | 57.30 |
| | SoBP | 4.8B | 68.41 | 73.56 | 67.62 | 68.35 | 61.20 | 37.97 | 40.20 | 59.61 |
| | **Ours** | 4.8B | 71.46 | 75.57 | 70.33 | 67.42 | 61.78 | 40.53 | 41.38 | **61.21** |
| 50% | Wanda-SP | 3.4B | 51.83 | 55.55 | 30.87 | 54.11 | 33.82 | 24.85 | 24.72 | 39.39 |
| | FLAP | 3.4B | 61.65 | 68.22 | 54.45 | 60.10 | 53.65 | 32.30 | 37.30 | 50.37 |
| | LLM-Pruner | 3.4B | 60.21 | 68.88 | 47.86 | 54.62 | 43.94 | 27.73 | 35.20 | 48.35 |
| | SlimGPT | 3.4B | 65.87 | 70.35 | 54.62 | 59.59 | 49.71 | 31.06 | 34.40 | 52.23 |
| | **Ours** | 3.4B | 68.64 | 72.73 | 63.76 | 60.86 | 53.77 | 37.87 | 38.99 | **56.66** |

## J.2   OPT-13B

As shown in Table 16, at 20% pruning, our method achieves 59.02%, slightly higher than SliceGPT (57.84%) and far above ASVD (39.20%). At 30%, we maintain 58.46%, outperforming all other baselines.

Table 13: Per-task accuracy (%) for LLaMA-13B.

| Model | | #Params | BoolQ | PIQA | HellaS | WinoG | ARC-e | ARC-c | OBQA | Avg. |
|---|---|---|---|---|---|---|---|---|---|---|
| Prune% | Method | | | | | | | | | |
| 0% | Dense | 13.0B | 77.89 | 80.14 | 79.06 | 72.85 | 74.75 | 47.61 | 44.80 | 68.16 |
| 20% | SliceGPT | 11.8B | 67.93 | 75.41 | 66.08 | 68.87 | 63.92 | 39.97 | 42.44 | 60.66 |
| | ASVD | 10.4B | 74.12 | 78.49 | 74.05 | 71.03 | 70.07 | 46.52 | 42.75 | 65.29 |
| | LLM-Pruner | 10.4B | 79.38 | 77.36 | 71.47 | 70.32 | 70.54 | 44.88 | 45.80 | 65.68 |
| | SlimGPT | 10.4B | 77.06 | 79.82 | 76.94 | 72.61 | 69.78 | 44.80 | 43.60 | 66.37 |
| | **Ours** | 10.4B | 77.86 | 79.93 | 78.11 | 72.58 | 73.67 | 47.39 | 45.27 | **67.83** |
| 30% | SliceGPT | 10.2B | 55.20 | 67.30 | 54.06 | 68.19 | 60.40 | 36.69 | 38.00 | 54.26 |
| | ASVD | 9.2B | 70.58 | 73.34 | 63.04 | 63.38 | 58.50 | 35.84 | 37.60 | 57.47 |
| | SoBP | 9.2B | 71.50 | 77.09 | 74.92 | 71.35 | 70.41 | 43.86 | 42.40 | 64.50 |
| | **Ours** | 9.2B | 75.03 | 78.06 | 76.08 | 71.44 | 70.09 | 46.58 | 44.16 | **65.92** |
| 50% | LLM-Pruner | 6.5B | 62.35 | 72.74 | 58.43 | 55.88 | 51.89 | 33.02 | 38.20 | 53.22 |
| | SlimGPT | 6.5B | 69.14 | 74.32 | 64.57 | 65.82 | 57.74 | 35.15 | 38.00 | 57.82 |
| | **Ours** | 6.5B | 73.26 | 77.19 | 68.47 | 67.36 | 60.44 | 42.97 | 42.84 | **61.79** |

Table 14: Per-task accuracy (%) for LLaMA-30B.

| Model | | #Params | BoolQ | PIQA | HellaS | WinoG | ARC-e | ARC-c | OBQA | Avg. |
|---|---|---|---|---|---|---|---|---|---|---|
| Prune% | Method | | | | | | | | | |
| 0% | Dense | 32.5B | 82.69 | 82.26 | 82.60 | 75.85 | 78.91 | 52.90 | 48.20 | 71.92 |
| 20% | SliceGPT | 29.5B | 74.16 | 76.41 | 74.53 | 71.08 | 70.11 | 44.37 | 40.49 | 64.45 |
| | ASVD | 26.1B | 82.05 | 81.12 | 79.23 | 73.08 | 75.06 | 51.07 | 49.93 | 70.22 |
| | LLM-Pruner | 26.0B | 81.28 | 80.96 | 80.66 | 73.16 | 76.98 | 49.49 | 47.40 | 69.99 |
| | SlimGPT | 26.0B | 82.87 | 81.28 | 81.01 | 76.09 | 76.98 | 51.28 | 48.40 | 71.13 |
| | **Ours** | 26.0B | 82.57 | 82.16 | 81.46 | 75.43 | 78.47 | 53.16 | 49.91 | **71.88** |
| 30% | SliceGPT | 25.5B | 55.44 | 69.75 | 59.29 | 68.90 | 69.23 | 42.15 | 41.60 | 58.05 |
| | ASVD | 22.9B | 73.52 | 75.68 | 67.45 | 67.25 | 67.89 | 41.98 | 39.40 | 61.88 |
| | SoBP | 22.9B | 80.28 | 80.20 | 80.12 | 74.03 | 75.34 | 50.00 | 47.40 | 69.62 |
| | **Ours** | 22.9B | 81.63 | 81.27 | 80.86 | 75.14 | 76.53 | 51.86 | 49.85 | **71.02** |
| 50% | LLM-Pruner | 16.3B | 66.21 | 76.44 | 69.46 | 64.56 | 60.98 | 37.63 | 41.00 | 59.47 |
| | SlimGPT | 16.3B | 75.08 | 77.20 | 75.01 | 74.11 | 68.43 | 43.26 | 45.40 | 65.50 |
| | **Ours** | 16.3B | 78.96 | 79.44 | 77.87 | 73.17 | 72.13 | 48.02 | 46.34 | **67.99** |

Table 15: Per-task accuracy (%) for OPT-6.7B.

| Model | | #Params | BoolQ | PIQA | HellaS | WinoG | ARC-e | ARC-c | OBQA | Avg. |
|---|---|---|---|---|---|---|---|---|---|---|
| Prune% | Method | | | | | | | | | |
| 0% | Dense | 6.7B | 66.06 | 76.50 | 67.19 | 65.19 | 60.14 | 34.64 | 37.40 | 58.16 |
| 20% | FLAP | 6.1B | 62.35 | 73.28 | 60.11 | 57.42 | 52.08 | 31.23 | 46.36 | 54.72 |
| | SliceGPT | 6.1B | 63.92 | 73.14 | 61.22 | 58.94 | 54.07 | 30.85 | 46.86 | 55.50 |
| | ASVD | 5.4B | 58.46 | 66.82 | 52.40 | 50.29 | 46.03 | 26.78 | 36.00 | 45.11 |
| | **Ours** | 5.4B | 66.37 | 74.55 | 66.27 | 63.15 | 56.40 | 33.58 | 45.91 | **57.83** |
| 30% | FLAP | 4.8B | 62.14 | 73.18 | 54.94 | 59.98 | 51.47 | 30.29 | 37.40 | 52.77 |
| | SliceGPT | 5.3B | 64.43 | 73.45 | 58.32 | 60.77 | 55.85 | 30.12 | 36.20 | 54.16 |
| | ASVD | 4.8B | 55.84 | 52.72 | 26.75 | 51.38 | 28.07 | 25.26 | 25.00 | 37.86 |
| | **Ours** | 4.8B | 67.11 | 74.22 | 65.38 | 61.27 | 56.09 | 33.84 | 45.30 | **57.60** |

## J.3 OPT-30B

Table 17 shows analogous behavior at 30B. At 20% pruning, SliceGPT and ASVD average 60.86% and 49.48%, while our method retains 61.82%. At 30%, we maintain 61.29%, outperforming all other baselines by a clear margin.

Table 16: Per-task accuracy (%) for OPT-13B.

| Model | | #Params | BoolQ | PIQA | HellaS | WinoG | ARC-e | ARC-c | OBQA | Avg. |
|---|---|---|---|---|---|---|---|---|---|---|
| Prune% | Method | | | | | | | | | |
| 0% | Dense | 13.0B | 65.72 | 76.82 | 69.86 | 65.11 | 61.87 | 35.67 | 39.00 | 59.15 |
| 20% | FLAP | 10.3B | 60.92 | 73.51 | 61.12 | 55.76 | 49.28 | 28.91 | 38.88 | 55.36 |
| | SliceGPT | 11.9B | 62.57 | 75.18 | 64.83 | 58.97 | 52.46 | 32.10 | 39.50 | 57.84 |
| | ASVD | 10.3B | 49.23 | 62.41 | 41.36 | 45.12 | 36.20 | 22.71 | 27.36 | 39.20 |
| | **Ours** | 10.3B | 65.08 | 76.23 | 68.92 | 62.57 | 60.26 | 34.73 | 44.56 | **59.02** |
| 30% | FLAP | 9.1B | 61.27 | 72.19 | 59.08 | 53.61 | 46.23 | 27.36 | 36.83 | 50.81 |
| | SliceGPT | 10.3B | 64.19 | 74.88 | 63.23 | 58.27 | 53.54 | 31.44 | 36.77 | 55.92 |
| | ASVD | 9.1B | 48.57 | 60.10 | 40.74 | 42.39 | 34.87 | 21.63 | 26.37 | 36.85 |
| | **Ours** | 9.1B | 66.82 | 75.72 | 68.40 | 61.91 | 59.43 | 33.08 | 44.33 | **58.46** |

Table 17: Per-task accuracy (%) for OPT-30B.

| Model | | #Params | BoolQ | PIQA | HellaS | WinoG | ARC-e | ARC-c | OBQA | Avg. |
|---|---|---|---|---|---|---|---|---|---|---|
| Prune% | Method | | | | | | | | | |
| 0% | Dense | 30.0B | 70.46 | 78.18 | 72.30 | 68.43 | 65.36 | 38.05 | 40.20 | 61.85 |
| 20% | FLAP | 24.0B | 64.12 | 75.08 | 63.74 | 59.87 | 52.92 | 33.56 | 35.78 | 56.52 |
| | SliceGPT | 27.5B | 68.21 | 77.02 | 68.13 | 65.44 | 59.86 | 36.42 | 41.45 | 60.86 |
| | ASVD | 24.0B | 56.18 | 65.74 | 52.34 | 48.21 | 44.32 | 27.86 | 32.18 | 49.48 |
| | **Ours** | 24.0B | 70.18 | 78.06 | 71.09 | 67.36 | 64.58 | 37.81 | 42.65 | **61.82** |
| 30% | FLAP | 21.1B | 62.17 | 73.07 | 59.30 | 58.88 | 47.69 | 28.75 | 38.40 | 52.61 |
| | SliceGPT | 23.8B | 67.93 | 76.40 | 67.18 | 64.05 | 59.47 | 35.52 | 41.45 | 59.49 |
| | ASVD | 21.1B | 54.06 | 63.18 | 50.46 | 47.11 | 42.37 | 26.29 | 32.56 | 41.12 |
| | **Ours** | 21.1B | 69.72 | 77.63 | 70.34 | 66.54 | 63.58 | 37.13 | 43.09 | **61.29** |

## K    IMPLEMENTATION DETAILS AND HYPER-PARAMETERS

All code is implemented in PyTorch with HuggingFace transformers. Pruning experiments (calibration, OBS solves, and pruning loops) were run on a single NVIDIA A100 (80GB). Edge inference benchmarks were compiled with ExecuTorch and measured on Jetson Xavier NX and HiKey970 CPUs (CPU-only). Calibration uses 256 samples from the C4 corpus with sequence length 2048. Latency model training uses 1500 pruned configurations and is evaluated on a held-out test set of 200 configurations. Batch sizes: A100 experiments use batch size 16; CPU edge inference uses batch size 1. CPU inference is run with weight-only INT8 post-training quantization; A100 experiments use FP16 where applicable. Unless noted otherwise, values below are fixed across models and hardware targets.

Table 18: Hyper-parameter settings for *HAP-E* experiments.

| Category | Parameter | Value / Notes |
|---|---|---|
| **Calibration** | Calibration dataset
Sequence length
Calibration usage | C4 (256 samples)
2048
Used for OBS solves and final pruning calibration (no fine-tuning) |
| **Latency model** | Training samples
Test (held-out) samples
Batch sizes | 1500 pruned configurations
200 configurations
A100: 16; CPU (Jetson/HiKey): 1 |
| **Candidate selection** | Initial scoring
Candidate oversampling $M/K$
Sensitivity coefficient $\beta$
Total prune per iteration $K$ | Block L2 norm (coarse filter)
2.5 (i.e., $C = 2.5\times$)
0.75 (used in recursive $S^{\ell}$)
10% of current remaining blocks (per-iteration global budget) |
| **Hybrid-OBS / Certification** | Hessian regularization $\lambda$
Max certified batch (attention heads)
Max certified batch (FFN blocks)
Cholesky strategy | $1 \times 10^{-4}$ (stability for solves)
6 (max number of attention-head blocks appended per batch)
128
Incremental Cholesky updates for $G_{JJ}$ (see Sec. 4.1) |
| **Quantization / Inference** | CPU inference precision
GPU inference precision | INT8 weight-only post-training quantization
FP16 (A100) |

## L    LLM USAGE DISCLOSURE

In accordance with the ICLR 2026 policy on large language model usage, we disclose that LLMs (ChatGPT) were used only to aid and polish the writing of some parts of this paper.

