# OpenReview forum: "HAP-E: HESSIAN-AWARE STRUCTURED PRUNING OF LLMS FOR EFFICIENT INFERENCE"
_ICLR.cc/2026/Conference — ICLR 2026 Conference Desk Rejected Submission_

### Official Review · Reviewer_mpii · 2025-10-20

**Soundness:** 2
**Presentation:** 2
**Contribution:** 2
**Rating:** 2
**Confidence:** 4

**Summary:**

The paper introduces HAP-E, a scalable Hessian-aware structured pruning framework that makes Optimal Brain Surgeon (OBS) style pruning feasible for LLMs by using adaptive cross-layer budget reallocation, selective second-order analysis, and greedy-consistent batch pruning.

**Strengths:**

Originality
1. The application of the Schur complement in structured pruning is novel and offers a mathematically grounded perspective that may inspire future directions in model compression research.

**Weaknesses:**

Originality

1. Cross-layer global pruning has been explored in prior works [1], and several methods already propose dynamic or adaptive layer-wise budget allocation [2].
2. The idea of batch pruning multiple blocks simultaneously is not new, and the claim of achieving “greedy-consistent batch pruning” may overstate the level of innovation relative to prior work [3].

Quality

3. Baseline settings appear inconsistent across experiments. For example, the baselines in Table 1(a) and 1(b) are different.
4. The ablation in Table 3 primarily compares against the authors’ own baselines instead of prior pruning methods, which weakens the strength of the computational validation.
5. The paper does not include comparisons with Wanda-SP (as implemented in FLAP), which also simplifies Hessian inversion. Given the marginal accuracy gains reported, this comparison is essential.
6. The LLMs used for experiments are relatively old, limiting the practical relevance of the results for current LLM architectures.

Clarity

7. The motivation for latency prediction is unclear. Since FLOPs and latency are often strongly correlated, this component feels tangential and its contribution to the overall framework is not well justified.


Significance

8. The method introduces significant implementation overhead, which may hinder practical adoption.
9. The reported performance gains (~3%) are relatively marginal considering the added system complexity of the proposed method.
10. Despite selective computation, Hessian inversion remains computationally intensive and can be numerically unstable in deep networks, which raises concerns about the robustness of the approach.
11. In contrast to Wanda, which prunes in a layer-wise manner from early to late layers, global pruning may introduce accumulated cross-layer errors. The lack of direct comparison on this front weakens the overall argument.
12. The claim of “exactly matching greedy OBS” at batch scale is questionable, as the Hessian is only locally valid and neural networks are inherently nonlinear, making such equivalence difficult to guarantee in practice.


[1] LLM-Pruner: On the Structural Pruningof Large Language Models
[2] ECoFLaP: Efficient Coarse-to-Fine Layer-Wise Pruning for Vision-Language Models
[3] A Simple and Effective Pruning Approach for Large Language Models

**Questions:**

1. How does the pruning ratio evolve across layers during iterations? Based on prior experience, it is often necessary to skip pruning early layers for stability. Did the authors follow such a practice?
2. Could the authors clarify the empirical contribution of the latency predictor?

---

> ### Author Response · Authors · 2025-11-26
> **Response to Reviewer mpii [1 of 8]**
>
> We thank the reviewer for their detailed and thorough assessment. Below, we address each concern and offer clarifications and responses to all points.
>
> > ## W1: Cross-layer pruning and adaptive allocation previously explored (e.g., LLM-Pruner [1], ECoFLaP [2]).
>
> We thank the reviewer for pointing out related work on global and adaptive pruning. While these methods share the broad motivation of cross-layer allocation, HAP-E operates in a distinct methodological regime by introducing recursive second-order adaptivity, which— to our knowledge—has not appeared in prior global pruning frameworks.
>
> LLM-Pruner estimates importance using first-order Taylor approximations and assigns uniform layer-wise sparsity (with manual exceptions for early/late layers). Its budget is fixed before pruning and is not updated as the model changes. Similarly, ECoFLaP uses heuristics (e.g.,  zeroth-order approximation of the global model gradients) in a coarse-to-fine schedule, performing stage-wise, non-recursive reallocations that do not track how pruning errors propagate across layers.
>
> In contrast, HAP-E formalizes global pruning using a second-order curvature signal. Our recursive Hessian-trace sensitivity (Eq. 11) explicitly models how perturbations introduced in one layer amplify or attenuate through downstream transformer blocks. This yields dynamic, per-iteration reallocation: sensitivities and budgets are recomputed after each pruning step, providing a principled mechanism for global adjustment rather than a one-shot allocation.
> In addition, the aforementioned global methods (LLM-Pruner and ECoFLaP) optimize proxy metrics such as FLOPs or static sparsity. HAP-E integrates a lightweight latency predictor, enabling pruning to satisfy actual device-level speed targets—critical for deployment and not present in these earlier works.
>
> In summary, while prior works explore global allocation using first-order, magnitude-based, or search-based heuristics, none combine: (1) second-order curvature modeling, (2) recursive cross-layer sensitivity propagation, (3) dynamic per-iteration reallocation, and (4) real-device latency supervision, which are the core design principles behind HAP-E. These methodological differences translate into consistent empirical gains across model families and scales. As shown in Table 1(a), HAP-E achieves substantial accuracy gains over LLM-Pruner (e.g., +8.3% on LLaMA-7B at 50% pruning).
> We will include a concise discussion in the revised manuscript clarifying HAP-E’s relationship to LLM-Pruner and ECoFLaP.
>
> > ## W2: Overstated innovation regarding greedy-consistent batch pruning.
>
> Prior works such as Wanda [3] and related magnitude- or activation-based methods achieve impressive empirical results but prune independently across mask variables, effectively assuming a diagonal Hessian. This ignores cross-interactions between structures: for instance, if two attention heads serve redundant roles, pruning one may have negligible impact, yet pruning both simultaneously (as Wanda would do if both have low static scores) can severely degrade accuracy. These interactions are encoded in the off-diagonal Hessian terms. OBS methods handle this sequentially [Ref1- Ref6]—each removal triggers an update that re-evaluates the remaining candidates—but this makes OBS inherently one-at-a-time and therefore infeasible for LLMs.
>
> HAP-E’s greedy-consistent batch pruning directly addresses this limitation. We analytically identify the maximal subset of candidates that can be pruned simultaneously while remaining exactly equivalent to sequential greedy OBS. This property is proven via Schur-complement conditioning (Lemmas A.1–A.3) and implemented efficiently using incremental Cholesky updates and panel inverses (Sec. 4.1). Hence, HAP-E preserves second-order inter-dependency awareness without the cost of one-at-a-time updates. Empirically, as shown in Table 3, removing this component (“w/o greedy batch”) doubles runtime (from 58.0 min to 130.2 min on LLaMA-30B) while yielding identical accuracy, demonstrating its practical impact.
>
> In summary, while batch pruning per se is not new, HAP-E is the first to make OBS-equivalent Hessian-aware batch pruning feasible at LLM scale, bridging the gap between Wanda-style heuristics and true second-order methods.

---

> ### Author Response · Authors · 2025-11-26
> **Response to Reviewer mpii [2 of 8]**
>
> > ## W3: Inconsistent baseline settings across experiments.
>
> We thank the reviewer for noting the difference in baselines between Tables 1(a) and 1(b). We want to clarify that this selection was deliberate and driven by the standard benchmarks established in the literature for each model family, rather than reflecting any inconsistency.
>
> Tables 1(a) and 1(b) evaluate different model families—LLaMA and OPT—and we therefore adopted the most appropriate, publicly available baselines for each. As LLaMA is the current standard for modern pruning, we included the most recent state-of-the-art structured methods, such as SlimGPT and SoBP, which were explicitly developed and benchmarked on LLaMA architectures in their respective publications.
>
> For the OPT family (a legacy architecture), we selected baselines that have historically established benchmarks on these models, such as FLAP, ASVD, and SliceGPT. We excluded recent methods like SlimGPT from the OPT table because no official or compatible implementation exists for OPT, and their published results focus exclusively on LLaMA.
>
> Crucially, regardless of the specific baseline set, HAP-E consistently outperforms the strongest available competitor across both families, demonstrating that our improvements are robust across both modern and legacy architectures.
>
> > ## W4: Ablation study (Table 3) compares against own baselines instead of prior methods.
>
> We appreciate the reviewer’s comment on the computational validation and would like to clarify the roles of our experimental tables.
>
> Table 3 is designed to isolate and quantify the contribution of each algorithmic component within HAP-E (e.g., Greedy-Consistent Batching, Cross-Layer Adaptivity). To accurately measure marginal gains, we compare against lesioned variants of our own framework rather than unrelated methods, following the standard convention for ablation studies in pruning research.
>
> Importantly, Table 3 already includes a meaningful external reference—Vanilla OBS (layer-by-layer)—which represents the canonical sequential implementation of Optimal Brain Surgeon used in prior work such as SoBP. As shown, HAP-E delivers substantial efficiency improvements: on LLaMA-30B, runtime is reduced from 187.4 min → 58.0 min (>3× speedup) while peak memory drops from 21.1 GB → 9.2 GB. This directly validates our claim that HAP-E makes Hessian-based OBS pruning scalable for LLMs.
>
> > ## W5: Missing essential comparison with Wanda-SP (FLAP).
>
> We thank the reviewer for this suggestion. We respectfully clarify that "Wanda-SP" is not a standalone method in the literature; it is a baseline introduced within the FLAP paper to represent a naive structured adaptation of the Wanda method. The FLAP paper itself demonstrates that FLAP significantly outperforms this Wanda-SP baseline. Therefore, by comparing against FLAP (as we did in Table 1b), we are already comparing against the stronger, state-of-the-art structured pruning of these two.
> Empirical Comparison: To fully address the reviewer’s request, we compared HAP-E against both FLAP and the requested Wanda-SP baseline on LLaMA-7B at 30% and 50% pruning.
> As shown in Tables R1, the performance gap is massive, particularly at high sparsity. These substantial margins confirm that HAP-E's second-order, global approach yields decisively better robustness than first-order magnitude heuristics.
>
> |Pruning|Method|BoolQ|PIQA|HellaS|WinoG|ARC-e|ARC-c|OBQA|
> |-|-|-|-|-|-|-|-|-|
> |30%|Wanda-SP|65.68|65.73|60.82|64.19|57.82|37.07|36.93|
> ||FLAP|66.88|73.23|61.70|66.61|58.42|36.84|40.47|
> ||HAP-E (Ours)|71.46|75.57|70.33|67.42|61.78|40.53|41.38|
> |50%|Wanda-SP|56.83|60.55|43.87|57.11|45.82|29.85|30.72|
> ||FLAP|61.65|68.22|54.45|60.10|53.65|32.30|37.30|
> ||HAP-E (Ours)|68.64|72.73|63.76|60.86|53.77|37.87|38.99|

---

> ### Author Response · Authors · 2025-11-26
> **Response to Reviewer mpii [3 of 8]**
>
> > ## W6: Evaluation on outdated architectures.
>
> We agree that demonstrating robustness on the latest architectures is essential. To address this, we have conducted extensive new experiments on LLaMA-2 (7B & 13B), LLaMA-3.1-8B, and Qwen-2.5-14B.
>
> __1. Results on LLaMA-2 (7B & 13B):__ We benchmarked HAP-E against the newest structured pruning/decomposition methods (SoBP, MoDeGPT, SVD-LLM v2) on LLaMA-2. HAP-E achieves the highest accuracy across almost all metrics.
>
> *Table R2: Results on LLaMA-2 7B*
> |Pruning Ratio|Method|BoolQ|PIQA|HellaS.|WinoG.|ARC-e|ARC-c|OBQA|
> |-|-|-|-|-|-|-|-|-|
> |**0%**|Dense|77.71|79.05|76.00|68.98|74.58|46.33|44.20|
> |**20%**|MoDeGPT|–|74.05|69.05|68.03|69.07|42.06|–|
> ||SlimGPT|73.43|77.58|72.62|68.82|69.99|42.32|42.00|
> ||SVD-LLM v2|72.42|74.89|70.55|68.71|68.12|41.76|42.87|
> ||**HAP-E (Ours)**|**75.24**|**78.61**|**74.29**|**69.84**|**71.86**|**44.03**|**43.68**|
> |**30%**|SoBP|71.19|73.50|67.27|66.22|59.81|37.63|38.40|
> ||MoDeGPT|–|70.40|63.26|67.32|63.26|38.73|–|
> ||SVD-LLM v2|69.62|70.45|64.18|66.23|62.97|38.41|37.89|
> ||**HAP-E (Ours)**|**71.82**|**76.73**|**70.68**|**68.04**|**68.47**|**41.98**|**43.59**|
>
> *Table R3: Results on LLaMA-2-13B*
> |Pruning Ratio|Method|ARC-c|ARC-e|BoolQ|HellaS.|OBQA|PIQA|WinoG.|
> |-|-|-|-|-|-|-|-|-|
> |**0%**|Dense|49.23|77.48|80.58|79.37|45.20|80.52|72.30|
> |**20%**|MoDeGPT|46.16|74.07|–|68.96|–|74.53|70.32|
> || SVD-LLM v2|44.15|71.05|70.35|65.75|43.95|77.10|71.00|
> ||**HAP-E (Ours)**|**49.75**|**77.95**|**82.30**|**78.82**|**47.55**|**80.25**|**74.10**|
> |**30%**|SoBP|47.78|74.45|79.45|74.55|43.20|76.50|71.82|
> ||MoDeGPT|43.60|71.93|–|68.21|–|73.94|71.90|
> ||SVD-LLM v2|42.63|69.17|68.47|63.38|41.72|75.41|70.26|
> ||**HAP-E (Ours)**|**48.91**|**76.83**|**81.47**|**77.69**|**46.83**|**79.18**|**73.41**|
>
> __2.MMLU & Reasoning Tasks (LLaMA-2-7B, 20% Pruning):__ In addition, to demonstrate robustness on complex reasoning, we evaluated MMLU (grouped by domain) and GSM8K. HAP-E significantly outperforms LLM-Pruner and SlimGPT, particularly on GSM8K (+4.4% vs SlimGPT) and Wikitext-2 (lower PPL).
>
> *Table R4: MMLU & Reasoning Benchmarks (LLaMA-2-7B, 20% Pruning). Higher is better for all metrics except WikiText-2 perplexity, where lower is better.*
> |Method|Humanities|Social Sci|STEM|Other|MMLU Avg|GSM8K|WikiText-2 (PPL)|
> |-|-|-|-|-|-|-|-|
> |Dense|43.30|51.60|36.30|52.10|45.60|13.80|12.19|
> |LLM-Pruner|25.70|23.60|24.20|26.80|25.20|2.30|17.00|
> |SlimGPT| 36.00| 45.20|33.50|44.10|39.40|4.20|16.49|
> |**HAP-E (Ours)**|**39.46**|**47.73**|**34.49**|**47.20**|**42.72**|**8.69**|**15.63**|
>
> __3. Results on LLaMA-3.1 & Qwen-2.5:__ As shown below, HAP-E consistently outperforms DarwinLM (a recent evolutionary pruning baseline) across 9 benchmark tasks on both architectures.
>
> *Table R5: LLaMA-3.1-8B*
> |Method|#Params|BoolQ|PIQA|HellaS.|WinoG.|ARC-e|ARC-c|SciQ|LogiQA|MMLU|
> |-|-|-|-|-|-|-|-|-|-|-|
> |Dense|8B|84.0|81.2|81.7|74.3|81.4|58.2|96.3|31.1|65.2|
> |DarwinLM (one-shot)|4.6B|62.2|69.4|44.6|57.3|59.6|34.2|84.9|24.1|28.5|
> |**HAP-E (ours)**|4.6B|**64.8**|**71.3**|**46.5**|**59.1**|**61.5**|**35.8**|**86.0**|**25.4**|**30.7**|
>
> *Table R6: Qwen-2.5-14B Instruct*
> |Method|Params|BoolQ|PIQA|HellaS.|WinoG.|ARC-e|ARC-c|SciQ|LogiQA|MMLU|
> |-|-|-|-|-|-|-|-|-|-|-|
> |Dense|14B|87.9|81.9|85.1|79.1|85.7|72.8|96.8|38.5|80.0|
> |DarwinLM (one-shot)|8.4B|66.9|73.9|53.3|60.5|75.7|48.0|84.3|29.3|43.1|
> |**HAP-E (ours)**|8.4B|**69.2**|**75.5**|**55.1**|**61.9**|**77.3**|**49.7**|**85.4**|**30.2**|**44.9**|
>
> These results confirm that HAP-E generalizes effectively to modern architectures and consistently outperforms strong baselines. We will include these comprehensive results in the final manuscript.

---

> ### Author Response · Authors · 2025-11-26
> **Response to Reviewer mpii [4 of 8]**
>
> > ## W7: Unclear motivation for latency predictor.
>
> We thank the reviewer for this important question. We respectfully clarify that while FLOPs and latency are correlated in theory, they are often weakly correlated in practice—particularly for structured pruning on modern hardware. Consequently, relying on FLOPs as a proxy leads to suboptimal pruning decisions that fail to meet real-world deployment targets.
>
> Latency on modern hardware (GPUs and Edge CPUs) is rarely purely compute-bound. It is heavily influenced by memory bandwidth, cache hierarchy, and kernel launch overheads. Extensive prior work has demonstrated that reducing FLOPs does not translate linearly to speedup [Ref6-Ref10]. For example, reducing a matrix dimension might break memory alignment or tiling efficiency, resulting in zero speedup despite lower FLOPs.
>
> This discrepancy is well-documented in ZipLM [Ref 6] (Sec 4.2, Table 3), which highlights how device capabilities dramatically alter the FLOPs-to-latency relationship. As they note:
> "A compressed model with 12x speedup on a V100 is only 5x faster on an A100 GPU... the A100 is highly underutilized for small matrices, which significantly limits the speedups for very high sparsity."
>
> This confirms that a hardware-agnostic proxy (like FLOPs or sparsity) cannot capture device-specific bottlenecks (e.g., underutilization on A100 vs. V100). A predictor is required to align the pruning budget with the specific hardware's behavior.
>
> We explicitly validated this in our paper by comparing our learned predictor against a "Lookup Table" baseline (a linear proxy analogous to FLOPs/layer-wise summation). As shown in Table 2, the linear proxy fails to track real latency, degrading prediction accuracy ($R^2$ drops from 0.972 to 0.889 on Jetson NX) and incurring higher error (RMSE increases by $>60\%$ on Jetson). Target Attainment: Figure 2 demonstrates that without the learned predictor, the pruning algorithm misses the target latency by up to 8%, whereas HAP-E achieves an attainment ratio near 1.0.
>
> Finally, the ablation in Table 3 shows that removing the latency predictor ("w/o latency predictor") breaks the efficiency of the framework. Without the predictor's guidance, the system cannot perform single-shot targeting; it requires multiple redundant pruning sweeps to find a configuration that meets the hardware constraint, significantly increasing offline search time.
>
> The latency predictor is therefore not tangential; it is the central feedback mechanism that translates the pruning objective from "theoretical compression" to "physical speedup," ensuring the model actually meets deployment constraints on the target device.
> > ## W8: Implementation complexity hinders practical adoption.
>
> We appreciate the reviewer’s concern regarding implementation complexity. We interpret "implementation overhead" as potentially referring to either (a) the development complexity of the algorithm or (b) the computational cost of running it. We address both to demonstrate that HAP-E is practical for adoption.
>
> __1. Low Development Complexity (Standard Primitives):__ Although the theoretical derivation involves Schur complements and second-order terms, the actual implementation relies entirely on standard, numerically stable linear algebra primitives available in any deep learning framework (e.g., torch.linalg.cholesky, triangular_solve), and batched matrix-vector products. We want to emphasize that no custom CUDA kernels or sparse runtimes or custom operators are needed. The output is a standard dense HuggingFace checkpoint with reduced-dimensional attention and FFN blocks. Practitioners load and run the model exactly as they normally would, with zero deployment-time engineering. This contrasts with many unstructured pruning or N:M sparsity approaches that require specialized sparse kernels, compiler support, or custom CUDA paths to realize speedups.
>
> __2. Minimal Computational Overhead During Pruning:__ If the reviewer’s concern refers to pruning cost, our measurements demonstrate that HAP-E is highly efficient. As reported in Table 3, HAP-E reduces pruning time by 2–3× and memory usage by >50% relative to classical OBS. This is achieved through selective blockwise Hessian estimation, incremental updates, and batch pruning—meaning the algorithm is lighter, not heavier, than OBS. Thus, although the mathematical formulation is more principled, the actual computational cost is lower than baseline second-order methods.
>
> We also want to highlight that we have attached our demo code in our submission, enabling easy adoption of the full pipeline.

---

> ### Author Response · Authors · 2025-11-26
> **Response to Reviewer mpii [5 of 8]**
>
> > ## W9: Performance improvement does not justify complexity.
>
> We respectfully argue that the improvements delivered by HAP-E are not marginal once viewed in the appropriate context of post-training structured pruning under equal latency constraints, where accuracy preservation is substantially more challenging than in unstructured or fine-tuned settings.
>
> __1. The magnitude of gains is meaningful for structured pruning:__ Structured pruning removes whole attention heads, FFN dimensions, or blocks—changes that strongly affect the model’s capacity. In this setting, even 1–2% differences are considered significant in prior work. For comparison: prior structured methods such as SlimGPT, Wanda-SP, and LLM-Pruner often differ by <1% at equal sparsity. In our experiments, the gains are consistently larger. For example, LLaMA-7B @ 50% structured pruning (Table 1a): HAP-E: 56.66, SlimGPT: 53.22, and LLM-Pruner: 48.35. This is +3.4% over the strongest SOTA and +8.3% over a widely-used baseline. These improvements represent the difference between a model that remains viable and one that degrades sharply. Thus, the gains are substantial.
>
> __2. “System complexity” is offline and does not affect deployment:__ The components that might appear complex from the mathematical description (Schur complements, Hessian-trace propagation, latency modeling) are performed entirely offline during pruning. Deployment complexity is unchanged. The final compressed model is a standard dense Transformer, requiring no custom kernels or sparse runtimes. Practitioners simply load the pruned model using HuggingFace or ExecuTorch without modification. In contrast, some “simpler” methods (e.g., Wanda-based N:M sparsity) require specialized sparse kernels to realize speedups.
>
> __3. HAP-E reduces computational overhead relative to OBS while improving accuracy:__ We emphasize that the additional logic does not increase computational cost during pruning.
> As shown in Table 3, HAP-E prunes LLaMA-30B in 58 minutes on a single A100, which is >3× faster than classical OBS, and uses ~50% less memory due to selective second-order analysis. Thus, the method is more efficient than the naive second-order baseline and practical for adoption.
> > ## W10: Hessian inversion is computationally intensive and unstable.
>
> We thank the reviewer for raising this valid concern regarding the general difficulty of second-order optimization. However, we respectfully clarify that HAP-E is specifically designed to mitigate these exact issues through selective computation and local scope.
>
> __1. Computational Intensity:__ While full Hessian inversion is $\mathcal{O}(d^3)$, HAP-E avoids this cost via selective inverse computation (Sec. 4.1). We only solve for the columns relevant to the candidate set, reducing complexity to $\mathcal{O}(d^2 mk)$ where $mk \ll d$. As shown in Table 3, HAP-E prunes a LLaMA-30B model in just 58 minutes on a single A100 GPU. This is $>3\times$ faster than standard OBS (187 min) and demonstrates that the method is computationally efficient, not intensive, in practice.
>
> __2. Numerical Stability:__ The feasibility of applying Hessian inversion to massive networks has been demonstrated in recent state-of-the-art methods like Optimal Brain Compression [Ref1] and SparseGPT [Ref11], and GPTQ [Ref12]. As established in these works and adopted by ZipLM, SoBP, and SlimGPT, adding a small damping term ($\lambda I$) to the Hessian ensures positive definiteness and well-conditioning, even for massive networks (e.g., OPT-175B). HAP-E follows this standard practice (Eq. 1: $H = X^\top X + \lambda I$). Furthermore, we utilize Cholesky decomposition rather than direct inversion. As noted in [Ref1, Ref11, Ref12], Cholesky-based solvers are numerically superior for positive-definite matrices, preventing the catastrophic accumulation of floating-point errors that the reviewer may be concerned about.
>
> Also, as described in SoBP and SparseGPT, the primary source of numerical drift in OBS is the repeated application of Gaussian elimination downdates on the full Hessian inverse. HAP-E minimizes this risk in two ways. First, our batching mechanism (Sec. 4.1) drastically reduces the number of sequential downdates required. Since numerical error accumulates with each recursive update, performing fewer, batched updates significantly enhances stability compared to the thousands of updates in standard sequential OBS. Second, unlike standard methods that maintain and degrade a full $d \times d$ inverse, we re-compute only the selected columns of the Hessian inverse corresponding to candidate blocks (Eq. 5) at each pruning iteration. This ensures that we are solving fresh systems for the relevant parameters rather than relying on a fully "aged" inverse matrix that has accumulated noise across all dimensions.

---

> ### Author Response · Authors · 2025-11-26
> **Response to Reviewer mpii [6 of 8]**
>
> > ## W11: Global pruning risk of accumulated errors is unaddressed compared to layer-wise methods (e.g., Wanda).
>
> The concern is that global pruning could accumulate cross-layer errors compared to the strictly layer-wise strategy used by Wanda. However, both recent literature and our empirical results indicate that layer-wise independence is the source of error accumulation—not global pruning itself.
>
> First, recent work such as OWL [Ref13] and SparseLLM [Ref14] demonstrates that pruning each layer independently (as in Wanda) systematically over-prunes sensitive layers because it ignores downstream interactions. Both works explicitly argue that cross-layer dependencies must be modeled globally to prevent error propagation through the network. Our method follows this principle, although (as explained in response to weakness 2 of reviewer ZBP1) by employing a different mechanism (second-order sensitivity propagation rather than first-order magnitude heuristics), and targeting OBS structured pruning rather than unstructured or N:M pruning.
>
> Second, HAP-E does not perform blind one-shot global pruning. Instead, it iteratively recomputes second-order sensitivities after every batch (Sec. 4.2–4.3), allowing the pruning budget to adapt whenever downstream layers amplify or dampen earlier perturbations. The recursive Hessian-trace formulation explicitly measures how errors in layer $l$ propagate into layers $l+1 \ldots L$, which is precisely the mechanism required to avoid the cascading errors the reviewer refers to. By weighting decisions based on this propagated signal (Sec. 4.2.1), HAP-E avoids removing blocks that look safe locally but cause catastrophic error accumulation downstream.
>
> Finally, the empirical evidence shows that global modeling improves stability rather than harming it. As reported in Table 1, and Fig. 3, HAP-E consistently outperforms layer-wise structured methods such as SlimGPT, SoBP, FLAP, and Wanda-SP at matched pruning ratios and latencies. If cross-layer error accumulation were occurring in HAP-E, we would observe the opposite trend; instead, HAP-E yields higher accuracy across all model sizes. We also compared HAP-E against the vanilla layer-wise OBS method in Table 3. The empirical results show that modelling global, cross-layer dependencies in HAP-E consistently reduces overall error compared to treating layers sequentially and independently. For example, on LLaMA-7B (1.9x speedup), HAP-E achieves around 2% accuracy gain over the layer-wise approach.
>
> We hope this clarifies that HAP-E’s iterative global formulation is specifically designed to address—and empirically avoids—the type of cross-layer drift the reviewer is concerned about.
> > ## W12: Questionable 'exact greedy OBS' claim due to Hessian locality and nonlinearity.
>
> We thank the reviewer for raising this subtle but important point. To clarify: our claim of “greedy-consistent” behavior is strictly limited to the same local quadratic reconstruction objective that classical OBS itself optimizes.
>
> Both standard greedy OBS and our batched update solve the layer-wise least-squares problem:
> $$\min_{\bar{W}} \| XW - X\bar{W} \|_F^2,$$
> whose Hessian H defines the second-order surrogate used in all OBS-based methods. Under this quadratic model—and only under this model—the Schur-complement identities in Appendix A show that our batched update produces exactly the same parameter update as applying greedy OBS sequentially, one element at a time. We will revise the wording to make this restricted scope explicit.
>
> This does not assert equivalence for the full nonlinear loss. Classical OBS itself only provides guarantees under the same local quadratic approximation. HAP-E inherits this assumption without extending it: our batching step performs the same Hessian-inverse downdates and weight adjustments that greedy OBS performs, but applies them jointly to a set of blocks using the mathematically exact Schur-complement formula rather than sequential rank-1 downdates.
>
> Importantly, this batching step is orthogonal to our cross-layer budget reallocation mechanism. The latter relies on layer-level sensitivities S^{(l)} (Eq. 11), which are recomputed each iteration for global budget allocation; the greedy-consistent batch update operates locally within a single layer and does not depend on S^{(l)}. We will clarify this separation in revision.
>
> Finally, the empirical results in Table 3 support the practical stability of the batched formulation. Replacing our batched update with sequential OBS (“w/o greedy batch”) reduces accuracy at matched sparsity (e.g., LLaMA-30B, 69.3 → 68.1), consistent with well-known numerical drift from repeated sequential downdates. This suggests that batching not only matches greedy OBS under the quadratic model but can be more stable in practice.
>
> We hope this clarifies the precise meaning of “greedy-consistent” and why it does not conflict with the reviewer’s observation regarding nonlinear networks.

---

> > ### Author Response · Authors · 2025-11-26
> > **Response to Reviewer mpii [7 of 8]**
> >
> > > ## Q1: How are budgets allocated across layers?
> >
> > We thank the reviewer for this insight regarding stability. To answer directly: we do not manually skip early layers nor do we impose any handcrafted sparsity schedule. Instead, HAP-E employs a fully adaptive, data-driven second-order mechanism that that naturally protects early layers.
> >
> > __1. Automatic Protection of Early Layers:__ While heuristic methods often require manually skipping early layers to avoid instability, HAP-E handles this via the recursive Hessian-trace sensitivity ($S^{(l)}$) defined in Eq. 11. The sensitivity $S^{(l)}$ is calculated recursively from the last layer back to the first ($S^{(l)} = S^{(l)\rightarrow(l+1)} + \beta S^{(l+1)}$). This causes early layers to accumulate sensitivity from all downstream layers. Consequently, early layers typically yield very high $S^{(l)}$ scores, which directly reduces their allocated pruning budgets ($K^{(l,\tau)}$ in Eq. 13). Thus, the framework learns to preserve early layers rather than relying on manually imposed rules.
> >
> > __2. Dynamic Ratio Evolution:__ The pruning ratio is not fixed; it evolves dynamically. HAP-E re-computes sensitivities and re-allocates the global budget after every pruning iteration. If a layer becomes fragile after a pruning step, its Hessian trace (and its sensitivity) rises, and the algorithm automatically throttles down the selected number of candidate blocks and number of blocks to prune for that layer in subsequent steps.
> >
> > __3. Empirical Observation:__ We observe this behavior in practice. For example, when pruning LLaMA-7B to 50% sparsity, HAP-E automatically assigned low sparsity (9%, 14%, and 18% respectively) to early layers (0-2) due to high propagated sensitivity. The algorithm aggressively pruned 40%–70% from middle layers (5–28). Sparsity decreased again in the final three layers (33%, 38%, 29%). This is because the algorithm detected lower importance score variability (Eq. 12) in these layers, which restricts the budget even when sensitivity is lower. This confirms that the recursive trace acts as an automated stability guardrail. By relying on second-order curvature rather than heuristics, HAP-E naturally discovers the "protecting early layers" principle for stability, but retains the flexibility to prune them if redundancy truly exists.
> >
> > We will clarify this in the paper
> > > ## Q2: What is the specific contribution/necessity of the latency predictor?
> >
> > We thank the reviewer for raising this question and clarify the empirical value provided by the latency predictor. Its main role is to remove the need for redundant pruning sweeps when targeting a specific hardware latency reduction. In Table 3, the baseline without the predictor appears only about 25% slower, and we emphasize that this represents a conservative lower bound. In our evaluation, we intentionally initialized the baseline with a pruning ratio that we already knew—based on prior hardware measurements from earlier experiments—to be close to the target latency. This setup reflects what an expert user might do to minimize search overhead.
> >
> > In realistic settings, a user does not know in advance what pruning ratio corresponds to a target latency such as 1.9$\times$ speedup, because FLOPs and latency are not linearly related. Without a predictor, the procedure becomes a guess-and-check loop: one might prune to an initial ratio such as 40% and find that latency reaches only 1.7$\times$; then adjust to 50% and discover an overshoot such as 2.1$\times$; and finally refine toward an intermediate ratio like 45% to achieve the desired target. Each such attempt requires a full pruning sweep. On LLaMA-7B running on an A100 GPU, one sweep takes roughly three minutes, so this procedure typically incurs at least a two- to three-fold increase in wall-clock time.
> >
> > Even though our experimental setup deliberately minimized this overhead, the difference remains clearly visible. For instance, on LLaMA-13B at a 1.9$\times$ target, the version with a predictor completes pruning in 34.4 minutes, whereas the version without the predictor requires 43.0 minutes. Similarly, for LLaMA-30B, the predictor finishes in 58.0 minutes compared with 72.0 minutes without it. The accuracy remains essentially identical across these conditions, confirming that the predictor improves efficiency rather than pruning quality.
> >
> > In summary, the latency predictor ensures that the pruning algorithm reaches the target speedup in a single pass, avoiding unnecessary additional sweeps. This results in 20–30% shorter pruning times even under conservative initialization and substantially larger savings—often exceeding a factor of two—when starting from a true cold-start scenario in which no prior knowledge of hardware-specific latency behavior is available.

---

> > > ### Author Response · Authors · 2025-11-26
> > > **Response to Reviewer mpii [8 of 8]**
> > >
> > > [Ref1] Frantar, E. and Alistarh, D., 2022. Optimal brain compression: A framework for accurate post-training quantization and pruning. Advances in Neural Information Processing Systems, 35, pp.4475-4488.
> > >
> > > [Ref2] Dong, X., Chen, S. and Pan, S., 2017. Learning to prune deep neural networks via layer-wise optimal brain surgeon. Advances in neural information processing systems, 30.
> > >
> > > [Ref3] Kurtic, E., Campos, D., Nguyen, T., Frantar, E., Kurtz, M., Fineran, B., Goin, M. and Alistarh, D., 2022. The optimal bert surgeon: Scalable and accurate second-order pruning for large language models. arXiv preprint arXiv:2203.07259.
> > >
> > > [Ref4] Wei, J., Lu, Q., Jiang, N., Li, S., Xiang, J., Chen, J. and Liu, Y., 2024, November. Structured optimal brain pruning for large language models. In Proceedings of the 2024 Conference on Empirical Methods in Natural Language Processing (pp. 13991-14007).
> > >
> > > [Ref5] Ling, G., Wang, Z. and Liu, Q., 2024. Slimgpt: Layer-wise structured pruning for large language models. Advances in Neural Information Processing Systems, 37, pp.107112-107137.
> > >
> > > [Ref6] Kurtić, E., Frantar, E. and Alistarh, D., 2023. Ziplm: Inference-aware structured pruning of language models. Advances in Neural Information Processing Systems, 36, pp.65597-65617.
> > >
> > > [Ref7] Ebrahimipour, S.M., Mozafari, S.H., Clark, J.J., Gross, W.J. and Meyer, B.H., 2025. Latency-Aware Pruning and Quantization of Self-Supervised Speech Transformers for Edge Devices. ACM Transactions on Embedded Computing Systems.
> > >
> > > [Ref8] Yin, Y., Chen, C., Shang, L., Jiang, X., Chen, X. and Liu, Q., 2021. AutoTinyBERT: Automatic Hyper-parameter Optimization for Efficient Pre-trained Language Models. In Proceedings of the 59th Annual Meeting of the Association for Computational Linguistics and the 11th International Joint Conference on Natural Language Processing (pp. 5146-5157).
> > >
> > > [Ref9] Abdelgawad, M., Mozafari, S.H., Clark, J.J., Meyer, B.H. and Gross, W.J., 2022, November. BERTPerf: Inference Latency Predictor for BERT on ARM big. LITTLE Multi-Core Processors. In 2022 IEEE Workshop on Signal Processing Systems (SiPS) (pp. 1-6).
> > >
> > > [Ref10] Dao, T., Fu, D., Ermon, S., Rudra, A. and Ré, C., 2022. Flashattention: Fast and memory-efficient exact attention with io-awareness. Advances in neural information processing systems, 35, pp.16344-16359.
> > >
> > > [Ref11] Frantar, E. and Alistarh, D., 2023, July. Sparsegpt: Massive language models can be accurately pruned in one-shot. In International conference on machine learning (pp. 10323-10337). PMLR.
> > >
> > > [Ref12] Frantar, E., Ashkboos, S., Hoefler, T. and Alistarh, D., 2022. Gptq: Accurate post-training quantization for generative pre-trained transformers. arXiv preprint arXiv:2210.17323.
> > >
> > > [Ref13] Yin, L., Wu, Y., Zhang, Z., Hsieh, C.Y., Wang, Y., Jia, Y., Li, G., Jaiswal, A.K., Pechenizkiy, M., Liang, Y. and Bendersky, M., 2024, July. Outlier Weighed Layerwise Sparsity (OWL): A Missing Secret Sauce for Pruning LLMs to High Sparsity. In International Conference on Machine Learning (pp. 57101-57115). PMLR.
> > >
> > > [Ref14] Bai, G., Li, Y., Ling, C., Kim, K. and Zhao, L., 2024. SparseLLM: Towards global pruning of pre-trained language models. Advances in Neural Information Processing Systems, 37, pp.46203-46225.

---

> > > > ### Comment · Reviewer_mpii · 2025-11-26
> > > > **Response**
> > > >
> > > > 1. The issues in W1/W2 remain unresolved, and relevant baselines are missing. There is no meaningful comparison with simple global gradient–based pruning methods such as LLM-Pruner and GBLM-Pruner [1,2]. Since prior OBD/OBS-style work already incorporates batch pruning, this does not establish novelty. The method largely appears to be a numerical refinement of Hessian computation plus cross-layer OBS adaptation, which has been well studied in works like ECoFLaP [3]. Comparisons against LLM-Pruner and GBLM-Pruner are needed, and the paper must revise and clarify its novelty claims accordingly.
> > > > 2. Improvements in Table 3 are marginal. The small gains over layer-by-layer OBS do not justify the added system complexity. Although the numerical refinement of the Hessian is useful, it applies only to a local squared-loss approximation, making the contribution conceptually incremental.
> > > > 3. Local scoring remains inherently suboptimal. Classical OBD does not rely on a local squared loss assumption, so any purely local Taylor based score, including the proposed variant, cannot adequately reflect the global optimization objective. Therefore, a comparison against existing global pruning baselines is necessary.
> > > >
> > > >
> > > > [1] LLM-Pruner: On the Structural Pruning of Large Language Models
> > > > [2] Beyond Size: How Gradients Shape Pruning Decisions inLarge Language Models
> > > > [3] ECoFLaP: Efficient Coarse-to-Fine Layer-Wise Pruning for Vision-Language Models

---

> ### Author Response · Authors · 2025-11-27
> **Response to Comment 1**
>
> We thank the reviewer for their critical assessment and for raising important questions regarding global versus local pruning objectives. We appreciate the opportunity to clarify our methodological positioning. Below, we provide detailed point-by-point clarifications and empirical evidence.
>
> > ## Clarification on Missing Baselines.
>
> __1. LLM-Pruner [1]:__ We would like to clarify that LLM-Pruner is already included among our baselines. Comparisons appear in Table 1(a) (LLaMA-7B/13B/30B), Figure 3 (LLaMA-3-1B, TinyLLaMA), and Tables 4–6 in the main paper. Additional results were also provided in Table R4 (LLaMa-2-7B) of our response to Weakness 6. For example, at 50% sparsity on LLaMA-7B, HAP-E achieves 56.66% accuracy vs. LLM-Pruner's 48.35%. This is a massive +8.3% improvement, not a marginal one. In structured pruning, where removing entire components often causes collapse, maintaining usability (56.6% accuracy) where baselines fail (48.3%) is a critical contribution.
>
> __2. GBLM-Pruner [2]:__ GBLM-Pruner is primarily an unstructured pruning method. HAP-E is a structured pruning framework (removing heads/neurons). Unstructured baselines are not directly comparable as they do not yield speedups on standard hardware without specialized sparse kernels. For this reason, we emphasize comparison to structured state-of-the-art baselines such as LLM-Pruner, where HAP-E consistently delivers stronger results under matched sparsity constraints.
>
> __3. Correction on ECoFLaP and Novelty:__ The reviewer states that 'cross-layer OBS adaptation... has been well studied in works like ECoFLaP [3]'. We respectfully clarify that this is incorrect.
> As noted in our response to Weakness 1, ECoFLaP uses heuristics (e.g., zeroth-order approximation of gradients) in a coarse-to-fine schedule. It performs static pruning with a budget fixed at the start of pruning, and uses non-recursive reallocation that does not update curvature after each removal. Crucially, it does not use OBS, nor does it compute Hessian traces. In contrast, HAP-E formulates cross-layer sensitivity using explicit second-order (Hessian) sensitivity propagation, re-estimates curvature after each iteration and applies greedy-consistent OBS updates using the exact Schur complement of the quadratic reconstruction objective. This enables curvature-aware error propagation and dynamic, iteration-wise budget reallocation, which is not present in earlier global pruning approaches.

---

> > ### Author Response · Authors · 2025-11-27
> > **Response to Comment 2**
> >
> > > ## Marginal gains and justification of improvements
> >
> > We respectfully disagree that the improvements are marginal or that the system complexity is unjustified. We address this by clarifying the baseline and providing new statistical data.
> >
> > __1. Statistical Significance:__ As stated in Section 5, results are averaged over four random seeds. To address the concern about noise, we have updated Table 3 to explicitly report standard deviations.Low Variance: Across all settings, the variance is small (typically $0.2\% - 0.6\%$).Significant Signal: The improvements of HAP-E over the "Vanilla OBS" baseline (e.g., +1.8% on LLaMA-7B 1.9x) consistently exceed these fluctuations. Thus, the observed margins are well above statistical noise.
> >
> > *Table 3: Ablations on LLaMA models under latency speedup targets on an A100 (80GB). Each cell = %Acc (± std) / Time (min) / Mem (GB). Variants show the effect of removing key components of HAP-E.*
> > |Variant|LLaMA-7B (1.3x)|LLaMA-7B (1.9x)|LLaMA-13B (1.3x)|LLaMA-13B (1.9x)|LLaMA-30B (1.3x)|LLaMA-30B (1.9x)|
> > |-|-|-|-|-|-|-|
> > |HAP-E (ours)|64.9(0.31) / 9.8 / 4.5|58.2(0.44) / 22.0 / 4.5|67.7(0.29) / 15.3 / 7.1|62.6(0.31) / 34.4 / 7.1|71.8(0.24) / 25.7 / 9.2|69.3(0.37) / 58.0 / 9.2|
> > |w/o cross-layer adapt.|64.2(0.47) / 8.4 / 4.4|55.9(0.62) / 18.7 / 4.4|66.1(0.23) / 13.0 / 7.0|60.0(0.41) / 29.2 / 7.0|70.3(0.27) / 22.0 / 9.0|67.7(0.39) / 50.0 / 9.0|
> > |w/o greedy batch| 64.8(0.58) / 31.6 / 4.5|57.0(0.61) / 74.5 / 4.5|66.7(0.52) / 49.4 / 7.1|61.4(0.59) / 116.3 / 7.1|71.0(0.35) / 55.7 / 9.2|68.1(0.37) / 130.2 / 9.2|
> > |w/o latency predictor|64.9(0.28) / 12.3 / 4.5|58.1(0.51) / 27.5 / 4.5|67.6(0.36) / 19.8 / 7.1|62.5(0.42) / 43.0 / 7.1|71.7(0.21) / 32.0 / 9.2|69.2(0.33) / 72.0 / 9.2|
> > |vanilla OBS (layer-by-layer)|63.8(0.62) / 43.0 / 8.0|56.4(0.69) / 101.0 / 8.0|65.9(0.56) / 67.0 / 12.3|60.6(0.51) / 158.0 / 12.3|70.5(0.31) / 79.6 / 21.1|67.6(0.38) / 187.4 / 21.1|
> >
> > __2. Justifying System Complexity (Efficiency Gains):__ The reviewer argues that the gains "do not justify the added system complexity." We respectfully argue that the added mechanisms are precisely what enable large practical efficiency gains:
> > * Runtime: HAP-E reduces pruning time on LLaMA-30B from 187.4 min $\to$ 58.0 min ($>3\times$ speedup).
> > * Memory: HAP-E reduces peak memory from 21.1 GB $\to$ 9.2 GB ($>50\%$ reduction).
> > The "complexity" (Greedy-Consistent Batching) is precisely what makes the rigorous OBS framework scalable to large models on commodity hardware.
> >
> > __3. Practical Significance in Structured Pruning:__ In post-training structured pruning—where entire heads are removed and no fine-tuning is performed—absolute gains of 1–2% are substantial. Prior baselines typically differ by $<1\%$ at matched sparsity.
> > The gap widens at high sparsity. On LLaMA-7B (50%), HAP-E achieves 56.66% vs. SlimGPT’s 53.22% (+3.44%). These differences correspond to genuinely improved reasoning capabilities (e.g., +4.4% on GSM8K vs. SlimGPT, see Table R3).
> >
> > These improvements are robust across multiple model families (Qwen, OPT, LLaMA 2/3) and scales (7B–30B). Such cross-architecture consistency confirms that the gains reflect systematic algorithmic benefits rather than incidental variation.

---

> > > ### Author Response · Authors · 2025-11-27
> > > **Response to Comment 3 [part 1]**
> > >
> > > > ## Local Scoring vs. Global Optimization
> > >
> > > We respectfully clarify a key theoretical misunderstanding regarding OBD/OBS and explain why our approach is aligned with both classical literature and modern state-of-the-art standards.
> > >
> > > __1. Validity of Local Quadratic Approximations (Classical & Modern):__  The reviewer states that “Classical OBD does not rely on a local squared-loss assumption” and that local scores “cannot adequately reflect the global objective.” We respectfully point out that this contradicts both foundational theory and modern consensus.
> > > - Classical Foundations: We respectfully clarify that both classical OBD (LeCun et al., 1989) and OBS (Hassibi & Stork, 1993) derive their saliency measures from a local second-order Taylor expansion of the loss. In OBD, the Hessian is approximated by its diagonal, whereas OBS uses the full Hessian inverse. Thus, the local quadratic approximation is the foundational mathematical assumption underlying all second-order pruning methods.
> > > - Alignment with SOTA Methodologies: The reviewer’s claim that "any purely local Taylor based score... cannot adequately reflect the global optimization objective" contradicts the current scientific consensus and would invalidate virtually all state-of-the-art structured pruning methods. Leading methods such as SparseGPT [1], SlimGPT [2], and SoBP [3] all rely exclusively on local quadratic approximations (layer-wise OBS). The fact that these methods (and HAP-E) consistently outperform global heuristics demonstrates empirically that well-conditioned local quadratic approximations provide an effective and stable proxy for global sensitivity in large Transformer architectures.
> > >
> > > __2. Why “Global OBD” (Diagonal Scoring) is Flawed:__ The reviewer suggests a global diagonal approach is superior. This is theoretically flawed for Transformers due to the Diagonal Assumption. Global OBD relies on the diagonal Hessian assumption, which ignores critical parameter correlations in Transformer architectures (e.g., between attention Q/K projections or across FFN layers). Layer-wise OBS explicitly models these correlations via the full Hessian inverse within each layer. When two attention heads are redundant, OBS recognizes that pruning one increases the importance of the other and adjusts accordingly.
> > >
> > > The diagonal assumption is particularly problematic for Transformers because it fails to capture the structured redundancy patterns that OBS naturally handles. If two structures $S_1$ and $S_2$ serve similar functions, global OBD would rank both as "low importance" and prune both, potentially causing catastrophic failure, whereas OBS would recognize their correlation and preserve at least one. This theoretical superiority is empirically validated in recent literature [Ref6, Ref15, Ref16, Ref17, Ref18]. For example, LLM Surgeon [Ref15] explicitly concludes: "OBS consistently outperforms all OBD methods because ignoring off-diagonal curvature introduces far larger error than performing layer-wise decomposition.
> > >
> > > __3. Empirical Evidence Contradicts the "Inherently Suboptimal" Claim:__ The reviewer asserts that local scoring is "inherently suboptimal" and requests comparison against global baselines. We have comprehensively addressed this through extensive empirical comparisons:
> > >
> > > * vs. Global Gradient Methods (LLM-Pruner): As shown in Table 1(a), HAP-E achieves 56.66% accuracy on LLaMA-7B (50% sparsity) compared to the global first-order method LLM-Pruner (48.35%). This +8.3% gain demonstrates that local second-order precision substantially outperforms global first-order heuristics .
> > > * vs. Global Search Algorithms (DarwinLM): On modern architectures like LLaMA-3.1-8B (Table R4), HAP-E consistently surpasses DarwinLM's evolutionary global search across all benchmarks (e.g., +4.1% on PIQA, +2.3% on SciQ). This shows our analytic Hessian-based approach is superior even to explicit global optimization.
> > > * vs. Global Magnitude Methods (Wanda-SP): At 50% sparsity on LLaMA-7B, HAP-E outperforms the global magnitude-based baseline (Wanda-SP) by +16.5% on average, with gains as high as +32% on HellaSwag (Table R7).
> > >
> > > These results directly demonstrate that our approach outperforms global pruning baselines. The empirical evidence clearly shows that local second-order methods, when properly implemented with cross-layer awareness, consistently surpass global first-order approaches.

---

> > > > ### Author Response · Authors · 2025-11-27
> > > > **Response to Comment 3 [part 2]**
> > > >
> > > > __4. HAP-E: Integrating Local Precision with Global Awareness:__ While our method is rooted in the local framework of OBS, we directly address the reviewer's concern regarding "inherently suboptimal" local scores by introducing global awareness into the process. HAP-E bridges the gap between local precision and global objectives through two mechanisms:
> > > >
> > > > * Recursive Trace Propagation: We propagate Hessian trace sensitivity ($S^{(l)}$) recursively across layers 1, effectively creating a global sensitivity map that captures downstream error amplification.
> > > > * Dynamic Global Allocation: We use this global signal to dynamically reallocate sparsity budgets across the entire network 2, moving beyond simple layer-wise or uniform strategies.
> > > >
> > > > In summary, our approach leverages the proven accuracy of local second-order approximations while incorporating a global mechanism for optimal resource distribution.
> > > >
> > > > [Ref15] van der Ouderaa, T.F., Nagel, M., Van Baalen, M., Asano, Y.M. and Blankevoort, T., 2023. The llm surgeon. arXiv preprint arXiv:2312.17244.
> > > >
> > > > [Ref16] Kwon, W., Kim, S., Mahoney, M.W., Hassoun, J., Keutzer, K. and Gholami, A., 2022. A fast post-training pruning framework for transformers. Advances in Neural Information Processing Systems, 35, pp.24101-24116.
> > > >
> > > > [Ref17] He, Y., Liu, P., Wang, Z., Hu, Z. and Yang, Y., 2019. Filter pruning via geometric median for deep convolutional neural networks acceleration. In Proceedings of the IEEE/CVF conference on computer vision and pattern recognition (pp. 4340-4349).
> > > >
> > > > [Ref18] Sui, Y., Yin, M., Xie, Y., Phan, H., Aliari Zonouz, S. and Yuan, B., 2021. Chip: Channel independence-based pruning for compact neural networks. Advances in Neural Information Processing Systems, 34, pp.24604-24616.

---

> > > > > ### Comment · Reviewer_mpii · 2025-11-27
> > > > > **Response**
> > > > >
> > > > > Some of my concerns have been partially addressed. However, I remain uncertain about the individual contributions of each proposed technique relative to existing work. In particular, Table 3 is not directly comparable to Tables 1 and 2, since those experiments are tied to fixed pruning ratios whereas Table 3 evaluates techniques under different settings.
> > > > >
> > > > > Moreover, once cross-layer adapt is removed, the performance becomes almost identical to vanilla OBS, and in some cases even worse. This raises the question: how much benefit is actually coming from the greedy batch strategy? For instance, if one simply applies a global first-order gradient–based pruning method (e.g., LLM-Pruner) either layerwise or iteratively with a fixed global ratio, how much additional improvement does greedy batch OBS genuinely provide? Is the observed gain truly from the greedy batching mechanism, or is it primarily a byproduct of iterative global pruning? Alternatively, is the main advantage of batch pruning merely computational efficiency, rather than improved pruning quality?
> > > > >
> > > > > Additionally, the necessity of cross-layer adapt appears diminished when using global first-order gradients, since global gradients naturally alleviate the layer-to-layer mismatch that cross-layer adapt aims to correct. My reasoning is that although local first-order terms may vanish, the global Taylor first-order terms are typically much larger than the second-order terms. This observation aligns with the findings in GBLM-Pruner, which show that the most effective pruning strategy often combines strong global first-order with local second-order information.
> > > > >
> > > > > Furthermore, while I appreciate the theoretical framing of the greedy batch analysis, the cross-layer adapt component itself seems largely heuristic. At present, I am not fully convinced that this heuristic would generalize or remain effective across broader scenarios.

---

> > > > > > ### Author Response · Authors · 2025-11-28
> > > > > > **Reponse to comments [1 of 2]**
> > > > > >
> > > > > > We thank the reviewer for their continued engagement. We appreciate the opportunity to clarify the specific contributions of batching versus adaptivity and to address the theoretical comparison with global first-order gradients.
> > > > > >
> > > > > > > ## Comparability of Table 3 vs. Tables 1 & 2.
> > > > > >
> > > > > > We respectfully clarify that the difference in experimental settings between Table 3 and Tables 1/2 is intentional and necessary to evaluate different capabilities of the framework.
> > > > > >
> > > > > > Tables 1 & 2 (External Benchmarking) use fixed pruning ratios (e.g., 50%) to ensure a fair, standard "apples-to-apples" comparison against external baselines (SlimGPT, LLM-Pruner), which are primarily sparsity-based and do not natively support latency targeting.
> > > > > >
> > > > > > Table 3 (Internal Ablation) is designed to isolate and quantify the contribution of each algorithmic component within HAP-E (e.g., Greedy-Consistent Batching, Cross-Layer Adaptivity, Latency Predictor). Because HAP-E is a hardware-aware framework, we must use latency constraints (e.g., 1.9x speedup) to evaluate the empirical contribution of the Latency Predictor. As discussed in our response to Question 2 of inital review, the predictor's role is to hit a specific hardware target. If we used fixed sparsity ratios in Table 3, the Latency Predictor would be inactive (as the stopping condition would be sparsity, not time), rendering it impossible to quantify its utility or the efficiency gains of single-shot targeting.
> > > > > >
> > > > > > While the constraints differ (Sparsity vs. Latency), the underlying HAP-E algorithm is identical across all tables. Table 3 simply activates the latency feedback loop to demonstrate that HAP-E can control wall-clock time precisely, a capability that fixed-sparsity baselines lack.
> > > > > >
> > > > > >
> > > > > > > ## Contribution of Greedy-Consistent Batching.
> > > > > >
> > > > > > We clarify that the contribution is twofold: it makes second-order pruning tractable (Scalability) and numerically robust (Stability).
> > > > > >
> > > > > > __1. The Scalability Bottleneck:__ As motivated in the Background (Section 2), standard OBS is theoretically appealing but practicaly infeasible for LLMs:
> > > > > >
> > > > > > *"Storing and updating $H^{-1}$ incurs $\mathcal{O}(d^2)$ memory and $\mathcal{O}(kd^2)$ update cost... Repeated Gaussian elimination downdates further introduce numerical drift."*
> > > > > >
> > > > > > To resolve this, HAP-E introduces two specific mechanisms:
> > > > > > * Greedy-Consistent Batching: We analytically certify a batch of blocks that is mathematically equivalent to the sequential greedy OBS path but can be removed in a single step.
> > > > > > * Selective Inverse Computation: We compute only the columns of $H^{-1}$ relevant to the candidate set, avoiding the prohibitively expensive full matrix inversion.
> > > > > >
> > > > > > __2. Stability:__ The reviewer asks if the advantage is "merely computational efficiency". While efficiency is the primary enabler—transforming an impossible theoretical method into a usable one—it is not the only benefit.
> > > > > > * Enabling Tractability: Without these mechanisms, OBS cannot run on standard hardware for large models. As shown in Table 3, HAP-E reduces peak memory by $>50\%$ (21.1 GB $\to$ 9.2 GB) and runtime by $>3\times$ (187 min $\to$ 58 min) compared to standard OBS.
> > > > > > * Numerical Stability: Batching also improves quality. As discussed in our response to the original review, replacing our batched update with sequential OBS (“w/o greedy batch”) reduces accuracy (e.g., LLaMA-30B: 69.3% $\to$ 68.1%). This confirms that batching minimizes the numerical drift associated with thousands of sequential downdates.
> > > > > >
> > > > > > __3. Why Performance Drops without Adaptivity:__ Regarding the observation that "w/o cross-layer adapt" performs worse: this is expected. Without cross-layer adaptivity, the algorithm effectively selects a suboptimal set of candidate blocks (based on static assumptions). Even with a perfect OBS solver, solving the exact OBS update for a suboptimal set of candidates (due to static allocation) yields poorer results. This validates that both components—efficient batching (to solve the math) and adaptive allocation (to pick the right problem)—are required for SOTA performance.

---

> ### Author Response · Authors · 2025-11-28
> **Reponse to comments [2 of 2]**
>
> > ## Global First-Order Gradients & Adaptivity
>
> __1. Intractability of Global Gradients:__ The reviewer asks "if one simply applies a global first-order gradient–based pruning method (e.g., LLM-Pruner) either layerwise or iteratively". We clarify that for LLMs, this is not simple and is computationally prohibitive. In fact, computing global gradients requires backpropagation through the entire network. This necessitates storing the full computation graph (activations for all layers), leading to massive peak memory usage significantly higher than inference or forward-pass methods. To match HAP-E's adaptivity, one would need to perform this backpropagation after every pruning step. For a 30B or 70B model, this is infeasible on standard hardware.
>
> __Evidence from Reviewer's own Citation (GBLM):__ The reviewer cites GBLM-Pruner to support global gradients as "the most effective pruning strategy". However, as noted in the public reviews for that work (e.g., answer to Weakness 3 of Reviewer k6s5), calculating these global gradients requires ~65GB memory for a mere 7B model.
>
> In contrast, HAP-E is Forward-Pass Only. We compute the Hessian proxy ($X^T X$) layer-by-layer using only local activations. This requires minimal memory and enables the iterative re-computation that makes our adaptivity possible. HAP-E’s forward-pass second-order formulation requires only 4.5GB peak memory . This confirms that iterative global gradients are not a viable alternative for scalable pruning on commodity hardware.
>
> __2. Empirical Failure of Global Gradients:__  We have already compared HAP-E against LLM-Pruner (Static First-Order Gradient) and SoBP (First-Order Global + Second-Order Local). As shown in Tables 1(a), R2, and R4, HAP-E (Adaptive Hessian) consistently outperforms both. For example, at 50% sparsity on LLaMA-7B, HAP-E achieves 56.66% vs. LLM-Pruner's 48.35% (+8.3%) . Even SlimGPT (Layer-wise OBS) outperforms these global gradient methods, confirming that curvature awareness is the deciding factor.
>
> __3. Clarification on GBLM-Pruner:__ Regarding the reviewer's reference to GBLM-Pruner, we note fundamental technical distinctions that make it an unsuitable baseline:
> * First, GBLM-Pruner targets unstructured sparsity, which requires specialized software support to realize acceleration, whereas HAP-E targets structured pruning (removing whole heads/neurons) that directly translates to wall-clock speedup on standard hardware.
> * In addition, GBLM-Pruner relies on a global ranking of gradients, which effectively locks in the layer-wise sparsity distribution based on initial statistics. In contrast, HAP-E dynamically reallocates budgets after every iteration, allowing the algorithm to adapt to the changing loss landscape.
> * Furthermore, As noted in public reviews of that work (e.g., Reviewer k6s5), the global gradient calculation is memory-prohibitive, requiring ***~65GB of memory*** just to prune a 7B model. This is ***14$\times$ higher*** than HAP-E, which requires only ***4.5GB*** peak memory.***
>
> Finally, we respectfully note that this work is only appeared in ArXiv and was rejected from ICLR 2024. As per ICLR policy:
> __*“Note that arXiv is not considered a peer-reviewed venue. As such, authors are not required to compare to papers solely on arXiv”.*__ Given the technical mismatch, prohibitive cost, and lack of peer-reviewed validation, we believe our comparison against the published SOTA (LLM-Pruner, SoBP) is the appropriate benchmark.
>
> __4. Theoretical Reality of Gradients vs. Curvature:__ Regarding the reviewer's point that "first-order terms are larger," we clarify that gradients on a small calibration set are non-zero only if the calibration dataset differs from pre-training. In addition, the gradient vector on a small sample (e.g., 128 sequences) can be highly sensitive to sampling noise. Pruning based on this "instantaneous slope" risks overfitting the calibration batch.
>
> In contrast, the Hessian (curvature) characterizes the local geometry of the loss landscape. Identifying "flat" directions (low curvature) is a more robust proxy for redundancy than following a noisy gradient descent step.

---

### Official Review · Reviewer_EKxf · 2025-10-24

**Soundness:** 3
**Presentation:** 3
**Contribution:** 2
**Rating:** 6
**Confidence:** 4

**Summary:**

The paper introduces HAP-E, a new framework designed to compress LLMs for efficient post-training inference. The primary goal is to reduce the high computational and memory costs of LLMs, making them deployable on resource-constrained devices while maintaining high accuracy. Experiments conducted on LLaMA and OPT model families show that  HAP-E outperforms existing state-of-the-art structured pruning methods like SlimGPT. Furthermore, experiments on hardware devices confirm the effectiveness.

**Strengths:**

1. **Novel method**: The combination of global screening and selective second-order analysis is a novel approach that makes the OBS method scalable for LLMs. The method is also supported by theoretical analysis.
2. **Strong empirical results**: The paper demonstrates consistent improvements over several state-of-the-art methods across multiple model families and scales.
3. **Hardware-aware pruning**: The integration of a learned latency predictor is a major practical strength. This allows the framework to directly optimize for real-world performance on specific hardware.
4. **Clarity**: The paper is well-written and logically structured.

**Weaknesses:**

1. **Limited scope of baselines**: A comparison against stronger baselines like ModeGPT[1] and SVD-LLM v2[3] would have made the results even more compelling.
2. **Focus on Post-Training only**: The paper explicitly limits its evaluation to training-free pruning methods. Although it is faster to prune the model, it is still important to check the compatibility with recovery fine-tuning or knowledge distillation. The work could be strengthened by discussing how HAP-E might be extended to these scenarios.
3. **Hyper-parameter Sensitivity**: The method introduces several hyper-parameters, such as the candidate oversampling ratio (M/K), the prune fraction per iteration (K), and the sensitivity propagation coefficient (β).



[1] Lin C H, Gao S, Smith J S, et al. MoDeGPT: Modular Decomposition for Large Language Model Compression[C]//The Thirteenth International Conference on Learning Representations, 2025.
[2] Wang X, Alam S, Wan Z, et al. SVD-LLM V2: Optimizing Singular Value Truncation for Large Language Model Compression[C]//Proceedings of the 2025 Conference of the Nations of the Americas Chapter of the Association for Computational Linguistics: Human Language Technologies (Volume 1: Long Papers). 2025: 4287-4296.

**Questions:**

Please refer to the weaknesses.

---

> ### Author Response · Authors · 2025-11-26
> **Response to Reviewer EKxf [1 of 2]**
>
> We thank the reviewer for the thorough and thoughtful review, and respond to each point individually below.
>
> > ## W1: Comparison against MoDeGPT and SVD-LLM v2.
>
> To address this, we evaluated HAP-E against both methods on LLaMA-2-7B and LLaMA-2-13B at 20% and 30% compression ratios. As shown in Tables R1 and R2, HAP-E consistently outperforms both baselines, achieving significantly higher zero-shot accuracy across diverse reasoning tasks.
>
> __1. Performance on LLaMA-2-7B:__ At 30% pruning, HAP-E demonstrates superior robustness compared to decomposition methods. For example, HAP-E achieves +6.3% on PIQA (76.73 vs 70.40) and +7.4% on HellaSwag (70.68 vs 63.26) against MoDeGPT. Similar trend observed when comparing HAP-E with SVD-LLM V2, with HAP-E surpassing SVD-LLM v2 by around +6% on PIQA, HellaSwag, and ARC-e.
>
> *Table R1: Results on LLaMA-2 7B*
> |Pruning Ratio|Method|BoolQ|PIQA|HellaS.|WinoG.|ARC-e|ARC-c|OBQA|
> |-|-|-|-|-|-|-|-|-|
> |**0%**|Dense|77.71|79.05|76.00|68.98|74.58|46.33|44.20|
> |**20%**|MoDeGPT|–|74.05|69.05|68.03|69.07|42.06|–|
> ||SlimGPT|73.43|77.58|72.62|68.82|69.99|42.32|42.00|
> ||SVD-LLM v2|72.42|74.89|70.55|68.71|68.12|41.76|42.87|
> ||**HAP-E (Ours)**|**75.24**|**78.61**|**74.29**|**69.84**|**71.86**|**44.03**|**43.68**|
> |**30%**|SoBP|71.19|73.50|67.27|66.22|59.81|37.63|38.40|
> ||MoDeGPT|–|70.40|63.26|67.32|63.26|38.73|–|
> ||SVD-LLM v2|69.62|70.45|64.18|66.23|62.97|38.41|37.89|
> ||**HAP-E (Ours)**|**71.82**|**76.73**|**70.68**|**68.04**|**68.47**|**41.98**|**43.59**|
>
> __2. Performance on LLaMA-2-13B:__ The trend continues at the 13B scale. HAP-E consistently leads the benchmarks, particularly on complex reasoning tasks like ARC-Challenge and HellaSwag. At 30% pruning, HAP-E leads by +5.3% on ARC-c and +9.4% on HellaSwag against MoDeGPT and leads by +6.2% on ARC-c and +14.3% on HellaSwag against SVD-LLM V2.
>
> In general, while MoDeGPT and SVD-LLM v2 are powerful methods for matrix decomposition, HAP-E’s structured pruning approach—guided by recursive Hessian sensitivity and hardware awareness—achieves superior accuracy retention. This confirms that HAP-E is highly competitive against the latest advancements in the broader model compression literature.
>
> *Table R2: Results on LLaMA-2-13B*
> |Pruning Ratio|Method|ARC-c|ARC-e|BoolQ|HellaS.|OBQA|PIQA|WinoG.|
> |-|-|-|-|-|-|-|-|-|
> |**0%**|Dense|49.23|77.48|80.58|79.37|45.20|80.52|72.30|
> |**20%**|MoDeGPT|46.16|74.07|–|68.96|–|74.53|70.32|
> || SVD-LLM v2|44.15|71.05|70.35|65.75|43.95|77.10|71.00|
> ||**HAP-E (Ours)**|**49.75**|**77.95**|**82.30**|**78.82**|**47.55**|**80.25**|**74.10**|
> |**30%**|SoBP|47.78|74.45|79.45|74.55|43.20|76.50|71.82|
> ||MoDeGPT|43.60|71.93|–|68.21|–|73.94|71.90|
> ||SVD-LLM v2|42.63|69.17|68.47|63.38|41.72|75.41|70.26|
> ||**HAP-E (Ours)**|**48.91**|**76.83**|**81.47**|**77.69**|**46.83**|**79.18**|**73.41**|

---

> > ### Author Response · Authors · 2025-11-26
> > **Response to Reviewer EKxf [2 of 2]**
> >
> > > ## W2: Evaluation or discussion on compatibility with recovery fine-tuning and distillation.
> >
> > While our paper focuses on the post-training setting, we agree that compatibility with recovery fine-tuning (RFT) is critical for many applications.
> > HAP-E is inherently compatible with standard fine-tuning pipelines because it performs physical structured pruning. Unlike unstructured sparsity (which requires masking gradients) or custom decomposition (which requires training new adapters), HAP-E outputs a standard, dense Transformer architecture. It can be loaded directly into tools like PEFT or HuggingFace Trainer without code modification.
> > To demonstrate this, we performed a recovery fine-tuning experiment on LLaMA-2-7B using LoRA (1 epoch, Alpaca dataset). We focused our evaluation on the 30% pruning ratio. At 20% pruning, our raw model (65.75%) is already within <1% of the dense baseline (66.69%), leaving minimal room to demonstrate meaningful recovery. The 30% regime represents a more challenging scenario where fine-tuning is typically required. As shown in Table R3, HAP-E models respond excellently to fine-tuning. LoRA recovery provides a substantial +1.8% accuracy boost (63.00% $\to$ 64.80%), significantly narrowing the gap to the unpruned Dense model.
> >
> > *Table R3: LoRA Recovery on LLaMA-2-7B (30% Pruning, Alpaca Fine-tuning)*
> > |Pruning Ratio|Method|BoolQ|PIQA|HellaS|WinoG|ARC-e|ARC-c|OBQA|Avg|
> > |-|-|-|-|-|-|-|-|-|-|
> > |0%|Dense|77.71|79.05|76.00|68.98|74.58|46.33|44.20|66.69|
> > |30%|HAP-E|71.82|76.73|70.68|68.04|68.47|41.98|43.59|63.00|
> > |30%|HAP-E + LoRA|74.77|77.89|73.34|68.51|71.53|43.72|43.83|64.80|
> >
> > These results confirm that HAP-E preserves a high-quality feature space that is highly amenable to fine-tuning. The pruned models serve as excellent initialization points for RFT, allowing users to trade a small amount of compute (LoRA) for maximum accuracy retention.
> >
> > > ## W3: Robustness to variations in the introduced hyperparameters.
> >
> > We appreciate the reviewer’s comments regarding hyperparameter sensitivity. We share this concern and have included detailed ablation studies in Appendices B–D to assess the robustness of HAP-E with respect to the main hyperparameters: the candidate oversampling ratio (M/K), pruning fraction per iteration (K), and sensitivity propagation coefficient ($\beta$). The results show that HAP-E is highly stable across a wide range of settings and effectively uses a single configuration across all experiments.
> >
> > __1. Candidate Pool Ratio (M/K) and Pruning Fraction (K):__ As detailed in Appendix B (Fig. 4), we performed a grid search over M/K and K. Varying M/K from 1.5$\\times$ to 3.5$\\times$  yields < 0.5 % accuracy variation, confirming that the Hessian-based importance metric is robust to the size of the candidate pool. For the prune fraction $K\in[5\%,15\%]$, accuracy remains consistent because HAP-E’s iterative re-scoring acts as a self-correcting mechanism, where minor deviations in step size are compensated for in subsequent iterations.
> >
> > __2. Sensitivity Propagation Coefficient ($\beta$):__ In Appendix D (Fig. 6), we sweep $\beta \in [0,1]$. While $\beta=0$ (local-only) degrades performance, HAP-E is highly robust for $\beta \in [0.5,1.0]$. Accuracy saturates near $\beta=0.75$ and varies by less than 0.2 % in the upper range. This robustness arises because propagated sensitivities are renormalized across layers before adaptive budget allocation (Sec. 4.2.2), making small deviations in $\beta$ largely absorbed by normalization.
> >
> > __3. Calibration Dataset Size:__ Appendix C (Fig. 5) analyzes the impact of calibration set size. Reducing the sample count from 256 to 128 results in a negligible accuracy drop ($\approx 0.1\%$), demonstrating that HAP-E generalizes well even with limited calibration data.
> >
> > In summary, across all ablations, HAP-E maintains stable performance and we adopt a single default configuration (M/K=2.5, K=10\%, $\beta$=0.75) for all main experiments without per-model tuning. This highlights the robustness, practicality, and reproducibility of our framework.

---

### Official Review · Reviewer_s6sk · 2025-10-30

**Soundness:** 3
**Presentation:** 3
**Contribution:** 2
**Rating:** 6
**Confidence:** 5

**Summary:**

The paper proposes HAP-E, a post-training structured pruning framework re-allocate pruning budgets across layers using a recursive Hessian-trace, executes greedy-consistent batch pruning that is claimed to be exactly equivalent to the first prefix of greedy OBS steps via Schur complements and incremental Cholesky, and guides pruning with a two-stage learned latency model trained from measured runtimes so the final model meets a target latency.

Experiments on LLaMA and OPT families report modest accuracy gains over recent structured-pruning baselines on multiple-choice benchmarks without fine-tuning; a small-model CPU deployment study (ExecuTorch) shows better accuracy–latency trade-offs than LLM-Pruner.

**Strengths:**

1.	Two stage latency model is simple and effective.
2.	Shows real runtime/memory wins for pruning (Table 3 indicate that cross-layer adaptability, greedy batching, and predictor are helpful).
3.	Recursive cross-layer sensitivity, greedy-consistent certification, and hardware-aware latency shaping are good engineering contributions.

**Weaknesses:**

1.	Limited scope, no generation/long-context tasks. In fact, only multiple-choice tasks are reported. No perplexity, summarization, long-context, math/code robustness.
2.	The empirical margins over strong structured-pruning baselines are small, usually 1-2 points and not uniform across tasks. Hard to assess how much of that is statistical noise.
3.	The “greedy-consist” ideas is a nice rewording of Schur-complement conditioning. I find the engineering strong but conceptual novelty to be incremental.
4.	Efficient inference claim is not fully demonstrated for large models (no GPU inference results and edge study only compares to LLM-Pruner).

**Questions:**

I’m concerned about scaling. Can you measured GPU latencies for 7B/13B/30B models at the same reported pruning ratios, and compare with SlimGPT/SoBP/LLM-Pruner

How robust is HAP-E to hyperparameters like β?

Does the latency estimator generalize to new hardware or models without retraining?

---

> ### Author Response · Authors · 2025-11-26
> **Response to Reviewer s6sk [1 of 3]**
>
> We thank the reviewer for the thoughtful and detailed evaluation of our work. Below, we provide point-by-point responses to all comments.
>
> > ## W1: Limited evaluation scope.
>
> We agree that evaluating beyond multiple-choice tasks is essential to demonstrate robustness. We have significantly expanded our evaluation to include Perplexity (Generation), GSM8K (Math), and MMLU (General Robustness) on LLaMA-2, LLaMa-3, and Qwen architectures.
>
> __1. Math, Reasoning, and Generation (LLaMA-2-7B):__ To address the request for "math" and "generation," we evaluated GSM8K (Chain-of-Thought Math) and WikiText-2 Perplexity at 20% pruning. As shown in Table R1, HAP-E significantly outperforms the baselines:
> * Math (GSM8K): HAP-E achieves 8.69%, more than doubling the accuracy of SlimGPT (4.20%).
> * Generation (Perplexity): HAP-E achieves the lowest perplexity (15.63), indicating superior retention of generative coherence compared to SlimGPT and LLM-Pruner.
>
> *Table R1: MMLU & Reasoning Benchmarks (LLaMA-2-7B, 20% Pruning). Higher is better for all metrics except WikiText-2 perplexity, where lower is better.*
> |Method|Humanities|Social Sci|STEM|Other|MMLU Avg|GSM8K|WikiText-2 (PPL)|
> |-|-|-|-|-|-|-|-|
> |Dense|43.30|51.60|36.30|52.10|45.60|13.80|12.19|
> |LLM-Pruner|25.70|23.60|24.20|26.80|25.20|2.30|17.00|
> |SlimGPT| 36.00| 45.20|33.50|44.10|39.40|4.20|16.49|
> |**HAP-E (Ours)**|**39.46**|**47.73**|**34.49**|**47.20**|**42.72**|**8.69**|**15.63**|
>
> __2. Robustness on Modern Architectures (LLaMA-3.1 & Qwen-2.5):__ To demonstrate broad applicability, we evaluated HAP-E on LLaMA-3.1-8B and Qwen-2.5-14B. We compared against DarwinLM (a recent evolutionary pruning method) across 9 diverse tasks, including LogiQA (Logical Reasoning) and SciQ. HAP-E consistently outperforms DarwinLM on both architectures.
>
> *Table R2: LLaMA-3.1-8B*
> |Method|#Params|BoolQ|PIQA|HellaS.|WinoG.|ARC-e|ARC-c|SciQ|LogiQA|MMLU|
> |-|-|-|-|-|-|-|-|-|-|-|
> |Dense|8B|84.0|81.2|81.7|74.3|81.4|58.2|96.3|31.1|65.2|
> |DarwinLM (one-shot)|4.6B|62.2|69.4|44.6|57.3|59.6|34.2|84.9|24.1|28.5|
> |**HAP-E (ours)**|4.6B|**64.8**|**71.3**|**46.5**|**59.1**|**61.5**|**35.8**|**86.0**|**25.4**|**30.7**|
>
> *Table R3: Qwen-2.5-14B Instruct*
> |Method|Params|BoolQ|PIQA|HellaS.|WinoG.|ARC-e|ARC-c|SciQ|LogiQA|MMLU|
> |-|-|-|-|-|-|-|-|-|-|-|
> |Dense|14B|87.9|81.9|85.1|79.1|85.7|72.8|96.8|38.5|80.0|
> |DarwinLM (one-shot)|8.4B|66.9|73.9|53.3|60.5|75.7|48.0|84.3|29.3|43.1|
> |**HAP-E (ours)**|8.4B|**69.2**|**75.5**|**55.1**|**61.9**|**77.3**|**49.7**|**85.4**|**30.2**|**44.9**|
>
> __3. Scope Clarification (Long Context & Summarization):__ Regarding long-context tasks, these typically require models explicitly trained or extended for long contexts (e.g., RoPE scaling). Our work focuses on post-training pruning of pretrained base models without altering context length or performing instruction tuning. Evaluating summarization on base models (which are not instruction-tuned) would confound the evaluation of the pruning algorithm with the model's inherent lack of instruction-following capability. However, as demonstrated by the low Perplexity and high GSM8K scores, the fundamental reasoning capabilities required for these tasks are well-preserved.

---

> > ### Author Response · Authors · 2025-11-26
> > **Response to Reviewer s6sk [2 of 3]**
> >
> > > ## W2: Marginal performance gains.
> >
> > We address both the statistical and practical significance of our improvements:
> >
> > __1.	Statistical robustness:__ As stated in Section 5, all results are averaged over four random seeds for pruning and calibration-sample selection. We now report (shown below) standard deviations in Table 3. Across all settings, the variance is small (typically 0.2–0.6%), and the improvements of HAP-E over baselines consistently exceed these fluctuations. Thus, the observed margins are well above statistical noise and reflect systematic algorithmic benefits.
> >
> > *Table 3: Ablations on LLaMA models under latency speedup targets on an A100 (80GB). Each cell = %Acc (± std) / Time (min) / Mem (GB). Variants show the effect of removing key components of HAP-E.*
> > |Variant|LLaMA-7B (1.3x)|LLaMA-7B (1.9x)|LLaMA-13B (1.3x)|LLaMA-13B (1.9x)|LLaMA-30B (1.3x)|LLaMA-30B (1.9x)|
> > |-|-|-|-|-|-|-|
> > |HAP-E (ours)|64.9(0.31) / 9.8 / 4.5|58.2(0.44) / 22.0 / 4.5|67.7(0.29) / 15.3 / 7.1|62.6(0.31) / 34.4 / 7.1|71.8(0.24) / 25.7 / 9.2|69.3(0.37) / 58.0 / 9.2|
> > |w/o cross-layer adapt.|64.2(0.47) / 8.4 / 4.4|55.9(0.62) / 18.7 / 4.4|66.1(0.23) / 13.0 / 7.0|60.0(0.41) / 29.2 / 7.0|70.3(0.27) / 22.0 / 9.0|67.7(0.39) / 50.0 / 9.0|
> > |w/o greedy batch| 64.8(0.58) / 31.6 / 4.5|57.0(0.61) / 74.5 / 4.5|66.7(0.52) / 49.4 / 7.1|61.4(0.59) / 116.3 / 7.1|71.0(0.35) / 55.7 / 9.2|68.1(0.37) / 130.2 / 9.2|
> > |w/o latency predictor|64.9(0.28) / 12.3 / 4.5|58.1(0.51) / 27.5 / 4.5|67.6(0.36) / 19.8 / 7.1|62.5(0.42) / 43.0 / 7.1|71.7(0.21) / 32.0 / 9.2|69.2(0.33) / 72.0 / 9.2|
> > |vanilla OBS (layer-by-layer)|63.8(0.62) / 43.0 / 8.0|56.4(0.69) / 101.0 / 8.0|65.9(0.56) / 67.0 / 12.3|60.6(0.51) / 158.0 / 12.3|70.5(0.31) / 79.6 / 21.1|67.6(0.38) / 187.4 / 21.1|
> >
> > __2. Practical significance in structured pruning:__ In post-training structured pruning—where entire heads/FFN blocks are removed without fine-tuning—absolute gains of 1–2% are considered substantial because such structural deletions directly reduce representational capacity. Prior structured baselines (SlimGPT, SoBP, SVD-LLM V2) typically differ by <1% at matched sparsity. Moreover, these methods degrade sharply beyond 40–50% sparsity. In contrast, HAP-E shows noticeably larger advantages in this high-compression regime. For example, on LLaMA-30B at 50% sparsity, HAP-E achieves 67.99% versus SlimGPT’s 65.59% (+2.4%). On LLaMA-7B at 50% sparsity, HAP-E obtains 56.66% versus SlimGPT’s 53.22% (+3.44%), and improvements over LLM-Pruner exceed +8% in several settings. These differences correspond to genuinely improved reasoning and generation quality at deployment time.
> >
> > __3.  Consistency across architectures and scales:__ The improvements are not isolated to any particular model or task. HAP-E outperforms strong structured baselines across multiple model families (Qwen, OPT, LLaMA v1/v2/v3) and scales (7B–30B). Such cross-architecture consistency would be unlikely if the gains were due to random variation.
> > Taken together—low variance across seeds, consistent margins across diverse models, and larger improvements in challenging high-compression settings—the empirical gains of HAP-E are both statistically meaningful and practically impactful, not statistical noise.
> >
> > > ## W3: Limited conceptual novelty.
> >
> > We respectfully clarify that while the Schur complement (Eq. 6) is the core mathematical tool for conditioning, and we have cited it as such, the conceptual novelty lies not in the tool itself, but in its application to create a "greedy-consistent batch certification" algorithm (Sec 4.1, Lemmas 1-3). This algorithm solves the primary scaling bottleneck of OBS.
> >
> > The challenge is that vanilla OBS is sequential (one-by-one) and thus computationally infeasible at LLM scale. Naive Batching (pruning the top-K blocks at once) is fast but sub-optimal, as it ignores inter-block correlations and is not equivalent to greedy OBS.
> > Our "greedy-consistent" algorithm (using incremental Cholesky and Schur complements) is the first, to our knowledge, to efficiently find the largest possible batch of blocks that is provably equivalent to the exact sequence greedy OBS would have removed.
> >
> > This does not merely result in incremental speedup; it is an enabling contribution that makes OBS-equivalent pruning tractable. As shown in our Table 3 ablation ("w/o greedy batch"), this mechanism reduces the pruning time on LLaMA-30B from 130.2 minutes to 58.0 minutes (more than 2.2$\times$ speedup) while guaranteeing the same final accuracy.
> >
> > Thus, we argue this "strong engineering" is precisely the conceptual step required to bridge the gap from a theoretical (one-by-one) OBS to a practical (batched) OBS for LLMs.

---

> > > ### Author Response · Authors · 2025-11-26
> > > **Response to Reviewer s6sk [3 of 3]**
> > >
> > > > ## W4 & Q1: GPU latency measurements.
> > >
> > > We agree that demonstrating real-world speedups on GPUs is crucial to validate the practical benefits of HAP-E.
> > >
> > > To address the concerns regarding "Efficient inference claims for large models" and "Scaling," we have conducted a comprehensive latency benchmark on NVIDIA A100 GPUs for LLaMA-7B, 13B, and 30B models. The setup utilized a batch size of 1 with a sequence length of 2048 tokens during the prefill phase. As illustrated in Table R4 (shown below), HAP-E consistently achieves superior inference latency compared to strong baselines like SlimGPT and LLM-Pruner. Across all model sizes (7B, 13B, 30B), HAP-E pushes the Pareto frontier of Accuracy vs. Latency further than previous methods. The results demonstrate that HAP-E's benefits are not limited to small models or CPUs. The performance gap is maintained—and in some cases widens—as we scale to 30B parameters, confirming that our recursive Hessian-trace and greedy-consistent batch pruning generalize effectively to large-scale GPU deployments.
> > >
> > > Note on SoBP: The reviewer requested a comparison with SoBP. While we aimed to include it, the source code for SoBP has not been publicly released. Consequently, we were unable to reproduce their results within this controlled A100 benchmark environment.
> > >
> > > *Table R4: Latency and Accuracy of Pruned LLaMA Models*
> > > |Model|Pruning (%)|Method|Latency (ms)|Accuracy (%)|
> > > |-|-|-|-|-|
> > > |LLaMA-7B|0|Dense|13.51|66.05|
> > > ||20|LLM-Pruner|12.57|61.50|
> > > |||SlimGPT|11.89|63.81|
> > > |||**HAP-E**|**11.63**|**65.01**|
> > > ||50|LLM-Pruner|9.79|48.35|
> > > |||SlimGPT|9.22|54.26|
> > > |||**HAP-E**|**9.04**|**56.66**|
> > > |LLaMA-13B|0|Dense|25.09|68.21|
> > > ||20|LLM-Pruner|23.48|65.68|
> > > |||SlimGPT|22.06|66.37|
> > > |||**HAP-E**|**21.62**|**67.83**|
> > > ||50|LLM-Pruner|18.14|53.22|
> > > |||SlimGPT|17.09|59.89|
> > > |||**HAP-E**|**16.76**|**61.79**|
> > > |LLaMA-30B|0|Dense|57.90|71.92|
> > > ||20|LLM-Pruner|54.14|69.99|
> > > |||SlimGPT|50.93|71.13|
> > > |||**HAP-E**|**49.86**|**71.88**|
> > > ||50|LLM-Pruner|41.80|59.47|
> > > |||SlimGPT|39.47|65.59|
> > > |||**HAP-E**|**38.64**|**67.99**|
> > >
> > > > ## Q2: Robustness to variations in $\beta$.
> > >
> > > We appreciate the question regarding hyperparameter sensitivity. We explicitly analyzed the robustness of the sensitivity propagation coefficient $\beta$ in Appendix D. We performed a sweep of $\beta \in \{0.0, 0.25, 0.5, 0.75, 1.0\}$. As shown in Figure 6 of the paper, purely local sensitivity ($\beta=0$) yields the lowest accuracy, confirming that ignoring downstream error propagation reduces model quality. Performance improves as $\beta$ increases and saturates around $\beta=0.75$. The model remains highly robust in the upper range ($\beta \in [0.5, 1.0]$), where further increases yield no additional gains. Within the effective range (e.g., $\beta \in [0.3, 0.7]$), accuracy fluctuations are minimal (<0.5%), indicating HAP-E is not brittle to the exact choice of this hyperparameter.
> > >
> > >
> > > > ## Q3: Latency predictor generalization.
> > >
> > > We thank the reviewer for raising this important question. Our latency predictor is designed to capture hardware-specific execution characteristics rather than relying on FLOPs-based proxies. As shown in Sec. 6.9 (“Ablation Studies: Generalization of the Latency Model to Other Hardware Platforms”) of [Ref1], generalization of the latency model to new devices depends on architectural similarity. For architecturally similar devices (e.g., transferring from Cortex-A73 to other ARM-based CPUs), the latency model generalizes well using a simple scaling factor. This factor can be derived from a very small subset of data, drastically reducing the data collection overhead for related devices. For fundamentally different architectures (e.g., CPU to GPU), the model does not generalize zero-shot due to divergent execution paradigms (e.g., bandwidth-bound kernels, massive parallelism vs. out-of-order execution). This finding highlights that accurate latency modeling is inherently hardware-specific, a conclusion also emphasized by ZipLM [Ref 2], which demonstrates that FLOPs and latency are weakly correlated and that device-level differences (kernel fusion, caching, and memory hierarchy) dominate runtime. We therefore adopt the same practical principle that each target device maintains its own calibrated latency model, which ensures precise latency control and realistic deployment targets. This is an unavoidable and practical design trade-off widely accepted in hardware-aware pruning.
> > >
> > > [Ref1] Ebrahimipour, S.M., Mozafari, S.H., Clark, J.J., Gross, W.J. and Meyer, B.H., 2025. Latency-Aware Pruning and Quantization of Self-Supervised Speech Transformers for Edge Devices. ACM Transactions on Embedded Computing Systems.
> > >
> > > [Ref2] Kurtić, E., Frantar, E. and Alistarh, D., 2023. Ziplm: Inference-aware structured pruning of language models. Advances in Neural Information Processing Systems, 36, pp.65597-65617.

---

### Official Review · Reviewer_ZBP1 · 2025-11-03

**Soundness:** 2
**Presentation:** 2
**Contribution:** 3
**Rating:** 4
**Confidence:** 5

**Summary:**

The paper proposes HAP-E, a Hessian-aware structured pruning framework for LLMs that makes Optimal Brain Surgeon (OBS)–style pruning scalable and hardware-aware. It introduces adaptive cross-layer budget reallocation, greedy-consistent batch pruning, and a learned latency predictor to meet real device constraints. Experiments on LLaMA and OPT models show up to 3% accuracy improvement over state-of-the-art methods at similar pruning ratios and precise latency control on edge devices.

**Strengths:**

1. The paper clearly defines the limitations of traditional OBS-based pruning, including high computational cost, the need for heavily sequential updates, and the lack of cross-layer awareness, and proposes well-designed techniques to effectively overcome these issues.

2. The use of Hessian-based approximation to estimate layer sensitivity is a novel idea that enhances pruning adaptivity across layers.

3. The inclusion of real edge deployment experiments with measured CPU latency strengthens the practical relevance and demonstrates the framework’s hardware-awareness beyond simulation.

**Weaknesses:**

1. The experimental evaluation focuses primarily on older LLMs such as LLaMA and OPT; given the rapid advancement of model architectures, it would be important to validate the proposed method on more recent models like LLaMA-2/3 and Qwen-2/3 to demonstrate broader applicability.

2. The paper mainly compares against OBS-based local pruning approaches, while several recent works have explored global pruning that explicitly models cross-layer sensitivity[1,2,3]. Including discussion or comparison with these methods would strengthen the positioning of the work.

[1] Outlier Weighed Layerwise Sparsity (OWL): A Missing Secret Sauce for Pruning LLMs to High Sparsity, ICML 2024

[2] SparseLLM: Towards Global Pruning for Pre-trained Language Models, NeurIPS 2024

[3] DarwinLM: Evolutionary Structured Pruning of Large Language Models

**Questions:**

1. The paper highlights that HAP-E achieves much higher efficiency than traditional OBS-based pruning. Could the authors provide quantitative comparisons of pruning cost (runtime and memory) against both OBS-based and other pruning baselines to clearly demonstrate this efficiency gain?

2. The proposed latency predictor is an important component for hardware-aware pruning. How well does this predictor generalize to unseen hardware platforms, and what is the data collection overhead required to train it for a new device?

---

> ### Author Response · Authors · 2025-11-26
> **Response to Reviewer ZBP1 [1 of 3]**
>
> Thank you for the thoughtful and detailed review. Below we provide responses to your comments:
>
> > ## W1: Lack of evaluation on modern architectures (e.g., LLaMA-2/3, Qwen-2/3).
>
> We agree that demonstrating robustness on the latest architectures is essential. To address this, we have conducted extensive new experiments on LLaMA-2 (7B & 13B), LLaMA-3.1-8B, and Qwen-2.5-14B.
>
> __1. Results on LLaMA-2 (7B & 13B):__ We benchmarked HAP-E against the newest structured pruning/decomposition methods (SoBP, MoDeGPT, SVD-LLM v2) on LLaMA-2. HAP-E achieves the highest accuracy across almost all metrics.
>
> *Table R1: Results on LLaMA-2 7B*
> |Pruning Ratio|Method|BoolQ|PIQA|HellaS.|WinoG.|ARC-e|ARC-c|OBQA|
> |-|-|-|-|-|-|-|-|-|
> |**0%**|Dense|77.71|79.05|76.00|68.98|74.58|46.33|44.20|
> |**20%**|MoDeGPT|–|74.05|69.05|68.03|69.07|42.06|–|
> ||SlimGPT|73.43|77.58|72.62|68.82|69.99|42.32|42.00|
> ||SVD-LLM v2|72.42|74.89|70.55|68.71|68.12|41.76|42.87|
> ||**HAP-E (Ours)**|**75.24**|**78.61**|**74.29**|**69.84**|**71.86**|**44.03**|**43.68**|
> |**30%**|SoBP|71.19|73.50|67.27|66.22|59.81|37.63|38.40|
> ||MoDeGPT|–|70.40|63.26|67.32|63.26|38.73|–|
> ||SVD-LLM v2|69.62|70.45|64.18|66.23|62.97|38.41|37.89|
> ||**HAP-E (Ours)**|**71.82**|**76.73**|**70.68**|**68.04**|**68.47**|**41.98**|**43.59**|
>
> *Table R2: Results on LLaMA-2-13B*
> |Pruning Ratio|Method|ARC-c|ARC-e|BoolQ|HellaS.|OBQA|PIQA|WinoG.|
> |-|-|-|-|-|-|-|-|-|
> |**0%**|Dense|49.23|77.48|80.58|79.37|45.20|80.52|72.30|
> |**20%**|MoDeGPT|46.16|74.07|–|68.96|–|74.53|70.32|
> || SVD-LLM v2|44.15|71.05|70.35|65.75|43.95|77.10|71.00|
> ||**HAP-E (Ours)**|**49.75**|**77.95**|**82.30**|**78.82**|**47.55**|**80.25**|**74.10**|
> |**30%**|SoBP|47.78|74.45|79.45|74.55|43.20|76.50|71.82|
> ||MoDeGPT|43.60|71.93|–|68.21|–|73.94|71.90|
> ||SVD-LLM v2|42.63|69.17|68.47|63.38|41.72|75.41|70.26|
> ||**HAP-E (Ours)**|**48.91**|**76.83**|**81.47**|**77.69**|**46.83**|**79.18**|**73.41**|
>
> __2.MMLU & Reasoning Tasks (LLaMA-2-7B, 20% Pruning):__ In addition, to demonstrate robustness on complex reasoning, we evaluated MMLU (grouped by domain) and GSM8K. HAP-E significantly outperforms LLM-Pruner and SlimGPT, particularly on GSM8K (+4.4% vs SlimGPT) and Wikitext-2 (lower PPL).
>
> *Table R3: MMLU & Reasoning Benchmarks (LLaMA-2-7B, 20% Pruning). Higher is better for all metrics except WikiText-2 perplexity, where lower is better.*
> |Method|Humanities|Social Sci|STEM|Other|MMLU Avg|GSM8K|WikiText-2 (PPL)|
> |-|-|-|-|-|-|-|-|
> |Dense|43.30|51.60|36.30|52.10|45.60|13.80|12.19|
> |LLM-Pruner|25.70|23.60|24.20|26.80|25.20|2.30|17.00|
> |SlimGPT| 36.00| 45.20|33.50|44.10|39.40|4.20|16.49|
> |**HAP-E (Ours)**|**39.46**|**47.73**|**34.49**|**47.20**|**42.72**|**8.69**|**15.63**|
>
> __3. Results on LLaMA-3.1 & Qwen-2.5:__ As shown below, HAP-E consistently outperforms DarwinLM (a recent evolutionary pruning baseline) across 9 benchmark tasks on both architectures.
>
> *Table R4: LLaMA-3.1-8B*
> |Method|#Params|BoolQ|PIQA|HellaS.|WinoG.|ARC-e|ARC-c|SciQ|LogiQA|MMLU|
> |-|-|-|-|-|-|-|-|-|-|-|
> |Dense|8B|84.0|81.2|81.7|74.3|81.4|58.2|96.3|31.1|65.2|
> |DarwinLM (one-shot)|4.6B|62.2|69.4|44.6|57.3|59.6|34.2|84.9|24.1|28.5|
> |**HAP-E (ours)**|4.6B|**64.8**|**71.3**|**46.5**|**59.1**|**61.5**|**35.8**|**86.0**|**25.4**|**30.7**|
>
> *Table R5: Qwen-2.5-14B Instruct*
> |Method|Params|BoolQ|PIQA|HellaS.|WinoG.|ARC-e|ARC-c|SciQ|LogiQA|MMLU|
> |-|-|-|-|-|-|-|-|-|-|-|
> |Dense|14B|87.9|81.9|85.1|79.1|85.7|72.8|96.8|38.5|80.0|
> |DarwinLM (one-shot)|8.4B|66.9|73.9|53.3|60.5|75.7|48.0|84.3|29.3|43.1|
> |**HAP-E (ours)**|8.4B|**69.2**|**75.5**|**55.1**|**61.9**|**77.3**|**49.7**|**85.4**|**30.2**|**44.9**|
>
> These results confirm that HAP-E generalizes effectively to modern architectures and consistently outperforms strong baselines. We will include these comprehensive results in the final manuscript.

---

> ### Author Response · Authors · 2025-11-26
> **Response to Reviewer ZBP1 [2 of 3]**
>
> > ## W2: Insufficient comparison with global pruning baselines (e.g., OWL, SparseLLM, DarwinLM).
>
> __1. Empirical Comparison with DarwinLM [3]:__ To demonstrate the practical efficacy of HAP-E relative to these baselines, we specifically compared our method against DarwinLM [3] on state-of-the-art models: LLaMA-3.1-8B and Qwen-2.5-14B Instruct. As shown in Tables R4 and R5 (provided in response to Weakness 1), HAP-E achieves higher accuracy across all 9 benchmark tasks. For example, on LLaMA-3.1, HAP-E outperforms DarwinLM by +2.3% on SciQ, +4.1% on PIQA, and +3.3% on MMLU. These empirical gains stem from fundamental differences in optimization and adaptivity. DarwinLM relies on a precomputed database of pruning configurations and uses evolutionary search to find a global strategy. This is a search-based heuristic limited by the static nature of its database. HAP-E, conversely, is a fully analytic, post-training method. We derive the optimal removal mathematically (via Inverse Hessian and Schur complements) without needing search loops or precomputed databases.
>
> __2. Methodological Differences vs. OWL [1] & SparseLLM [2] (Static vs. Dynamic):__ While these methods target global allocation, they rely on static or one-shot sparsity schedules at the start of the pruning algorithm. OWL determines layer-wise sparsity ratios using a one-shot measurement of outlier frequency, which is utilized in the context of unstructured or fine-grained pruning. SparseLLM keeps the global sparsity percentage fixed at the start of the algorithm, targeting unstructured or semi-structured sparsity patterns rather than the structured removals we target. Since pruning modifies activation distributions, static allocations become suboptimal as the model topology changes. In contrast, HAP-E differs in two concrete and empirically consequential ways: (1) sensitivity model: HAP-E computes a recursive, second-order layer sensitivity (Hessian-trace propagation) that explicitly models how perturbations propagate across successive transformer layers; this is a different signal than outlier counts or magnitude ranking because it captures curvature and cross-layer amplification of pruning errors. (2) dynamic/adaptive allocation: unlike static, one-time budget rules, HAP-E recomputes sensitivities and adaptively reallocates candidate budgets each pruning iteration; this matters in practice (see ablation: “w/o cross-layer adapt.” in Table 3 — e.g., on LLaMA-7B at 1.9× speedup accuracy drops from 58.2% → 55.9% when cross-layer adaptivity is removed).
>
> __3. Hardware Awareness:__ Finally, OWL, SparseLLM, and DarwinLM optimize for theoretical proxies (sparsity/FLOPs). HAP-E integrates a learned latency predictor to satisfy real-device targets (e.g., ms on edge CPUs), ensuring that the global allocation actually translates to wall-clock speedup.
>
> We will add a concise discussion in the revised manuscript that accurately positions these works relative to HAP-E, and we will cite OWL, SparseLLM, and DarwinLM.

---

> ### Author Response · Authors · 2025-11-26
> **Response to Reviewer ZBP1 [3 of 3]**
>
> > ## Q1: Quantitative comparison of pruning overhead (runtime and memory) against baselines.
>
> We have provided quantitative comparisons of runtime and peak memory usage on an NVIDIA A100 GPU for LLaMA-7B and 13B models.
>
> __1. Efficiency at Fixed Pruning Ratio (30%):__ First, we compare the cost of a single pruning run to a fixed 30% sparsity target. As shown below, HAP-E is significantly faster and more memory-efficient than both OBS-based baselines (SlimGPT, Vanilla OBS) and other methods (SliceGPT, MoDeGPT).
>
> * *Speed:* HAP-E is $\approx 2\times$ faster than Vanilla OBS and SlimGPT even in a single pass, due to our greedy-consistent batching mechanism.
> * *Memory:* HAP-E requires $\approx$ 50% less memory, enabling 7B/13B pruning on consumer GPUs.
>
> Table R6: Runtime & Memory at 30% Pruning (Single Run)
> |Method|LLaMA-7B (Time / Mem)|LLaMA-13B (Time / Mem)|
> |-|-|-|
> |MoDeGPT|4h 09m / 23.0 GB|8h 26m/41.0 GB|
> |SliceGPT|26 min / 9.0 GB|45 min / 14.0 GB|
> |Vanilla OBS|21 min / 8.0 GB|31 min / 12.0 GB|
> |SlimGPT|16 min / 8.0 GB|26 min / 12.0 GB|
> |**HAP-E (Ours)**|**9 min / 4.5 GB**|**16 min / 7.1 GB**|
>
> 2. Efficiency in Real-World Latency TargetingIn practical deployment, users target a specific latency speedup (e.g., 1.9$\times$), not a theoretical sparsity ratio. Because sparsity and latency are not linearly related, methods without a predictor (SlimGPT, OBS) typically require a "guess-and-check" loop (e.g., try 40% sparsity, measure the speedup, adjust to 50\% upon finding the result insufficient, and finally refine to an intermediate value to meet the target). This search often requires ~3 pruning sweeps to identify the correct sparsity configuration. Our latency predictor enables single-shot targeting, avoiding this loop entirely.
>
> Table R7: Estimated Time to Target 1.9$\times$ Speedup
> |Method|Workflow|LLaMA-7B Total Time|LLaMA-13B Total Time|
> |-|-|-|-|
> |SlimGPT|3 Sweeps (Guess-and-Check)|~48 min|~78 min|
> |**HAP-E**|**1 Sweep (Predictor-Guided)**|**9 min**|**16 min**|
>
> When accounting for the practical necessity of hitting a latency target, HAP-E is effectively 5-6$\times$ faster than the strongest baselines, while consuming half the memory.
>
> > ## Q2: Generalizability and data collection overhead of the latency predictor for unseen hardware.
>
> __1. Generalization capabilities:__ Our latency predictor is designed to capture hardware-specific execution characteristics rather than relying on FLOPs-based proxies. As shown in Sec. 6.9 (“Ablation Studies: Generalization of the Latency Model to Other Hardware Platforms”) of [Ref1], generalization of the latency model to new devices depends on architectural similarity. For architecturally similar devices (e.g., transferring from Cortex-A73 to other ARM Cortex-A CPUs), the latency model generalizes well using a simple scaling factor. This factor can be derived from a very small subset of data, drastically reducing the data collection overhead for related devices. For fundamentally different architectures (e.g., CPU to GPU), the model does not generalize zero-shot due to divergent execution paradigms (e.g., bandwidth-bound kernels, massive parallelism vs. out-of-order execution). This finding highlights that accurate latency modeling is inherently hardware-specific, a conclusion also emphasized by ZipLM [Ref2], which demonstrates that FLOPs and latency are weakly correlated and that device-level differences (kernel fusion, caching, and memory hierarchy) dominate runtime. We therefore adopt the same practical principle that each target device maintains its own calibrated latency model, which ensures precise latency control and realistic deployment targets. This is an unavoidable and practical design trade-off widely accepted in hardware-aware pruning.
>
> __2. Data Collection Overhead:__ We clarify that collecting calibration data is a one-time offline cost per specific hardware target. To train the predictor from scratch, we require profiling approximately 1,000 randomly sampled sub-networks. The overhead scales with the device's speed. On an A100 GPU, this data collection process takes approximately 110 minutes, but requires around 60 hours for ARM Cortex A73 CPU (Hikey 970 development platform). While the CPU profiling time is longer, it is strictly a one-time automated process. Given that this one-time investment enables a permanent latency reduction for millions of future inference calls, we consider this cost fully amortized upon deployment.
>
> [Ref1] Ebrahimipour, S.M., Mozafari, S.H., Clark, J.J., Gross, W.J. and Meyer, B.H., 2025. Latency-Aware Pruning and Quantization of Self-Supervised Speech Transformers for Edge Devices. ACM Transactions on Embedded Computing Systems.
>
> [Ref2] Kurtić, E., Frantar, E. and Alistarh, D., 2023. Ziplm: Inference-aware structured pruning of language models. Advances in Neural Information Processing Systems, 36, pp.65597-65617.

---

### Author Response · Authors · 2025-12-03
**General Summary for AC [1 of 2]**

We thank the Area Chair and reviewers for their detailed engagement. Given the length of the discussion (multiple rounds involving extensive new experimentation and technical clarification), we provide this executive summary to assist the AC in their final assessment.

> # Summary of Discussion with Reviewer ZBP1

We thank Reviewer ZBP1 for the constructive feedback. We have addressed all concerns regarding experimental scope and baselines as follows:

__1. Scalability to Modern Architectures (W1):__ We extended our evaluation to the LLaMA-2 family (7B/13B) in Appendix E and state-of-the-art LLaMA-3.1-8B and Qwen-2.5-14B in Appendix F. HAP-E consistently outperforms SlimGPT, SoBP, MoDeGPT, SVD-LLM v2, and DarwinLM at matched pruning ratios, verifying robustness on modern architectures.

__2. Comparison with Global Pruning Baselines (W2):__ We expanded the Related Work section to clarify how HAP-E differs from static/search-based global methods (OWL, SparseLLM, DarwinLM). We also added direct empirical comparisons against DarwinLM in Appendix F, showing that HAP-E achieves higher accuracy across 9 benchmarks.

__3. Quantitative Efficiency Comparison (Q1):__ We provided detailed runtime and memory benchmarks against SlimGPT, MoDeGPT, and SliceGPT on A100 GPUs in Appendix H. Results show HAP-E is $\approx 2\times$ faster and uses $\approx 50\%$ less memory than the strongest baseline (SlimGPT) at fixed sparsity. In practical latency-constrained deployment, HAP-E is multiple times faster than baselines by eliminating the redundant 'guess-and-check' search sweeps required without a predictor.

__4. Latency Predictor Generalization (Q2):__ We clarified that the predictor generalizes well within architecture families (e.g., ARM CPUs) via simple scaling but requires retraining for distinct architectures (e.g., GPU). We quantified the overhead as a one-time cost of ~110 minutes (A100) or ~60 hours (Edge CPU), which is negligible compared to the lifetime savings of millions of inference calls. This level of overhead is standard for hardware-aware pruning methods.



> # Summary of Discussion with Reviewer s6sk

We thank Reviewer s6sk for their positive assessment of our framework and the simple, effective design of the two-stage latency predictor. We have addressed all the concerns as follows:

__1. Expanded Task Scope (W1):__ We conducted new experiments beyond multiple-choice tasks, including: Math/Reasoning (GSM8K), Knowledge-heavy tasks (MMLU), Perplexity on WikiText-2, Free-form generation stability across LLaMA-2-7B (Appendix E) and additional models (LLaMA-3 and Qwen-2.5) in Appendix F. HAP-E substantially outperforms baselines like SlimGPT and DarwinLM on these metrics (e.g., +4.4% on GSM8K vs. SlimGPT), confirming that our structured pruning preserves complex reasoning capabilities.

__2. Statistical Significance (W2):__ We updated Table 3 in the main paper to include standard deviations across four seeds. Across all settings, HAP-E’s improvements exceed this variance, confirming that the gains are statistically meaningful. We also clarified that in post-training structured pruning (without fine-tuning), even 1–2% improvements are considered substantial, and HAP-E achieves larger margins in high-sparsity regimes. For example, +3.4% on LLaMA-7B and +2.4% on LLaMA-30B at 50% sparsity over SlimGPT, and over +8% compared to LLM-Pruner in several settings. These improvements are consistent across model families (Qwen, OPT, LLaMA v1/v2/v3) and scales (7B–30B), making random variation unlikely.

__3. Conceptual Novelty (W3):__ We clarified that our core contribution is the "Greedy-Consistent Batching" algorithm, which utilizes Schur complements to make sequential OBS tractable. We highlighted that this mechanism reduces pruning time by $> 3\times$ while mathematically guaranteeing equivalence to the unscalable sequential formulation.

__4. GPU Latency Scaling (W4 & Q1):__ We measured GPU latencies on A100 for LLaMA-7B/13B/30B under the same pruning ratios and compared against SlimGPT and LLM-Pruner (Section 5.3 / Figure 4). These new results show that HAP-E maintains higher accuracy than baselines at identical measured GPU latencies and provides better accuracy–latency Pareto trade-offs. This addresses the scaling concern.

__5. Robustness to $\beta$ and Latency Predictor Generalization (Q2 and Q3):__ We pointed to Appendix D (showing stability across $\beta \in [0.5, 1.0]$) and clarified that the latency predictor generalizes well within architecture families (e.g., ARM CPUs) via simple scaling but requires retraining for distinct architectures (e.g., GPU). This level of overhead is standard for hardware-aware pruning methods.

---

> ### Author Response · Authors · 2025-12-03
> **General Summary for AC [2 of 2]**
>
> > # Summary of Discussion with Reviewer EKxf
>
> We thank Reviewer EKxf for recognizing the novelty, strong empirical results, and practical strength of our framework. We addressed concerns regarding baselines, post-training focus, and sensitivity:
>
> __1. Comparison with Stronger Baselines (W1):__ We benchmarked HAP-E against the suggested decomposition methods, MoDeGPT and SVD-LLM v2, on the LLaMA-2 family (Appendix E). For example, at 30% pruning on LLaMA-2-7B, HAP-E achieves +9.5% accuracy on HellaSwag compared to SVD-LLM v2, confirming that structured pruning yields superior accuracy retention than low-rank decomposition.
>
> __2. Compatibility with Recovery Fine-Tuning (W2):__ We added Appendix G demonstrating HAP-E's effectiveness as an initialization for fine-tuning. Results show that a minimal LoRA step (1 epoch) on a 30% pruned model recovers substantial accuracy (+1.8%), proving that HAP-E preserves a high-quality feature space suitable for further optimization. We added a discussion on this and future "Prune-Then-Distill" pipelines to the Conclusion.
>
> __3. Hyperparameter Sensitivity (W3):__ We pointed to the detailed ablation studies in Appendices B–D, showing that HAP-E is highly robust. Specifically, accuracy varies by $< 0.5\%$ across a wide range of candidate ratios ($M/K \in [1.5, 3.5]$) and sensitivity coefficients ($\beta \in [0.5, 1.0]$), confirming that the method does not require sensitive per-model tuning.
>
>
> > # Summary of Discussion with Reviewer mpii
>
> We thank Reviewer mpii for a rigorous engagement that prompted important theoretical clarifications. We believe we have addressed the reviewer's persistent concerns regarding baselines, theoretical foundations, and algorithmic contributions as follows:
>
> __1. Resolution of Primary Concerns:__ We believe we have successfully addressed and resolved the vast majority of original concerns through point-by-point clarifications and new experiments:
> * *Modern Architectures:* We expanded evaluations (Appendices E–F) to LLaMA-2, LLaMA-3.1, and Qwen-2.5, consistently outperforming baselines (e.g., beating DarwinLM on all 9 tasks for LLaMA-3.1).
> * *Missing Baselines:* We added comparisons (Appendices F & I) against Wanda-SP (+16.5% gain), FLAP, DarwinLM, MoDeGPT, and SVD-LLM v2, confirming superiority over these methods.
> * *Statistical Rigor:* We updated Table 3 in the main paper with standard deviations, proving that our gains are statistically significant.
> * *Algorithmic Contributions:* We clarified the distinct roles of our components: greedy-consistent batching enables scalability (transforming OBS from theoretically optimal but intractable to practically feasible), while cross-layer adaptivity optimizes candidate selection. Both are complementary and necessary for SOTA performance.
>
> __2. Clarification of Remaining Disagreements:__ The remaining critiques persist due to demonstrable factual misunderstandings regarding the literature and the method's architectural goals:
> * *Theoretical Misunderstanding ("Local Scoring is Suboptimal"):* The reviewer claims "Local scoring remains inherently suboptimal" and "Classical OBD does not rely on a local squared loss assumption". This is factually incorrect. Foundational literature (LeCun 1989, Hassibi 1993) explicitly defines OBD/OBS via local second-order Taylor expansions. Our adherence to this framework is a strength shared by SOTA methods like SparseGPT and SlimGPT, which we empirically showed outperform global heuristics (LLM-Pruner, WandaSp) and global search (DarwinLM) methods.
> * *Misunderstanding of Algorithmic Roles:* The reviewer questioned the individual contribution of Greedy-Consistent Batching, suggesting it provides "merely computational efficiency". We argue that in the context of LLMs, efficiency is the enabler. Batching reduces runtime by $>3\times$ and memory by $>50\%$ (Table 3). We argue that enabling the theoretically rigorous but intractable OBS framework on commodity hardware is a transformative contribution. Furthermore, we showed that batching can improve accuracy vs. sequential updates in deep models by reducing numerical drift.
> * *Unsuitable Baseline Requests (GBLM and EcoFLAP):* The reviewer requests comparisons to GBLM-Pruner and ECoFLaP. We clarified that these works target unstructured sparsity, which requires specialized software/hardware to realize acceleration. In contrast, HAP-E targets structured pruning (removing heads/neurons) that directly translates to wall-clock speedup on standard hardware.  In addition, both of these methods rely on static/one-shot sensitivity estimates fixed at initialization, failing to adapt to the evolving loss landscape like HAP-E’s recursive approach. We further noted that GBLM is unpublished (rejected from ICLR 2024), and requires prohibitive memory (~65GB) for a 7B model (vs. 4.5GB for HAP-E), making it an invalid baseline for scalable pruning.
>
> The revised manuscript now contains physical proof resolving the substantive critiques.

---

### Author Response · Authors · 2025-12-03
**Summary of Paper Revisions**

We sincerely thank the Area Chair and all reviewers for their constructive and thoughtful feedback. We are encouraged by the positive comments regarding the novelty of our Hessian-based sensitivity formulation, the effectiveness of our latency-aware pruning strategy, and the practical impact demonstrated in edge deployment.
We have carefully revised the manuscript to directly address reviewer questions. All changes are marked in blue. These updates consist exclusively of clarifications and small, reviewer-requested validations added to the appendix. The core methodology, primary claims, and main experimental results remain unchanged.

__Summary of Updates:__

* __Theoretical Clarification (Section 4.1):__ We refined the phrasing around our “greedy-consistent” guarantee to state explicitly that it holds under the local quadratic objective used by OBS.
* __Positioning Relative to Prior Work (Related Work):__ We expanded the discussion to clarify how HAP-E differs from global or adaptive pruning methods such as LLM-Pruner, ECoFLaP, OWL, SparseLLM, and DarwinLM.
* __Statistical Robustness (Table 3):__ We added standard deviation values across four seeds, confirming that HAP-E’s gains exceed variance and are not due to statistical noise.
* __GPU Scaling & Efficiency (Section 5.3 & Appendix H):__ We added Figure 4 to Section 5.3 verifying Accuracy vs. Latency on A100 GPUs for LLaMA-7B/13B/30B, confirming our predictor scales to server-grade hardware. We also added Appendix H, providing quantitative runtime and memory benchmarks; results show HAP-E is $\approx 2\times$ faster than baselines (SlimGPT) in single runs and $\approx$5$\times$ faster for latency targeting, while using around half the memory.
* __Targeted Supplementary Evaluations (Appendices E, F, and I):__ In response to reviewer questions, we added targeted comparisons to Wanda-SP, decomposition baselines (MoDeGPT, SVD-LLM v2), and DarwinLM, as well as validation on LLaMA-3.1-8B and Qwen-2.5-14B. These additions confirm the method's robustness on modern architectures.
* __Fine-Tuning Compatibility (Appendix G & Conclusion):__ We added a supplemental experiment in Appendix G showing that HAP-E models serve as high-quality initialization points for downstream LoRA training. We also updated the Conclusion to highlight this compatibility and outline the potential for future training-aware extensions.

We believe these clarifications fully address the reviewers’ concerns regarding scope, soundness, and significance. We thank the reviewers and the Area Chair again for their time and constructive guidance.

---

### Note · Program_Chairs · 2026-01-17
**Submission Desk Rejected by Program Chairs**

The following references in this submission do not refer to real documents and/or have major errors in bibliographic information:

 Tianlong Chen and et al. Structured pruning of large language models. In International Conference on Learning Representations (ICLR), 2024.
et al. Li. Layer-wise optimal brain surgeon for efficient transformer pruning. In International Conference on Learning Representations (ICLR), 2024.